# Unfolding $E_{11}$

**Nicolas Boulanger[1⋆], Paul P. Cook[2†], Josh A. O'Connor[1‡§] and Peter West[2,3∘]**

**1** Physique de l'Univers, Champs et Gravitation, Université de Mons –
UMONS, Place du Parc 20, 7000 Mons, Belgium
**2** Department of Mathematics, King's College London, Strand, London, WC2R 2LS, UK
**3** Mathematical Institute, University of Oxford, Woodstock Road, Oxford, OX2 6GG, UK

⋆ nicolas.boulanger@umons.ac.be , † paul.cook@kcl.ac.uk ,
‡ josh.o'connor@umons.ac.be , ∘ peter.west540@gmail.com

## Abstract

We work out the unfolded formulation of the fields in the non-linear realisation of $E_{11}$. Using the connections in this formalism, we propose, at the linearised level, an infinite number of first-order duality relations between the dual fields in $E_{11}$. In this way, we introduce extra fields that do not belong to $E_{11}$ and we investigate their origin. The equations of motion of the fields are obtained by taking derivatives and higher traces of the duality relations.



## Contents

§ FRIA grantee of the Fund for Scientific Research – FNRS, Belgium.

# 1  Introduction

E theory contains an infinite number of fields labelled by a level grading [1]. The only degrees of freedom in E theory are those of the bosonic sector of supergravity, so in eleven dimensions we have those of the graviton and the three-form field that are found at levels zero and one. At higher levels, one finds fields which provide dual descriptions of these degrees of freedom. Although these higher level fields have their own equations of motion, they also satisfy duality relations which are first-order in derivatives, relating them to gravity or to the three-form.

The duality equations in E theory have been formulated as equivalence relations, that is, they hold up to certain gauge transformations [2,3]. While this is a perfectly correct way to proceed, the aim of this paper is to formulate these relations as conventional gauge-covariant equations. We use the unfolded formalism[1] to achieve this, expressing the linearised equations of the theory in terms of a set of interlinked equations[2] relating the space-time derivatives of each field to a set of connections and zero-forms. Concretely, in this paper, we propose an infinite set of duality relations for the dual fields in E theory, written using their associated first-order connections. In this way we find, at the linearised level, the duality relations in the form of conventional, gauge-covariant equations that do not receive any extra contribution under a gauge transformation. This is possible since the unfolded formalism introduces extra fields that compensate for the gauge freedom, and they can all be gauged away algebraically. We also find that taking derivatives and higher traces of the duality relations leads directly to the linearised E theory equations of motion.

Since these subjects are unfamiliar to many readers, we will now give a brief review of some of the material. E theory is the non-linear realisation of the semi-direct product of $E_{11} = E_8^{+++}$ with its vector representation $\ell_1$ and it contains the bosonic fields and equations of motion of

---

[1]The term 'unfolding' only started to appear explicitly in Vasiliev's work in [4], although the techniques were already used earlier in [5,6].

[2]This idea of expressing a set of PDEs as an exterior differential system is old. It was initiated by E. Cartan, see [7] for a pedagogical exposition, although the introduction of the infinite-dimensional module of zero-forms representing the propagating degrees of freedom came later and is due to Vasiliev. For a more detailed, modern exposition, see [8] and references therein.

all maximal supergravity theories [1–3, 9]. For a review, see [10]. The adjoint representation of $E_{11}$ contains the fields of the theory, and they all depend on the generalised space-time whose coordinates correspond to $\ell_1$ generators. At levels zero and one we find the graviton and the three-form. At level two we find a six-form which is dual to the three-form, and at level three we find a mixed-symmetry field $h_{a_1 \cdots a_8, b}$ that is dual to the graviton. At higher levels the number of fields grows rapidly, and their roles are mostly unknown, but precisely one field at each level is understood to be dual to the original graviton or three-form. For example, at level four we find $A_{a_1 \cdots a_9, b_1 b_2 b_3}$, $B_{a_1 \cdots a_{10}, b, c}$, and $C_{a_1 \cdots a_{11}, b}$, the first of which is dual to the three-form.

The structure of each equation is fixed by $E_{11}$ symmetry. This has been worked out at the full non-linear level up to level three, that is, for gravity, the three-form [1, 9], six-form [2], and the dual graviton [11]. The equations of motion have also been also worked out at the linearised level for the fields in $E_{11}$ at level four [12]. The irreducible representation corresponding to the dynamics of the theory has been worked out completely, and it shows that the only dynamical degrees of freedom are those of the graviton and the three-form [13, 14]. Thus, although the non-linear realisation contains an infinite set of dual fields, the only degrees of freedom are those of maximal supergravity. If one restricts generalised space-time to be just the usual space-time then the equations of motion agree precisely with those of supergravity [2, 3, 12, 15, 16]. This restriction corresponds to the fact that one is considering a point particle theory and not taking account of the presence of branes [17]. In this sense the dynamics is completely known.

The large symmetries of the $E_{11}$ non-linear realisation also leave invariant an infinite set of duality relations which have so far been computed at low levels. In fact, acting with $E_{11}$ on the equations of motion and the duality relations at low levels, one generates the equations at higher and higher levels. The enormous $E_{11}$ symmetry fixes[3] the equations of motion and the duality relations precisely, although this has only been carried out explicitly at low levels so far. In particular, one can find unique quantities that are inert under rigid global $E_{11}$ symmetries and which also transform covariantly under the local symmetries of the theory. As such these quantities can be set to zero while still preserving $E_{11}$ symmetry. So far, work in E theory has been to find the equations of motion and duality relations rather than an action principle.[4] The $E_{11}$-invariant duality relations between the three-form $A_{a_1 a_2 a_3}$ and the six-form $A_{a_1 \cdots a_6}$ [2, 3] and between the graviton $h_a{}^b$ and the dual graviton $h_{a_1 \cdots a_8, b}$ [19] have been worked out at the full non-linear level while the higher duality relation between the three-form and the $A_{a_1 \cdots a_9, b_1 b_2 b_3}$ field has been worked out at the linearised level [12]. Relations at higher levels can be found in much the same way.

While a classification of the generators, and hence fields, of $E_{11}$ is unknown, the fields that have no blocks of ten or eleven indices are known [20]. As well as the fields from levels zero to three, that is $h_a{}^b$, $A_{a_1 a_2 a_3}$, $A_{a_1 \cdots a_6}$, and $h_{a_1 \cdots a_8, b}$, there is an infinite number of fields in $E_{11}$ that have additional blocks of nine antisymmetric indices, the first of which is $A_{9,3} = A_{a_1 \cdots a_9, b_1 b_2 b_3}$ at level four. It was proposed in [20] that these fields are dual to the fields at levels zero and one. In [21], analytic expressions relating the towers of dual fields in $E_{11}$ were found. An infinite set of dual action principles in the gravity sector were proposed in [22], and an infinite set of first-order duality relations in the gauge field sector generalising the relation between the three-form and $A_{9,3}$ was proposed in [23], supporting the conjecture of [20]. Relations between dual fields in $E_{11}$ were further discussed in [24].

Some of the fields in $E_{11}$ have blocks of ten antisymmetric indices and these are the fields responsible for all the maximally supersymmetric gauged supergravity theories in the different

---

[3]In each non-linear realisation, the form (i.e. the tensor structure and combination of terms) of the equations is fixed by the global and local symmetries of the theory.

[4]However, we note the $E_{11}$ pseudo-Lagrangian that was worked out in [18] using a different formalism.

dimensions. In works carried out across a twenty year period all these theories were classified (see [25,26] and references therein) and they can also be found in a simple way from $E_{11}$ [27,28]. The first example is the field $B_{a_1\cdots a_{10},b,c} = B_{a_1\cdots a_{10},(b,c)}$ at level four whose reduction from eleven to ten dimensions leads to a nine-form field [12] which is responsible for Romans theory [29]. Key to the work of [25,26] was the tensor hierarchy construction [30] which was also obtained in the $E_{11}$ non-linear realisation [31,32]. Aside from all the fields that we mentioned above, there remains an infinite number of fields in $E_{11}$ whose meaning is as yet unknown.

It is a result of the infinite set of duality relations that the theory only contains the degrees of freedom of the graviton and the three-form. In the context of $E_{11}$ alone these duality relations are equivalence relations meaning that they only hold modulo certain gauge transformations [2,3,33,34]. These have been worked out for the low level duality relations [2,3,12] and they are also completely known at the linearised level [35]. As mentioned in [34], the equivalence relations and the associated gauge transformations can be deduced by integrating up the equations of motion that follow from $E_{11}$ symmetry, as initiated in [23]. In the present paper we will work out several examples of this integration. Thus at least in principle the equivalence relations can be completely worked out solely in the context of $E_{11}$.

It was explained in the first paper on $E_{11}$ [1] that the duality relation between gravity and dual gravity could be written as a conventional equation rather than an equivalence relation by adding a nine-form. However, this field is not among those in $E_{11}$. Although the duality relations can be systematically and correctly given as as equivalence relations, it would be good to have duality relations which are of a more conventional kind and for this to be the case one must add fields in addition to those found in $E_{11}$. These fields do not contribute to the degrees of freedom of the theory but they ensure that the duality relations are gauge-covariant rather than equivalence relations. It is important to note that one does not need fields beyond those already in $E_{11}$ to formulate the equations of motion as these just involve the irreducible $E_{11}$ fields. For example, the dual gravity equation of motion involves just the irreducible $h_{a_1\cdots a_8,b}$ field which is subject to the condition $h_{[a_1\cdots a_8,b]} = 0$, that is, the equation of motion does not feature the extra nine-form field that is needed to write down a gauge-covariant duality relation between the graviton and the dual graviton.

There are various interesting and elegant ways to present the equations of eleven-dimensional supergravity [36]. A notable example is given by the rheonomic approach of [37] – see [38,39] for reviews – as well as the on-shell constraint approach developed in [40–44], see e.g. [45] for a review and recent developments involving pure spinors. Along those lines, a duality symmetric superspace formulation of supergravity was worked out in [46] that incorporates the fermionic degrees of freedom. Adding fermions or supersymmetry to theories with enormous Kac-Moody symmetries is an open problem. From the E theory perspective, fermions can be introduced by taking them to transform under the Cartan involution subalgebra of $E_{11}$. Progress can be found in [14,47] (see also [48]) which followed corresponding work on $E_{10}$ [49–51].

It is also possible to write down duality relations in the context of a parent action which contains implicitly the field and its dual. This is referred to as **off-shell dualisation**. One can eliminate either of the fields from the parent action using their equations of motion to obtain an action for the original field or an action for the dual field. In the first paper on $E_{11}$ such a parent action relating the graviton and the dual graviton was presented in any dimension [1]. This led to the duality relation between them, the correct equation of motion for the graviton, and also the well-known linearised action for the graviton. It also led to the equation of motion for the dual graviton and the action for the dual graviton, although this was not explicitly presented in [1]. This justified the use of the field $h_{a_1\cdots a_{D-3},b}$ to describe the dual graviton in $D$ dimensions and explained the presence of $h_{a_1\cdots a_8,b}$ at level three in $E_{11}$.

This was made explicit and generalised to higher-spin fields in [52] where it was observed that the dynamics of the dual graviton given in [1] agreed with the first account by Curtright of the dual graviton in five dimensions [53] and in any number of dimensions [54].

Parent actions have been used in a number of different contexts. As mentioned previously, dual action principles for all possible dual gravity fields were found in [22], where dualisation was performed on empty columns of the Young tableau. A parent action relating the three-form and the $A_{9,3}$ field was given in [23]. Relatedly, the dual fields in the IIA theory contained in the $E_{11}$ approach were introduced in the corresponding parent actions in [55].

One advantage of this approach is that it begins with an action principle that is invariant under gauge transformations in the conventional way, and the equations that follow do not need to be thought of as equivalence relations. Thus in this approach one finds the fields needed. The role of extra fields in preserving both gauge invariance and the propagating degrees of freedom was spelled out in [56].

The tensor hierarchy algebra $S(E_{11})$ is a differential graded superalgebra, and it underlies the construction of the dynamics of another $E_{11}$ field theory [18,48,57]. At grade zero $S(E_{11})$ contains $E_{11}$ itself alongside a tower of highest weight representations. The original motivation for tensor hierarchy algebras was to encode gauged supergravities into one algebraic structure, including the embedding tensor and the hierarchy of form fields for form degree up to and beyond the space-time dimension $D$ [58,59]. The role of tensor hierarchy algebras in extended geometry has been spelled out in [60–67]. Previous attempts to encode all these form fields involved extending the global $E_{11-D}$ symmetry either to $E_{11}$ [27,31,32] or to a graded Borcherds superalgebra $\mathcal{B}(E_{11-D})$ [68–71] (see also [60]). In contrast to these Borcherds algebras, tensor hierarchy algebras $S(E_{11-D})$ are constructed so as to preserve the Hodge duality of form fields for $1 \leq p \leq D-3$ and extends this duality to as many grades as possible. Both superalgebras can be 'very-extended' in the sense that we can work with $\mathcal{B}(E_{11})$ and $S(E_{11})$. One of the main aims of [57] was to work towards a theory based on $E_{11}$ which contains an enlarged spectrum of fields given by $S(E_{11})$ at grade zero, therefore including fields belonging to $E_{11}$ and to a tower of additional highest weight representations.

In the same way that the non-linear realisation of $E_{11}$ encodes the maximal supergravity theories, the non-linear realisation of the infinite-dimensional algebra $A_1^{+++}$ generalises pure gravity in four dimensions [19,72,73]. Alongside the graviton, the non-linear realisation of $A_1^{+++}$ features the infinite tower of dual gravity fields in four dimensions and an infinite set of fields whose role is less clear. The relationship between dual gravitons in $A_1^{+++}$ and dual action principles for gravity was studied in our previous work [74].

**Outline of the paper.** We take a bottom-up approach by applying the unfolded formalism [4–6,75,76] for mixed-symmetry fields in flat space [77–80] to the fields in $E_{11}$. The procedure that we apply to each dual field can be summarised as follows: (1) introducing a set of unfolded variables, i.e. connections; (2) writing down and solving the first few unfolded equations; and (3) proposing duality relations between our dual fields in terms of the first-order connections. This provides the extra fields required to formulate the $E_{11}$ duality relations as conventional, gauge-covariant equations. We also discuss the relation between the duality relations and the equations of motion. Following the familiar path, we derive the equations of motion from the duality relations, but we also show how to find the duality relations by integrating the equations of motion for several important examples that occur in $E_{11}$, as was initiated in [23].

The structure of this paper is as follows. In Section 2, we give a more detailed account of the $E_{11}$ non-linear realisation and we compute the gauge transformations of the fields at level four. Then, in Section 3, we review the unfolded formalism and we apply it to the fields in $E_{11}$ up to level three: the graviton, three-form, six-form, and dual graviton. Linearised duality relations between all these fields are obtained. In Section 4, we consider the higher dual

three-form field $A_{9,3}$ at level four and we use the unfolded formalism to work out its equation of motion and its duality relation with the three-form. We also unfold the $B_{10,1,1}$ field at level four.

In Section 5, we work out the linearised equations of motion for all higher dual fields in the $E_{11}$ non-linear realisation. We propose an infinite number of first-order duality relations that relate these fields. We also find all the gauge parameter constraints that must be imposed for our proposed duality relations to be gauge-covariant. Linearised equations of motion for all dual fields in $E_{11}$ are worked out by taking derivatives and traces of the duality relations, and these equations are then integrated back up to recover the duality relations with all the extra fields. These equations and our proposed duality relations match those of the $E_{11}$ non-linear realisation at low levels where they have already been worked out, which justifies *a posteriori* the choice of variables in the unfolded formulation of each field. We also discuss the spectrum of extra fields and we investigate their origin inside representations of $E_{11}$.

We perform a similar analysis in Section 6 for the $A_1^{+++}$ non-linear realisation: unfolding the dual fields, proposing linearised duality relations featuring extra fields, obtaining linearised equations of motion, and investigating where the extra fields come from. In Section 7, building upon [80], we provide explicit frame-like action principles for the higher dual three-form field in $E_{11}$ and the higher dual graviton in $A_1^{+++}$. This is followed by a discussion of our results in Section 8. We provide tables of useful representations in Appendix A, and in Appendix B we briefly unfold the dual fields in the $K_{27}$ non-linear realisation [81].

**Summary of notation.** The $i^{\text{th}}$ fundamental representation of the $E_{11}$ algebra is denoted by $\ell_i$ and defined to be the highest weight representation whose highest weight is the fundamental weight associated with vertex $i$ in the Dynkin diagram of $E_{11}$ below.

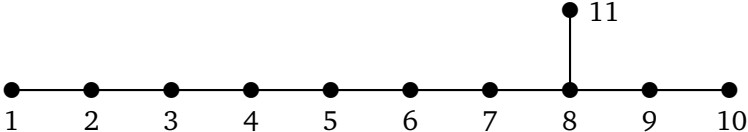

Tables of generators for useful representations of $E_{11}$ are given in Appendix A

Differential forms will often be written with their form degree as a subscript, although we do not give a subscript to any zero-forms. Wedge products are omitted and are taken to be implicit. In this paper $\mathbb{Y}[h_1, \ldots, h_n]$ denotes an irreducible Young diagram with $n$ columns, where $h_i$ is the height of the $i^{\text{th}}$ column. We use $\phi_{h_1, \ldots, h_n}$ to denote an irreducible mixed-symmetry field that transforms in the representation associated with this diagram. For example, $T_{3,2,1,1}$ denotes an irreducible rank-seven field $T_{a_1 a_2 a_3, b_1 b_2, c, d}$ whose symmetry type is given by the Young diagram

$$
\begin{array}{c} \phantom{X} \end{array} \quad = \quad \mathbb{Y}[3, 2, 1, 1]. \tag{1}
$$

Fields with blocks of symmetric or antisymmetric indices can be written as

$$
S_{a(n)} := S_{a_1, \cdots, a_n} \sim \mathbb{Y}[1, \ldots, 1], \qquad A_{a[n]} := A_{a_1 \cdots a_n} \sim \mathbb{Y}[n]. \tag{2}
$$

A reducible field transforms as a tensor product of irreducible representations, and we denote their symmetry types by tensor products of Young diagrams. Blocks of antisymmetric indices in a reducible field are separated by a comma if they belong to the same irreducible component, and they are separated by a vertical bar if they belong to different components. For example, we write $\Psi_{4|3,2,2}$ to denote a rank-eleven reducible field $\Psi$ that transforms as

$$
\Psi_{a_1 a_2 a_3 a_4 | b_1 b_2 b_3, c_1 c_2, d_1 d_2} \quad \sim \quad \begin{array}{c} \phantom{X} \end{array} \otimes \begin{array}{c} \phantom{X} \end{array} \quad = \quad \mathbb{Y}[4] \otimes \mathbb{Y}[3, 2, 2]. \tag{3}
$$

The first component is a four-form, and the mixed-symmetry nature of the second component implies that $\Psi$ obeys the following over-antisymmetrisation constraints:

$$\Psi_{a_1 a_2 a_3 a_4 | [b_1 b_2 b_3, c_1] c_2, d_1 d_2} = \Psi_{a_1 a_2 a_3 a_4 | [b_1 b_2 b_3 |, c_1 c_2, | d_1] d_2} = \Psi_{a_1 a_2 a_3 a_4 | b_1 b_2 b_3, [c_1 c_2, d_1] d_2} = 0. \quad (4)$$

The above diagrams are associated with GL($D$) tensors if we work in $D$ space-time dimensions. As we implicitly did above, to a given irreducible tensor we usually prescribe a Young tableau associated with the Young diagram depicted. If we consider SO($1, D-1$) tensors instead, then the irreducible fields also obey specific trace constraints.

## 2 The non-linear realisation of $E_{11}$

The fields of the non-linear realisation are parameters of a generic $E_{11}$ group element, although we can gauge away everything at negative levels using the local symmetry given by the Cartan involution invariant subgroup of $E_{11}$ denoted by $I_c(E_{11})$. As a result, the group element belongs to the Borel subgroup of $E_{11}$ and the fields of the theory are in a one-to-one correspondence with the generators of the Borel algebra. Up to level three the fields are the graviton and the three-form together with their magnetic duals, namely the six-form and the dual graviton:

$$h_{ab}, \qquad A_3 = A_{a_1 a_2 a_3}, \qquad A_6 = A_{a_1 \cdots a_6}, \qquad h_{8,1} = h_{a_1 \cdots a_8, b}. \quad (5)$$

Every field in $E_{11}$ is GL(11) irreducible, so they all obey over-antisymmetrisation constraints. For example, the dual graviton $h_{8,1}$ is a mixed-symmetry field that satisfies $h_{[a_1 \cdots a_8, b]} = 0$.

The fields of the theory all depend on an infinite number of coordinates that are associated with the generators of the $\ell_1$ representation, but here we take them to depend only on the usual coordinate $x^a$ at level zero. This corresponds to the fact that we are constructing a theory of point particles and not branes – see [17] for more details.

At levels four and above one finds an infinite tower of higher dual fields associated with the fields in (5) [20]. Exactly one dual field appears at each level together with some fields whose interpretations are less obvious, but many of them lead to the gauged supergravities [27, 31, 32]. For instance, at level four in $E_{11}$ there are three fields given by

$$A_{9,3} = A_{a_1 \cdots a_9, b_1 b_2 b_3}, \qquad B_{10,1,1} = B_{a_1 \cdots a_{10}, b, c}, \qquad C_{11,1} = C_{a_1 \cdots a_{11}, b}. \quad (6)$$

The first field $A_{9,3}$ is a higher dual [20] that provides an equivalent description of the three-form degrees of freedom, while the second field $B_{10,1,1}$ is the eleven-dimensional origin of Romans theory [12]. Indeed, reduction to ten dimensions leads to a nine-form $B_{a_1 \cdots a_9} := B_{a_1 \cdots a_9 11,11,11}$ that in turn leads to a supergravity theory with a cosmological constant. Similarly, one can find the next-to-top forms $A_{a_1 \cdots a_{D-1}}$ for the supergravities in dimension $D$ and in each case these lead to gauged supergravities with a cosmological constant. In this way one finds all such theories and one can recover in a simple way their classification that was found over many years. Such fields in lower dimensions can arise from fields in eleven dimensions that have one block of ten indices since in $D$ dimensions such a block can be made up of $11-D$ internal indices and a next-to-top form with $D-1$ indices. However, fields with blocks of eleven indices can not contribute in this way. Thus there are still many fields in the non-linear realisation whose role we do not understand, such as the third field $C_{11,1}$ at level four.

At level five there are four fields in the adjoint:

$$A_{9,6}, \quad B_{10,4,1}, \quad C_{11,3,1}, \quad C_{11,4}. \quad (7)$$

Recall that the subscripts are a shorthand for the symmetry types of each field. For example, $B_{10,4,1}$ denotes the GL(11)-irreducible field $B_{a_1 \ldots a_{10}, b_1 \ldots b_4, c}$. The first field $A_{9,6}$ is a higher dual

counterpart to the six-form, and the second field $B_{10,4,1}$ plays a role in gauged supergravity theories in lower dimensions, as mentioned above [27].

At level six there are nine fields in the adjoint:

$$h_{9,8,1}, \quad B_{10,6,2}, \quad B_{10,7,1}, \quad B_{10,8}, \quad C_{11,4,3}, \quad C_{11,5,1,1}, \quad C_{11,6,1}^{(1)}, \quad C_{11,6,1}^{(2)}, \quad C_{11,7}. \tag{8}$$

The first field $h_{9,8,1}$ is a higher dual that propagates the degrees of freedom of the graviton or the dual graviton, and the three fields with blocks of ten antisymmetric indices once again play a role in the gauged supergravities [27]. The field $C_{11,6,1}$ appears in $E_{11}$ with multiplicity two, and we have used a superscript to label each of them.

At higher levels in $E_{11}$ one finds three infinite families of higher dual fields at higher levels with the following Young diagrams:

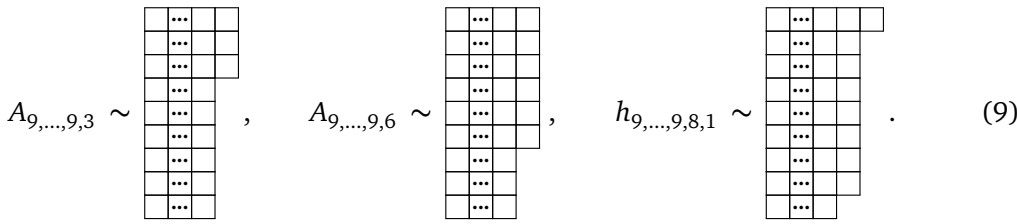

$$A_{9,\dots,9,3} \sim \quad , \quad A_{9,\dots,9,6} \sim \quad , \quad h_{9,\dots,9,8,1} \sim \quad . \tag{9}$$

It has been shown that these are all the fields in the non-linear realisation if we ignore fields whose tableaux contain columns of height ten or eleven [20].

One can work out irreducible representations of $I_c(E_{11}) \ltimes \ell_1$ [13, 14]. At level zero this reduces to the Poincaré group, so the procedure is similar to the Wigner method generalised to eleven dimensions – see [82]. The massless particle representation for which only the usual momentum is non-zero has worked out in all detail. Despite the infinite number of fields, one finds that the degrees of freedom in this representation are just those of gravity and the three-form [13]. This representation corresponds to the free on-shell states in the non-linear realisation. Higher level fields are related by rather trivial duality relations which are invariant under the little group. Thus we conclude that the very many additional fields in $E_{11}$ do not lead to any further degrees of freedom. While this is apparent for the dual fields and the fields that lead to gauged supergravities, it must also apply to the fields whose meaning we do not yet understand.

The form of the full non-linear equations for the fields follow uniquely from the non-linear realisation. This has been worked out for the graviton, three-form, six-form [2, 3], and more recently the dual graviton [11, 33, 83]. In each case, these fields are taken to depend only on the level zero coordinates at the end of the calculation, although to derive these results one requires the fields to depend on the higher level coordinates. Linearised equations for the fields at level four have also been found [12]. As such, the dynamics predicted by the non-linear realisation is known, at least if we restrict fields to depend on the usual space-time. This has been less completely worked out in lower dimensions [15, 16] and for gauged supergravities, but the conclusion is the same.

Duality relations that are first-order in derivatives relate all the dual fields to each other. The prototypical example is the relation between the three-form and the six-form, but the full non-linear duality relations have also been worked out between the three-form and six-form and between gravity and dual gravity. The existence of such relations ensures that the non-linear realisation contains only the degrees of freedom mentioned above and not, for example, many copies of the graviton arising from the infinite tower of dual gravity fields at higher levels.

The symmetries of the $E_{11}$ non-linear realisation lead uniquely to the equations of motion which turn out to be gauge-invariant even though this symmetry was not used to construct them. It is not understood why this happens. Integrating these equations one finds the duality relations although these are not gauge-invariant but hold as equivalence relations. This means

they hold up to some gauge transformations which also follow from the integration procedure. Alternatively one can derive the duality relations directly using the symmetries of the non-linear realisation but then one must take account of the gauge transformations.

The gauge transformations have parameters $\Lambda^A$ which belong to the $\ell_1$ representation

$$\Lambda^A = \{\Lambda^a, \Lambda_{a_1 a_2}, \Lambda_{a_1 \cdots a_5}, \Lambda_{a_1 \cdots a_7, b}, \Lambda_{a_1 \cdots a_8}, \ldots\}. \tag{10}$$

Linearised gauge transformations for fields $A_\alpha$ in the non-linear realisation have been deduced from $E_{11}$ [35] and they take the form

$$\delta_\Lambda A_\alpha = (D_\alpha)_A{}^B \partial_B \Lambda^A, \tag{11}$$

where $[R^\alpha, l_A] = -(D_\alpha)_A{}^B l_B$ are the commutation relations for $E_{11} \ltimes \ell_1$. Up to level two, we find that the gauge transformations of the graviton, three-form and six-form are

$$\delta_\Lambda h_{ab} = \partial_{(a}\Lambda_{b)}, \qquad \delta_\Lambda A_{a_1 a_2 a_3} = \partial_{[a_1}\Lambda_{a_2 a_3]}, \qquad \delta_\Lambda A_{a_1 \cdots a_6} = \partial_{[a_1}\Lambda_{a_2 \cdots a_6]}, \tag{12}$$

where we only consider derivatives with respect to the coordinates at level zero. At level three there are two gauge parameters $\Lambda^{(1)}_{a_1 \ldots a_7, b}$ and $\Lambda^{(2)}_{a_1 \ldots a_8}$ and the dual graviton transforms as

$$\delta_\Lambda h_{a_1 \cdots a_8, b} = \partial_{[a_1}\Lambda^{(1)}_{a_2 \cdots a_8], b} + \partial_{[a_1}\Lambda^{(2)}_{a_2 \cdots a_8]b} - \partial_b \Lambda^{(2)}_{a_1 \cdots a_8}. \tag{13}$$

We have scaled the parameters as they appear in $E_{11}$ by a factor of $\frac{3}{4}$.

Now we will work out the $E_{11}$ gauge transformations of the fields at level four. At this level we have six distinct gauge parameters:

$$\Lambda^{(1)}_{a_1 \cdots a_8, b_1 b_2 b_3}, \quad \Lambda^{(2)}_{a_1 \cdots a_9, b, c}, \quad \Lambda^{(3)}_{a_1 \cdots a_9, b_1 b_2}, \quad \Lambda^{(4)}_{a_1 \cdots a_{10}, b}, \quad \Lambda^{(5)}_{a_1 \cdots a_{10}, b}, \quad \Lambda^{(6)}_{a_1 \cdots a_{11}}. \tag{14}$$

Note that $\Lambda^{(4)}_{10,1}$ and $\Lambda^{(5)}_{10,1}$ have the same symmetry type since $l^{10,1} \in \ell_1$ has multiplicity two. The transformation of the $B_{10,1,1}$ field contains three parameters and is given by

$$\delta_\Lambda B_{a_1 \cdots a_{10}, b, c} = \frac{756}{5} \partial_{[a_1}\Lambda^{(2)}_{a_2 \cdots a_{10}], b, c} + \frac{126}{11}\left(\partial_{[a_1}\Lambda^{(4)}_{a_2 \cdots a_{10}](b,c)} - \frac{11}{10}\partial_{(b|}\Lambda^{(4)}_{a_1 \cdots a_{10}, |c)}\right)$$
$$+ 6\left(\partial_{[a_1}\Lambda^{(5)}_{a_2 \cdots a_{10}](b,c)} - \frac{11}{10}\partial_{(b|}\Lambda^{(5)}_{a_1 \cdots a_{10}, |c)}\right). \tag{15}$$

The transformation of $C_{11,1}$ is given by

$$\delta_\Lambda C_{a_1 \cdots a_{11}, b} = \frac{693}{10}\partial_{[a_1}\Lambda^{(4)}_{a_2 \cdots a_{11}], b} + \frac{11}{10}\partial_{[a_1}\Lambda^{(5)}_{a_2 \cdots a_{11}], b} - \frac{6}{5}\partial_b \Lambda^{(6)}_{a_1 \cdots a_{11}}. \tag{16}$$

We have scaled the gauge parameters in $\delta_\Lambda B_{10,1,1}$ and $\delta_\Lambda C_{11,1}$ by an inverse factor of 756,000. The gauge transformation of the field associated with the higher dual three-form contains the last two parameters and it is given by

$$\delta_\Lambda A_{a_1 \cdots a_9, b_1 b_2 b_3} = -12\, \partial_{[a_1}\Lambda^{(1)}_{a_2 \cdots a_9], b_1 b_2 b_3} + \frac{9}{5}\left(\partial_{[a_1}\Lambda^{(3)}_{a_2 \cdots a_9][b_1, b_2 b_3]} + \frac{7}{9}\partial_{[b_1|}\Lambda^{(3)}_{a_1 \cdots a_9, |b_2 b_3]}\right). \tag{17}$$

## 3 Unfolding $E_{11}$ up to level three

### 3.1 A brief review of unfolding

In this section we review some basic aspects of unfolding [5, 6] (see e.g. [75, 76]) with particular emphasis on mixed-symmetry gauge fields in flat space-time [77–79], see also Section 2 of [80]. Later in this section we will work out some examples at the linearised level.

The unfolded formulation of a theory is a way to express its dynamics as a set of first-order differential equations, thereby generalising the Hamiltonian formalism. In an unfolded system, the fundamental variables are an infinite tower of differential forms $W_{[p_\alpha]}{}^\alpha$ where $p_\alpha$ is the form degree and $\alpha$ is a set of indices. In practice, these variables are identified with objects such as the vielbein, spin connection, field strengths, and so on.

We must distinguish between off-shell and on-shell unfolding. For a given system in eleven dimensions with local degrees of freedom, unfolding off-shell means that the indices $\alpha$ of each variable $W_{[p_\alpha]}{}^\alpha$ are associated with an irreducible GL(11) representation. In contrast, on-shell unfolding amounts to imposing appropriate trace constraints on the zero-form variables so that they are valued in irreducible Lorentz representations. The strictness of these constraints can vary. For many fields it is required that the zero-forms are all completely traceless. Later we will observe that the on-shell unfolding of fields with complicated Young tableaux may feature zero-forms satisfying higher trace constraints where some traces survive and others do not.

The equations of an unfolded theory are a tower of first-order differential equations

$$F^\alpha := \mathrm{d}W^\alpha + Q^\alpha(W) = 0, \tag{18}$$

where $Q^\alpha$ are wedge product polynomials of the forms. Integrability of this differential system leads to the conditions

$$Q^\alpha \frac{\partial Q^\beta}{\partial W^\alpha} = 0. \tag{19}$$

Every differential form $W_{[p_\alpha]}{}^\alpha$ is associated with a generalised curvature $F_{[p_\alpha+1]}{}^\alpha$ of form degree $p_\alpha + 1$, and if $p_\alpha \geq 1$ then there is also a gauge parameter $\lambda_{[p_\alpha-1]}{}^\alpha$ of form degree $p_\alpha - 1$. Using (19) and its differential consequences, one can show that the tower of unfolded equations (18) is invariant under the gauge transformations

$$\delta_\lambda W_{[p_\alpha]}{}^\alpha = \mathrm{d}\lambda_{[p_\alpha-1]}{}^\alpha - \lambda^\beta \frac{\partial Q^\alpha}{\partial W^\beta}. \tag{20}$$

Of course, if $p_\alpha$ is zero then the $\mathrm{d}\lambda_{[p_\alpha-1]}{}^\alpha$ term is not present. Similarly, one can use (19) to obtain the Bianchi identity

$$\mathrm{d}F^\alpha - F^\beta \frac{\partial Q^\alpha}{\partial W^\beta} = 0. \tag{21}$$

For variables with form degree $p_\alpha > 1$, the equation $\delta_\lambda W^\alpha = 0$ can be satisfied identically, and this expresses the fact that there are reducibility (gauge-for-gauge) parameters. One is led to a chain of parameters $\lambda_{[p_\alpha-1]}{}^\alpha, \ldots, \lambda_{[1]}{}^\alpha$ of some higher-order gauge transformations.

It is known how to unfold fields that are totally symmetric or antisymmetric, and here we will outline the unfolding procedure for the most general mixed-symmetry fields. Consider a tensor field $\varphi_{h_1,\ldots,h_n}$ whose subscript corresponds to the irreducible GL(11) tableau $\mathbb{Y}[h_1,\ldots,h_n]$ with $n$ columns. In order to unfold $\varphi_{h_1,\ldots,h_n}$ we must rewrite $\{W_{[p_\alpha]}{}^\alpha\}$ (possibly after a redefinition) as an infinite tower of zero-forms $\{C^{\beta_i}\}$ and a finite tower of forms $\{X_{[h_i]}{}^{\alpha_i}\}$ with positive form degrees $h_i$. The full tower can be written as

$$\underbrace{e_{[h_1]}{}^{\alpha_1}, \quad \omega_{[h_2]}{}^{\alpha_2}, \quad X_{[h_3]}{}^{\alpha_3}, \quad \ldots, \quad X_{[h_n]}{}^{\alpha_n}}_{h_i\text{-form connections}}, \quad \underbrace{C^{\beta_1}, \quad C^{\beta_2}, \quad \ldots}_{\text{zero-forms}}, \tag{22}$$

where $X_{[h_1]}{}^{\alpha_1}$ and $X_{[h_2]}{}^{\alpha_2}$ are labelled $e_{[h_1]}{}^{\alpha_1}$ and $\omega_{[h_2]}{}^{\alpha_2}$, respectively. Unfolding off-shell,

the forms $X_{[h_i]}{}^{\alpha_i}$ and the zero-forms $C^{\beta_i}$ are valued in the following GL(11) tableaux:

$$\alpha_1 \sim \mathbb{Y}[h_2,\ldots,h_n], \qquad\qquad \beta_1 \sim \mathbb{Y}[h_1+1,\ldots,h_n+1], \qquad (23a)$$

$$\alpha_2 \sim \mathbb{Y}[h_1+1,h_3,\ldots,h_n], \qquad \beta_2 \sim \mathbb{Y}[h_1+1,\ldots,h_n+1,1], \qquad (23b)$$

$$\alpha_3 \sim \mathbb{Y}[h_1+1,h_2+1,h_4,\ldots,h_n], \quad \beta_3 \sim \mathbb{Y}[h_1+1,\ldots,h_n+1,1,1], \qquad (23c)$$

$$\vdots \qquad\qquad\qquad\qquad \vdots$$

$$\alpha_n \sim \mathbb{Y}[h_1+1,\ldots,h_{n-1}+1], \qquad \beta_k \sim \mathbb{Y}[h_1+1,\ldots,h_n+1,1,\ldots,1]. \qquad (23d)$$

In order to unfold our generic field $\varphi_{h_1,\ldots,h_n}$, we need to write down all the equations of the theory as an integrable Pfaffian system (18) that relates each variable in the tower with the differential of the one before it. The unfolded equations can be written schematically as

$$\mathrm{d}e^{\alpha_1} + \omega^{\alpha_2} = 0, \quad \mathrm{d}\omega^{\alpha_2} + X^{\alpha_3} = 0, \quad \ldots, \quad \mathrm{d}X^{\alpha_n} + C^{\beta_1} = 0, \quad \mathrm{d}C^{\beta_1} + C^{\beta_2} = 0, \quad \ldots \quad (24)$$

Unfolding the metric-like field $\varphi_{h_1,\ldots,h_n}$ on-shell amounts to imposing some trace constraints on the infinite set of zero-forms $\{C^{\beta_i}\}$ such that the labels $\beta_i$ effectively denote irreducible Lorentz (spin-)tensors. Upon solving these unfolded equations, the zero-form trace constraints will be equivalent to the equation of motion of $\varphi_{h_1,\ldots,h_i}$.

In order to write the unfolded equations in full, we need to define the background vielbein one-form for Minkowski space-time in Cartesian coordinates $h^a := \mathrm{d}x^\mu \delta_\mu^a$ and we write

$$h^{a[n]} = h^{a_1\cdots a_n} := h^{a_1} \wedge \cdots \wedge h^{a_n}. \qquad (25)$$

As such, a $p$-form $\omega_{[p]}$ is locally written as $\omega_{[p]} = \frac{1}{p!} \mathrm{d}x^{\mu_1} \cdots \mathrm{d}x^{\mu_p} \omega_{\mu_1\cdots\mu_p} = \frac{1}{p!} h^{a_1} \cdots h^{a_p} \omega_{a_1\cdots a_p}$.

## 3.2 Unfolding linearised gravity

Although it is well-known, it will be instructive to recall the unfolded formulation of linearised gravity – see, for example, the reviews [75, 76]. As explained above, one needs to introduce the variables presented in (22):

$$e_{[1]}{}^a, \quad \omega_{[1]}{}^{ab}, \quad C^{ab,cd}, \quad \ldots, \qquad (26)$$

where $\omega_{[1]}{}^{ab} = \omega_{[1]}{}^{[ab]}$ and $C^{ab,cd} = C^{[ab],cd} = C^{ab,[cd]}$ with the constraint $C^{[ab,c]d} = 0$ ensuring that the primary zero-form $C^{ab,cd}$ is valued in the irreducible GL(11) representation with Young diagram $\mathbb{Y}[2,2] = \begin{array}{|c|c|}\hline & \\\hline & \\\hline\end{array}$. The first two variables are the usual Cartan connection one-forms: the vielbein $e_{[1]}{}^a = \mathrm{d}x^\mu e_\mu{}^a$ and spin connection $\omega_{[1]}{}^{ab} = \mathrm{d}x^\mu \omega_\mu{}^{ab}$. They are followed by an infinite tower of zero-forms. Unfolding on-shell will require all these zero-forms to be valued in irreducible representations of the Lorentz group SO(1, 10) and consequently $C^{ab,cd}$ will be traceless, but for now we unfold off-shell[5] and we will not impose any trace constraints on the zero-forms. For the variables with positive form degree, one can think of the lower indices as world or form indices and the upper indices as tangent space indices. Since we are working at the linearised level in flat space-time, the distinction is less important.

Writing the schematic Pfaffian system in (24) completely using background vielbeins, the first two[6] unfolded equations are

$$\mathrm{d}e_{[1]}{}^a + h_b\, \omega_{[1]}{}^{ab} = 0, \qquad (27a)$$

$$\mathrm{d}\omega_{[1]}{}^{ab} + h_{cd}\, C^{ab,cd} = 0. \qquad (27b)$$

---

[5]Off-shell unfolding for non-linear Yang-Mills and Einstein gravity theories in flat space can be found in [84]. Off-shell unfolding in (A)dS background is discussed in [85].

[6]We say that these are the 'first' and 'second' unfolded equations because we are counting the number of derivatives. The first equation constrains the torsion, and the second constrains the curvature.

These equations are invariant under the gauge transformations

$$\delta e_{[1]}{}^a = d\lambda^a + h_b\,\alpha^{ab}, \qquad \delta\omega_{[1]}{}^{ab} = d\alpha^{ab}, \tag{28}$$

where $\alpha^{ab} = \alpha^{[ab]}$. It will be useful to express the unfolded equations in components as

$$\partial_{[a}e_{b]|c} + \omega_{[a|b]c} = 0, \tag{29a}$$

$$\partial_{[a}\omega_{b]|cd} + C_{ab,cd} = 0, \tag{29b}$$

with gauge transformations

$$\delta e_{a|b} = \partial_a\lambda_b - \alpha_{ab}, \qquad \delta\omega_{a|bc} = \partial_a\alpha_{bc}. \tag{30}$$

Decomposing $e_{a|b}$ into irreducible parts, we write

$$e_{a|b} = h_{ab} + \widehat{A}_{ab}, \tag{31}$$

where $h_{ab} = h_{(ab)}$ and $\widehat{A}_{ab} = \widehat{A}_{[ab]}$. These fields have the transformations

$$\delta h_{ab} = \partial_{(a}\lambda_{b)}, \qquad \delta\widehat{A}_{ab} = \partial_{[a}\lambda_{b]} - \alpha_{ab}. \tag{32}$$

We can use $\alpha_{ab}$ to set $\widehat{A}_{ab}$ to zero. In order to preserve this choice, we may carry out residual gauge transformations whereby $\alpha_{ab} = \partial_{[a}\lambda_{b]}$, leaving only the graviton $h_{ab}$.

Solving (29a) for $\omega_{a|bc}$ leads to

$$\omega_{a|bc} = 2\,\partial_{[b}h_{c]a} - \partial_a\widehat{A}_{bc}. \tag{33}$$

This is the usual spin connection with the opposite sign. In the $E_{11}$ non-linear realisation, among the positive roots at level zero, we find the field $h_{ab}$ with the gauge transformation of (32). However, at level zero we also find the field $\widehat{A}_{ab}$ which has the local $I_c(E_{11})$ transformation with parameter $\alpha_{ab}$ in (32). After solving (29a) and (29b) for $C_{ab,cd}$ in terms of the irreducible fields, we find $C_{ab,cd} = -2\,\partial_{[a}\partial_{[c}h_{d]b]}$. Off-shell, we interpret $C_{ab,cd}$ as the linearised Riemann tensor $R_{ab,cd}$.

Now we show how to proceed from off-shell unfolding towards on-shell unfolding by imposing appropriate zero-form trace constraints. Working on-shell, the well-known Ricci-flat equation of motion is equivalent to the primary zero-form being traceless:

$$R_{ac,b}{}^c = R_{ab} = 0 \qquad \Longleftrightarrow \qquad \text{Tr}(C_{ab,cd}) = 0. \tag{34}$$

The zero-form $C^{ab,cd}$ is now not only GL(11) irreducible but also Lorentz irreducible with the same Young tableau $\mathbb{Y}[2,2]$. On-shell, we interpret $C^{ab,cd}$ as the linearised Weyl tensor.

### 3.3 Unfolding dual gravity

The dual graviton at level three is represented by the irreducible field $h_{8,1} = h_{a_1\cdots a_8,b}$ and its unfolded formulation requires the introduction of the variables

$$e_{[8]}{}^a, \quad \omega_{[1]}{}^{a_1\cdots a_9}, \quad C^{a_1\cdots a_9,b_1b_2}, \quad \ldots \tag{35}$$

For now we will unfold the dual graviton off-shell so that the zero-form $C_{9,2} = C_{a_1\cdots a_9,b_1b_2}$ does not obey any trace constraints. The first two unfolded equations are given by

$$de_{[8]}{}^a + h_{b[8]}\,\omega_{[1]}{}^{b[8]a} = 0, \tag{36a}$$

$$d\omega_{[1]}{}^{a[9]} + h_{b[2]}\,C^{a[9],b[2]} = 0, \tag{36b}$$

with gauge symmetries

$$\delta e_{[8]}{}^a = d\lambda_{[7]}{}^a - h_{b[8]}\alpha^{b[8]a}, \qquad \delta\omega_{[1]}{}^{a[9]} = d\alpha^{a[9]}. \tag{37}$$

In components, the equations take the form[7]

$$\partial_{[a_1}e_{a_2\cdots a_9]|b} + \omega_{[a_1|a_2\cdots a_9]b} = 0, \tag{38a}$$

$$\partial_{[a_1}\omega_{a_2]|b_1\cdots b_9} + C_{b_1\cdots b_9,a_1a_2} = 0, \tag{38b}$$

and the gauge transformations are given by

$$\delta e_{a_1\cdots a_8|b} = \partial_{[a_1}\lambda_{a_2\cdots a_8]|b} - \alpha_{a_1\cdots a_8 b}, \qquad \delta\omega_{a|b_1\cdots b_9} = \partial_a\alpha_{b_1\cdots b_9}. \tag{39}$$

The reducible fields and gauge parameters of the local transformations can be decomposed into irreducible components as

$$e_{a_1\cdots a_8|b} = h_{a_1\cdots a_8,b} + \widehat{A}_{a_1\cdots a_8 b}, \qquad \lambda_{a_1\cdots a_7|b} = \lambda^{(1)}_{a_1\cdots a_7,b} + \lambda^{(2)}_{a_1\cdots a_7 b}, \tag{40}$$

with Young tableaux

$$\tag{41}$$

where $h_{[a_1\cdots a_8,b]} = 0$ and $\lambda^{(1)}_{[a_1\cdots a_7,b]} = 0$. In terms of all these fields and gauge parameters, the transformations of (39) become

$$\delta h_{a_1\cdots a_8,b} = \partial_{[a_1}\lambda^{(1)}_{a_2\cdots a_8],b} - \frac{1}{9}\left(\partial_b\lambda^{(2)}_{a_1\cdots a_8} - \partial_{[a_1}\lambda^{(2)}_{a_2\cdots a_8]b}\right), \tag{42a}$$

$$\delta\widehat{A}_{a_1\cdots a_9} = \partial_{[a_1}\lambda^{(2)}_{a_2\cdots a_9]} - \alpha_{a_1\cdots a_9}. \tag{42b}$$

Using the gauge symmetry with the nine-form parameter $\alpha_9$ in (39) we can set $\widehat{A}_9$ to zero. This choice is preserved under residual gauge transformations whereby $\alpha_{a_1\cdots a_9} = \partial_{[a_1}\lambda^{(2)}_{a_2\cdots a_9]}$ and only the dual graviton field $h_{8,1}$ will remain with the transformation of (42a).

Choosing for the moment to keep this extra field $\widehat{A}_9$ and its gauge symmetry, we can solve (38a) for $\omega_{[1]}{}^9$ in terms of both the irreducible fields as

$$\omega_{a|b_1\cdots b_9} = -9\,\partial_{[b_1}h_{b_2\cdots b_9],a} - \partial_a\widehat{A}_{b_1\cdots b_9}. \tag{43}$$

Solving the second equation (38b) for the primary zero-form $C_{9,2}$ implies that

$$C_{a_1\cdots a_9,b_1b_2} = 9\,\partial_{[b_1}\partial_{[a_1}h_{a_2\cdots a_9],b_2]}. \tag{44}$$

The linearised dual gravity equation of motion [1, 12] is given by

$$\partial^{[b}\partial_{[a_1}h_{a_2\cdots a_8c],}{}^{c]} = 0 \qquad \Longleftrightarrow \qquad \text{Tr}(C_{9,2}) = 0, \tag{45}$$

and this equation transforms in the irreducible GL(11) representation depicted by the Young tableau $\mathbb{Y}[8,1]$. Unfolding on-shell, the correct trace constraint is to take the zero-form $C_{9,2}$ to

---

[7]In this paper, we rescale the components of $p$-forms by a factor of $p!$ whenever we write unfolded equations in components.

be completely traceless, which is equivalent to it being an irreducible Lorentz representation. Note that we could have started by imposing this trace constraint inside the second unfolded equation (36b) thereby encoding the equations of motion from the very beginning.

We now make contact with the $E_{11}$ non-linear realisation. The field $h_{8,1}$ is the level three field in the theory and it has the same gauge transformation as in (42a) [12]. As done in the first papers on $E_{11}$ [1, 86], one can choose to add a nine-form $\widehat{A}_9$ with the gauge transformation (42b). One can reverse the above steps by starting from the $E_{11}$ field $h_{8,1}$ and then adding the nine-form field $\widehat{A}_9$ to build $e_{[8]}{}^1$ as in (40) and then form the connection $\omega_{[1]}{}^9$ with its shift symmetry as in (43).

### 3.4 The dual gravity duality relation

In the first paper on $E_{11}$ the duality relation

$$\omega_{a|b_1 b_2} = \frac{1}{4}\varepsilon_{b_1 b_2}{}^{c_1\cdots c_9}\omega_{a|c_1\cdots c_9} \tag{46}$$

was proposed [1]. This has been written in terms of the connections in the unfolded formalism, and it is invariant under the local transformations (30) and (39) provided we identify

$$\alpha_{a_1 a_2} = \frac{1}{4}\varepsilon_{a_1 a_2}{}^{b_1\cdots b_9}\alpha_{b_1\cdots b_9}. \tag{47}$$

By taking derivatives of (46) one obtains a Hodge duality between curvatures

$$2\,\partial_{[b_1}\partial_{[a_1}h_{a_2]b_2]} = -\frac{9}{4}\varepsilon_{b_1 b_2}{}^{c_1\cdots c_9}\partial_{[a_1}\partial_{[c_1}h_{c_2\cdots c_9],a_2]}. \tag{48}$$

Taking the trace on $a_2$ and $b_2$ leads to the equation of motion for gravity (34). If we instead contract both sides of (48) with $\varepsilon^{a_2 b_1 b_2 d_1\cdots d_8}$ then we find the linearised dual graviton equation of motion (45).

Notice that (48) is really just a relation between primary (curvature) zero-forms

$$C_{a_1\cdots a_9, b_1 b_2} \propto \varepsilon_{a_1\cdots a_9}{}^{c_1 c_2}C_{c_1 c_2, b_1 b_2}, \tag{49}$$

under which their tracelessness and over-antisymmetrisation constraints are exchanged:

$$\left.\begin{array}{l}\mathrm{Tr}(C_{2,2}) = 0\,,\\ C_{2,2}\ \text{is GL(11) irreducible}\end{array}\right\} \iff \left\{\begin{array}{l}C_{9,2}\ \text{is GL(11) irreducible},\\ \mathrm{Tr}(C_{9,2}) = 0\,.\end{array}\right. \tag{50}$$

This is just an exchange between equations of motion and Bianchi identities under dualisation. Going on-shell, one takes the trace of $C_{9,2}$ to find that the right-hand side of (49) vanishes, recovering the dual gravity equation of motion (45). Similarly, eliminating the dual graviton leads to the usual Ricci-flat equation for gravity (34) [12, 87]. Thus the first-order duality relation (46) can be used to deduce the linearised equations of motion for each field.

The dual gravity equation of motion (45) propagates the correct degrees of freedom in the sense that it corresponds to the UIR of the Poincaré group ISO(1, 10) induced from the $\mathbb{Y}[8, 1]$ UIR of the Wigner little group SO(9) for a massless particle. Relatedly, $\mathbb{Y}[1, 1]$ and $\mathbb{Y}[8, 1]$ are two equivalent representations of the little group. See [88, 89] for more details and [90, 91] for the general case.

Recalling that $\omega_{[1]}{}^2$ and $\omega_{[1]}{}^9$ are solutions of the zero-torsion equations (29a) and (38a), respectively, the duality relation (46) can be considered as a sum of two equations:

$$2\,\partial_{[b_1}h_{b_2]a} \doteq \frac{9}{4}\varepsilon_{b_1 b_2}{}^{c_1\cdots c_9}\partial_{c_1}h_{c_2\cdots c_9, a}\,, \tag{51a}$$

$$\partial_a\widehat{A}_{b_1 b_2} \doteq -\frac{1}{4}\varepsilon_{b_1 b_2}{}^{c_1\cdots c_9}\partial_{c_1}\widehat{A}_{c_2\cdots c_9 a}\,. \tag{51b}$$

Equation (51a) follows from the $E_{11}$ non-linear realisation. While it is not gauge-covariant, such equations were understood to be equivalence equations meaning that it only holds up to gauge transformations of the form $\partial_a \alpha_{b_1 b_2}$. We write $\doteq$ rather than $=$ to denote such relations. This is one of an infinite set of duality relations that are invariant under the symmetries of the $E_{11}$ non-linear realisation. The second equation (51b) contains the $E_{11}$ field $\widehat{A}_{a_1 a_2}$ at level zero which can be gauged away using the local $I_c(E_{11})$ transformation with parameter $\alpha_{a_1 a_2}$. It also contains the extra nine-form field $\widehat{A}_{a_1 \cdots a_9}$ which does not belong to $E_{11}$ and so it does not appear in the non-linear realisation. This duality relation is invariant under the above local transformations provided the gauge parameter constraint (47) holds.

We remark that (51a) forces the differential gauge parameters $\lambda_1$ and $\lambda_8^{(2)}$ in (32) and (40) to be related by

$$\partial_{[a_1} \lambda_{a_2]} = -\frac{1}{4} \varepsilon_{a_1 a_2}{}^{b_1 \cdots b_9} \partial_{b_1} \lambda^{(2)}_{b_2 \cdots b_9} . \tag{52}$$

As a result, it is impossible to relate these parameters to each other locally, but this problem is circumvented with the introduction of extra fields. Returning to the unfolded picture, if we decompose $\omega_{[1]}{}^2$ into GL(11) irreducible components

$$\omega_{a|b_1 b_2} = \omega^{(1)}_{b_1 b_2, a} + \omega^{(2)}_{a b_1 b_2} , \qquad \square \otimes \begin{array}{c}\square\\\square\end{array} = \begin{array}{cc}\square & \square\\\square\end{array} \oplus \begin{array}{c}\square\\\square\\\square\end{array} , \tag{53}$$

then we find

$$\omega^{(1)}_{b_1 b_2, a} = 2 \partial_{[b_1} h_{b_2] a} - \frac{2}{3} \left( \partial_a \widehat{A}_{b_1 b_2} - \partial_{[b_1} \widehat{A}_{b_2] a} \right) , \qquad \omega^{(2)}_{a b_1 b_2} = -\partial_{[a} \widehat{A}_{b_1 b_2]} . \tag{54}$$

These components transform as

$$\delta \omega^{(1)}_{b_1 b_2, a} = -\frac{2}{3} \left( \partial_a \alpha_{b_1 b_2} - \partial_{[b_1} \alpha_{b_2] a} \right) , \qquad \delta \omega^{(2)}_{a b_1 b_2} = -\partial_{[a} \alpha_{b_1 b_2]} . \tag{55}$$

Note that both sides of each irreducible component of (46) transform only with $\alpha_2$ and $\alpha_9$ that are related by (47). One could have chosen to work in a gauge where the extra fields $\widehat{A}_2$ and $\widehat{A}_9$ are set to zero, in which case (46) reduces to (51a) with residual gauge symmetry such that the gauge parameters are related by $\alpha_{a_1 a_2} = \partial_{[a_1} \lambda_{a_2]}$ and $\alpha_{a_1 \cdots a_9} = \partial_{[a_1} \lambda_{a_2 \cdots a_9]}$.

Lastly, it is important to note that one can obtain the duality relation (46) by integrating the curvature relation (49). The constants of integration describe the gauge freedom of this duality relation. Introducing extra fields allows us to absorb these gauge terms so that we end up with a duality relation that holds exactly and not just as an equivalence relation.

## 3.5 The three-form and six-form fields

Alongside gravity and dual gravity, the $E_{11}$ non-linear realisation contains a three-form $A_3$ and its dual six-form $A_6$ at levels one and two, respectively. Their unfolded formulations were worked out in [23] and here we provide a summary.

In order to begin unfolding the three-form and the six-form[8] fields, we write down their first unfolded equations in terms of their respective field strengths $F_4$ and $F_7$:

$$\mathrm{d}A_{[3]} + h_{a[4]} F^{a[4]} = 0 , \qquad \mathrm{d}A_{[6]} + h_{a[7]} F^{a[7]} = 0 . \tag{56}$$

These equations are invariant under the usual gauge transformations

$$\delta A_{[3]} = \mathrm{d}\lambda_{[2]} , \qquad \delta A_{[6]} = \mathrm{d}\lambda_{[5]} . \tag{57}$$

---

[8]This analysis is only given to linear order. In the full non-linear theory, it would not be $F_7$ but rather $G_7 := F_7 - \frac{1}{2} A_3 F_4$ (with all seven indices antisymmetrised) that is associated with the six-form potential.

The dynamics of a propagating three-form or six-form field is known to require (see e.g. [23]) an infinite number of field strength gradients[9]

$$F^{(n)}_{a_1a_2a_3a_4,b_1,\ldots,b_n} := \partial_{b_1}\cdots\partial_{b_n}\partial_{[a_1}A_{a_2a_3a_4]}\,, \qquad F^{(n)}_{a_1\cdots a_7,b_1,\ldots,b_n} := \partial_{b_1}\cdots\partial_{b_n}\partial_{[a_1}A_{a_2\cdots a_7]}\,. \tag{58}$$

Writing $F^{(n)}_{4,1,\ldots,1}$ and $F^{(n)}_{7,1,\ldots,1}$ in terms of the original three-form and six-form fields is possible upon solving a tower of unfolded equations. For example, the first unfolded equations (56) are solved by $F_{a[4]} = 4\,\partial_{[a_1}A_{a_2a_3a_4]}$ and $F_{a[7]} = 7\,\partial_{[a_1}A_{a_2\cdots a_7]}$, so the primary zero-forms are the usual four-form and seven-form field strengths, while the second unfolded equations

$$\mathrm{d}F^{a[4]} + h_b\, F^{a[4],b} = 0\,, \qquad \mathrm{d}F^{a[7]} + h_b\, F^{a[7],b} = 0\,, \tag{59}$$

are solved by $F^{(1)}_{a[4],b} = \partial_{\langle b} F_{a[4]\rangle}$ and $F^{(1)}_{a[7],b} = \partial_{\langle b} F_{a[7]\rangle}$, where angled brackets denote projection on Young tableaux associated with the diagrams $\mathbb{Y}[4,1]$ and $\mathbb{Y}[7,1]$. Combining these first two solutions leads to $F^{(1)}_{a[4],b} = 4\,\partial_b\partial_{[a_1}A_{a_2a_3a_4]}$ and $F^{(1)}_{a[7],b} = 7\,\partial_b\partial_{[a_1}A_{a_2\cdots a_7]}$. Notice that the GL(11) irreducibility properties of $F_{4,1}$ and $F_{7,1}$ in (59) are equivalent to $\partial_{[a_1}F_{a_2\cdots a_5]} = 0$ and $\partial_{[a_1}F_{a_2\cdots a_8]} = 0$ which are solved by writing the primary zero-forms as field strengths.

Integrability of the first unfolded equation leads to an infinite tower of unfolded equations relating all the higher field strength gradients. Every such equation is a relation between GL(11) irreducible zero-forms:

$$\mathrm{d}F^{a[4],b_1,\ldots,b_{n-1}} = h_{b_n} F^{a[4],b_1,\ldots,b_{n-1},b_n}\,, \qquad \mathrm{d}F^{a[7],b_1,\ldots,b_{n-1}} = h_{b_n} F^{a[7],b_1,\ldots,b_{n-1},b_n}\,, \tag{60}$$

which includes (59) for $n = 1$. In components, (60) can be expressed as

$$F_{a[4],b_1,\ldots,b_{n-1},b_n} = \partial_{\langle b_n} F_{a[4],b_1,\ldots,b_{n-1}\rangle}\,, \qquad F_{a[7],b_1,\ldots,b_{n-1},b_n} = \partial_{\langle b_n} F_{a[7],b_1,\ldots,b_{n-1}\rangle}\,, \tag{61}$$

where angled brackets denote projection onto the irreducible tableaux

$$F_{a[4],b_1,\ldots,b_n} \;\sim\; \begin{array}{|c|c|c|c|} \hline a_1 & b_1 & \cdots & b_n \\ \hline a_2 \\ \cline{1-1} a_3 \\ \cline{1-1} a_4 \\ \cline{1-1} \end{array}\,, \qquad\qquad F_{a[7],b_1,\ldots,b_n} \;\sim\; \begin{array}{|c|c|c|c|} \hline a_1 & b_1 & \cdots & b_n \\ \hline a_2 \\ \cline{1-1} \vdots \\ \cline{1-1} a_7 \\ \cline{1-1} \end{array}\,. \tag{62}$$

It is useful to define the unfolded modules of the three-form and the six-form which contain an infinite number of irreducible zero-form variables:

$$\mathcal{T}(A_3) := \left\{ F^{(n)}_{4,1^n} \,\middle|\, n \in \mathbb{N} \right\} = \left\{ F^{(0)}_4, F^{(1)}_{4,1}, F^{(2)}_{4,1,1}, \ldots \right\}\,, \tag{63}$$

$$\mathcal{T}(A_6) := \left\{ F^{(n)}_{7,1^n} \,\middle|\, n \in \mathbb{N} \right\} = \left\{ F^{(0)}_7, F^{(1)}_{7,1}, F^{(2)}_{7,1,1}, \ldots \right\}\,. \tag{64}$$

The unfolded equations (59) and (60) all now imply that every zero-form is an irreducible projection of the gradient of the previous one. The first object in each module is a primary zero-form, and we note that these modules are analogous to those of gravity and dual gravity containing the primary (Weyl) zero-forms $C_{2,2}$ and $C_{9,2}$ that were used earlier in this section:

$$\mathcal{T}(h_{1,1}) = \left\{ C^{(n)}_{2,2,1^n} \,\middle|\, n \in \mathbb{N} \right\} = \left\{ C^{(0)}_{2,2}, C^{(1)}_{2,2,1}, C^{(2)}_{2,2,1,1}, \ldots \right\}\,, \tag{65}$$

$$\mathcal{T}(h_{8,1}) = \left\{ C^{(n)}_{9,2,1^n} \,\middle|\, n \in \mathbb{N} \right\} = \left\{ C^{(0)}_{9,2}, C^{(1)}_{9,2,1}, C^{(2)}_{9,2,1,1}, \ldots \right\}\,. \tag{66}$$

All the descendants, i.e. the higher gradients $C_{2,2,1,\ldots,1}$ and $C_{9,2,1,\ldots,1}$, are contained inside these modules. The above zero-forms are all irreducible GL(11) tensors when unfolding off-shell.

---

[9]In particular, one considers [75] an expansion of the field in a neighbourhood of some point in space-time using $F^{(n)}$ as the Taylor coefficients. Thus higher-order gradients $F^{(n)}$ describe the field at longer distances.

Unfolding on-shell implies that all the zero-forms are irreducible Lorentz tensors and hence all completely traceless. From equation (59) we see that the tracelessness of $F_{4,1}$ is equivalent to the Maxwell equations

$$\partial^a F_{ab_1b_2b_3} = 0, \qquad \partial^a F_{ab_1\cdots b_6} = 0.$$ (67)

The equation of motion and Bianchi identities for a dynamical three-form and all information about higher gradients of its field strength are encoded in the Lorentz irreducibility properties of the zero-forms in $\mathcal{T}(A_3)$. Similarly, the properties of the zero-forms in $\mathcal{T}(A_6)$ encode the dynamics of the six-form. Note that space-time on which the Poincaré generators are realised as differential operators has already been introduced. Even without this space-time, we could still choose to work with unfolded modules containing irreducible tensors.

In order to ensure that the only propagating degrees of freedom are those of the original three-form, we relate the field strengths of the three-form and six-form fields using the on-shell duality relation that follows from the $E_{11}$ non-linear realisation[10] [92]:

$$F_{a_1\cdots a_7} = \varepsilon_{a_1\cdots a_7}{}^{b_1\cdots b_4} F_{b_1\cdots b_4}.$$ (68)

Their higher gradients are therefore also related with an infinite set of relations

$$F_{a_1\cdots a_7, c_1, \ldots, c_n} = \varepsilon_{a_1\cdots a_7}{}^{b_1\cdots b_4} F_{b_1\cdots b_4, c_1, \ldots, c_n}.$$ (69)

It was explained in [23] that, as expected, the equations of motion and Bianchi identities for the three-form and six-form are exchanged through these relations. Equivalently, tracelessness and over-antisymmetrisation constraints on all higher gradients are exchanged.

# 4 Unfolding $E_{11}$ at level four

## 4.1 Unfolding the field $A_{9,3}$

Much of the unfolded description of $A_{9,3}$ was given in [23] and here we revisit and build upon it by working out the gauge symmetries of all the irreducible fields. We introduce the objects

$$e_{[9]}{}^{a_1a_2a_3}, \quad \omega_{[3]}{}^{a_1\cdots a_{10}}, \quad C^{a_1\cdots a_{10}, b_1\cdots b_4}, \quad \ldots,$$ (70)

where the primary zero-form $C_{a_1\ldots a_{10}, b_1\ldots b_4}$ is the first zero-form in the infinite tower

$$\mathcal{T}(A_{9,3}) = \left\{ C^{(n)}_{10,4,1^n} \mid n \in \mathbb{N} \right\} = \left\{ C^{(0)}_{10,4}, C^{(1)}_{10,4,1}, C^{(2)}_{10,4,1,1}, \ldots \right\}.$$ (71)

In contrast to the unfolding of the fields at levels three and below, $C_{10,4}$ and its descendants do not need to be completely traceless on-shell – see [23]. They will turn out to obey certain higher trace constraints that ensure their equivalence with irreducible Lorentz tensors in (63) and so the higher dual field $A_{9,3}$ will be dynamically equivalent to the three-form.

Unfolding off-shell for the moment, the first two equations are given by

$$\mathrm{d}e_{[9]}{}^{a[3]} + h_{b[7]}\,\omega_{[3]}{}^{a[3]b[7]} = 0,$$ (72a)

$$\mathrm{d}\omega_{[3]}{}^{a[10]} + h_{b[4]}\,C^{a[10],b[4]} = 0,$$ (72b)

---

[10]This duality relation is not only linearised but it is also a truncation of the full duality relation in the sense that we drop any terms containing derivatives with respect to space-time coordinates at higher levels. We only retain derivatives with respect to the original eleven-dimensional coordinates at level zero.

with gauge transformations

$$\delta e_{[9]}{}^{a[3]} = \mathrm{d}\lambda_{[8]}{}^{a[3]} + h_{b[7]}\,\alpha_{[2]}{}^{a[3]b[7]}, \qquad \delta\omega_{[3]}{}^{a[10]} = \mathrm{d}\alpha_{[2]}{}^{a[10]}. \tag{73}$$

In components (after rescaling and renaming the $p$-form components), the equations are

$$\partial_{[a_1} e_{a_2\cdots a_{10}]|b_1 b_2 b_3} + \omega_{[a_1 a_2 a_3|a_4\cdots a_{10}]b_1 b_2 b_3} = 0, \tag{74a}$$

$$\partial_{[a_1}\omega_{a_2 a_3 a_4]|b_1\cdots b_{10}} + C_{b_1\cdots b_{10},a_1\cdots a_4} = 0, \tag{74b}$$

and the gauge transformations take the form

$$\delta e_{a_1\cdots a_9|b_1 b_2 b_3} = \partial_{[a_1}\lambda_{a_2\cdots a_9]|b_1 b_2 b_3} - \alpha_{[a_1 a_2|a_3\cdots a_9]b_1 b_2 b_3}, \tag{75a}$$

$$\delta\omega_{a_1 a_2 a_3|b_1\cdots b_{10}} = \partial_{[a_1}\alpha_{a_2 a_3]|b_1\cdots b_{10}}. \tag{75b}$$

We can decompose the parameter $\alpha_{[2]}{}^{10}$ into irreducible components as

$$\alpha_{a_1 a_2|b_1\cdots b_{10}} = 12\,\alpha^{(1)}_{b_1\cdots b_{10},a_1 a_2} - 3\,\alpha^{(2)}_{b_1\cdots b_{10}[a_1,a_2]}, \tag{76}$$

and over-antisymmetrisation constraints for each component leads to

$$\alpha^{(1)}_{b_1 b_2 b_3[a_1\cdots a_7,a_8 a_9]} = -\frac{1}{12}\alpha^{(1)}_{a_1\cdots a_9[b_1,b_2 b_3]}, \qquad \alpha^{(2)}_{b_1 b_2 b_3[a_1\cdots a_8,a_9]} = \frac{1}{3}\alpha^{(2)}_{a_1\cdots a_9[b_1 b_2,b_3]}. \tag{77}$$

Equation (75a) can now be written in the convenient form

$$\delta e_{a_1\cdots a_9|b_1 b_2 b_3} = \partial_{[a_1}\lambda_{a_2\cdots a_9]|b_1 b_2 b_3} - \alpha^{(1)}_{a_1\cdots a_9[b_1,b_2 b_3]} - \alpha^{(2)}_{a_1\cdots a_9[b_1 b_2,b_3]}. \tag{78}$$

Decomposing the fields and differential parameters into irreducible components, we find

$$e_{a_1\cdots a_9|b_1 b_2 b_3} = A_{a_1\cdots a_9,b_1 b_2 b_3} + \widehat{A}_{a_1\cdots a_9[b_1,b_2 b_3]} + \widehat{A}_{a_1\cdots a_9[b_1 b_2,b_3]}, \tag{79a}$$

$$\lambda_{a_1\cdots a_8|b_1 b_2 b_3} = \lambda^{(1)}_{a_1\cdots a_8,b_1 b_2 b_3} + \lambda^{(2)}_{a_1\cdots a_8[b_1,b_2 b_3]} + \lambda^{(3)}_{a_1\cdots a_8[b_1 b_2,b_3]} + \lambda^{(4)}_{a_1\cdots a_8 b_1 b_2 b_3}. \tag{79b}$$

It is direct to show that

$$\widehat{A}_{a_1\cdots a_{11},b} = \frac{11}{3} e_{[a_1\cdots a_9|a_{10}a_{11}]b}, \tag{80a}$$

$$\widehat{A}_{a_1\cdots a_{10},b_1 b_2} = \frac{15}{4} e_{[a_1\cdots a_9|a_{10}]b_1 b_2} - \frac{9}{4}\widehat{A}_{a_1\cdots a_{10}[b_1,b_2]}. \tag{80b}$$

As a result, we obtain

$$\delta A_{a_1\cdots a_9,b_1 b_2 b_3} = \partial_{[a_1}\lambda^{(1)}_{a_2\cdots a_9],b_1 b_2 b_3} + \frac{7}{72}\left(\partial_{[b_1|}\lambda^{(2)}_{a_1\cdots a_9,|b_2 b_3]} + \frac{9}{7}\partial_{[a_1}\lambda^{(2)}_{a_2\cdots a_9][b_1,b_2 b_3]}\right), \tag{81a}$$

$$\delta_\lambda\widehat{A}_{a_1\cdots a_{10},b_1 b_2} = \frac{35}{36}\partial_{[a_1}\lambda^{(2)}_{a_2\cdots a_{10}],b_1 b_2} - \frac{1}{5}\left(\partial_{[b_1|}\lambda^{(3)}_{a_1\cdots a_{10},|b_2]} - \frac{10}{9}\partial_{[a_1}\lambda^{(3)}_{a_2\cdots a_{10}][b_1,b_2]}\right), \tag{81b}$$

$$\delta_\lambda\widehat{A}_{a_1\cdots a_{11},b} = \frac{44}{45}\partial_{[a_1}\lambda^{(3)}_{a_2\cdots a_{11}],b} + \frac{11}{3}\partial_{[a_1}\lambda^{(4)}_{a_2\cdots a_{11}]b}. \tag{81c}$$

The first gauge transformation (81a) matches that of $A_{9,3}$ in the $E_{11}$ non-linear realisation [12]. In addition, the extra fields can be eliminated using the algebraic symmetries

$$\delta_\alpha\widehat{A}_{a_1\cdots a_{10},b_1 b_2} = -\alpha^{(1)}_{a_1\cdots a_{10},b_1 b_2}, \qquad \delta_\alpha\widehat{A}_{a_1\cdots a_{11},b} = -\alpha^{(2)}_{a_1\cdots a_{11},b}. \tag{82}$$

After having done so, there would still exist some residual gauge symmetry whereby the gauge parameters are related to each other as

$$\alpha^{(1)}_{a_1\cdots a_{10},b_1 b_2} = \frac{35}{36}\partial_{[a_1}\lambda^{(2)}_{a_2\cdots a_{10}],b_1 b_2} - \frac{1}{5}\left(\partial_{[b_1|}\lambda^{(3)}_{a_1\cdots a_{10},|b_2]} - \frac{10}{9}\partial_{[a_1}\lambda^{(3)}_{a_2\cdots a_{10}][b_1,b_2]}\right), \quad (83a)$$

$$\alpha^{(2)}_{a_1\cdots a_{11},b} = \frac{44}{45}\partial_{[a_1}\lambda^{(3)}_{a_2\cdots a_{11}],b} + \frac{11}{3}\partial_{[a_1}\lambda^{(4)}_{a_2\cdots a_{11}]b}. \quad (83b)$$

One can use the decomposition given in (79a) to solve for $\omega_{[3]}{}^{10}$ in terms of $e_{[9]}{}^3$ as follows. It is useful to define and work with $\widetilde{\omega}_{[3]}{}^1$ which is related to $\omega_{[3]}{}^{10}$ by

$$\widetilde{\omega}_{a_1 a_2 a_3|b} = \frac{1}{10!}\varepsilon_b{}^{c_1\cdots c_{10}}\omega_{a_1 a_2 a_3|c_1\cdots c_{10}}, \qquad \widetilde{\omega}_{a_1 a_2 a_3|b_1\cdots b_{10}} = -\varepsilon_{b_1\cdots b_{10}}{}^c\,\widetilde{\omega}_{a_1 a_2 a_3|c}. \quad (84)$$

Now rewrite equation (74a) in the form

$$0 = \varepsilon^{ca_1\cdots a_{10}}\left(\partial_{a_1}e_{a_2\cdots a_{10}|b_1 b_2 b_3} - \varepsilon_{a_4\cdots a_{10}b_1 b_2 b_3 d}\,\widetilde{\omega}_{a_1 a_2 a_3|}{}^d\right). \quad (85)$$

This leads to

$$\widetilde{\omega}_{b_1 b_2 c|}{}^c = -\frac{1}{3!\,8!}\varepsilon^{a_1\cdots a_{11}}\partial_{a_1}e_{a_2\cdots a_{10}|a_{11}b_1 b_2}. \quad (86)$$

Substituting back, we see that

$$\widetilde{\omega}_{b_1 b_2 b_3|}{}^c = \frac{1}{3!\,7!}\left(\varepsilon^{ca_1\cdots a_{10}}\partial_{[a_1}e_{a_2\cdots a_{10}]|b_1 b_2 b_3} - \frac{3}{8}\varepsilon^{a_1\cdots a_{11}}\delta^c_{[b_1}\partial_{[a_1}e_{a_2\cdots a_{10}|a_{11}]b_2 b_3]}\right), \quad (87)$$

and using equation (84) we obtain

$$\omega_{a_1 a_2 a_3|b_1\cdots b_{10}} = 75\,\partial_{[b_1}e_{b_2\cdots b_{10}]|a_1 a_2 a_3} - 45\,\partial_{[a_1}e_{[b_1\cdots b_9|b_{10}]a_2 a_3]} + 405\,\partial_{[b_1}e_{b_2\cdots b_9[a_1|a_2 a_3]b_{10}]}. \quad (88)$$

Then, decomposing $e_{[9]}{}^3$ with equation (79a) we conclude that

$$\omega_{a_1 a_2 a_3|b_1\cdots b_{10}} = 120\,\partial_{[b_1}A_{b_2\cdots b_{10}],a_1 a_2 a_3} - 12\,\partial_{[a_1|}\widehat{A}_{b_1\cdots b_{10},|a_2 a_3]} + 3\,\partial_{[a_1|}\widehat{A}_{b_1\cdots b_{10}|a_2,a_3]}. \quad (89)$$

Now we will revisit the primary zero-form. Remaining off-shell, equations (89) and (74b) imply that $C_{10,4}$ can be expressed as the curvature tensor

$$C_{a_1\cdots a_{10},b_1\cdots b_4} = \partial_{[b_1}\partial_{[a_1}A_{a_2\cdots a_{10}],b_2 b_3 b_4]}, \quad (90)$$

up to a factor. Unfolding on-shell will force $C_{10,4}$ to satisfy a higher trace constraint to ensure that the propagating degrees of freedom are those of the three-form field. This constraint can be found by relating the zero-forms $C_{10,4,1,\dots,1}$ in (71) to the zero-forms $F_{4,1,\dots,1}$ in the unfolded module (63). Concretely, for the primary zero-form, we set

$$C_{a_1\cdots a_{10},b_1\cdots b_4} = \varepsilon_{a_1\cdots a_{10}}{}^c F_{b_1\cdots b_4,c}, \quad (91)$$

so that $C_{10,4}$ is equivalent to $F_{4,1}$ in $\mathcal{T}(A_3)$. It was shown in [23] that the antisymmetrisation and trace constraints of $C_{10,4}$ and $F_{4,1}$ are exchanged under (91) as follows:

$$\left.\begin{array}{l}\mathrm{Tr}^4(C_{10,4}) = 0, \\ C_{10,4} \text{ is GL(11) irreducible}\end{array}\right\} \iff \left\{\begin{array}{l}F_{4,1} \text{ is GL(11) irreducible,} \\ \mathrm{Tr}(F_{4,1}) = 0.\end{array}\right. \quad (92)$$

In other words, the Lorentz irreducibility properties of the zero-form $F_{4,1}$ which are equivalent to the Bianchi identity $\partial_{[a_1}F_{a_2\dots a_5]} = 0$ and the Maxwell equation $\partial^a F_{abcd} = 0$, are also equivalent to $C_{10,4}$ being GL(11) irreducible and subject to the higher trace constraint

$$\mathrm{Tr}^4(C_{10,4}) = C^{a_1\cdots a_6 b_1\cdots b_4,}{}_{b_1\cdots b_4} = \partial^{[b_1}\partial_{[a_1}A_{a_2\cdots a_6 b_1\cdots b_4],}{}^{b_2 b_3 b_4]} = 0. \quad (93)$$

This is the linearised equation of motion for the $A_{9,3}$ field in the $E_{11}$ non-linear realisation [83]. Starting from the non-linear realisation one could take the field equation (93) and then work backwards to obtain the relation between $C_{10,4}$ and $F_{4,1}$ in (91).

## 4.2 Unfolding the field $B_{10,1,1}$

In the unfolding of $B_{10,1,1}$ at level four, we introduce the variables

$$e_{[10]}{}^{a,b}, \quad \omega_{[1]}{}^{a_1\cdots a_{11},b}, \quad X_{[1]}{}^{a_1\cdots a_{11},b_1 b_2}, \quad C^{a_1\cdots a_{11},b_1 b_2,c_1 c_2}, \quad \ldots, \tag{94}$$

where $e_{[10]}{}^{a,b} = e_{[10]}{}^{(a,b)}$ and the primary zero-form $C_{11,2,2}$ is the first in the infinite tower

$$\mathcal{T}(B_{10,1,1}) = \left\{ C^{(n)}_{11,2,2,1^n} \,\middle|\, n \in \mathbb{N} \right\} = \left\{ C^{(0)}_{11,2,2}, C^{(1)}_{11,2,2,1}, C^{(2)}_{11,2,2,1,1}, \ldots \right\}. \tag{95}$$

The first three unfolded equations are

$$\mathrm{d}e_{[10]}{}^{a,b} + h_{c[10]}\,\omega_{[1]}{}^{c[10](a,b)} = 0\,, \tag{96a}$$

$$\mathrm{d}\omega_{[1]}{}^{a[11],b} + h_c\,X_{[1]}{}^{a[11],bc} = 0\,, \tag{96b}$$

$$\mathrm{d}X_{[1]}{}^{a[11],b[2]} + h_{c[2]}\,C^{a[11],b[2],c[2]} = 0\,, \tag{96c}$$

and they are invariant under the transformations

$$\delta e_{[10]}{}^{a,b} = \mathrm{d}\lambda_{[9]}{}^{a,b} - h_{c[10]}\,\alpha^{c[10](a,b)}\,, \tag{97a}$$

$$\delta\omega_{[1]}{}^{a[11],b} = \mathrm{d}\alpha^{a[11],b} + h_c\,\beta^{a[11],bc}\,, \tag{97b}$$

$$\delta X_{[1]}{}^{a[11],b[2]} = \mathrm{d}\beta^{a[11],b[2]}\,. \tag{97c}$$

We will briefly explain the gauge invariance of (96a). The left-hand side clearly vanishes under the $\lambda$ part of the gauge transformation, while the $\alpha$ part is given by

$$\delta_\alpha\left(\mathrm{d}e_{[10]}{}^{a,b} + h_{c[10]}\,\omega_{[1]}{}^{c[10](a,b)}\right) = \mathrm{d}\left(-h_{c[10]}\,\alpha^{c[10](a,b)}\right) + h_{c[10]}\,\mathrm{d}\alpha^{c[10](a,b)} = 0\,. \tag{98}$$

The $\beta$ part also vanishes:

$$\delta_\beta\left(\mathrm{d}e_{[10]}{}^{a,b} + h_{c[10]}\,\omega_{[1]}{}^{c[10](a,b)}\right) = h_{c[11]}\,\beta^{c_1\cdots c_{10}(a,b)c_{11}} = 0\,. \tag{99}$$

To see this, recall that $\beta^{11,2}$ is irreducible, so it satisfies $\beta_{[a_1\cdots a_{11},b_1]b_2} = 0$. We can use this to move the symmetrised indices into the second antisymmetric block, and therefore the $\beta$ part is zero. Similarly, notice that $\mathbb{Y}[11,2]$ is not an irreducible component of the tensor product $\mathbb{Y}[11] \otimes \mathbb{Y}[1,1]$ and hence $\beta^{[c_1\cdots c_{10}(a,b)c_{11}]}$ vanishes.

In components, after the usual rescaling, (96a) is given by

$$\partial_{[a_1}e_{a_2\cdots a_{11}]|b,c} + \omega_{[a_1|a_2\cdots a_{11}](b,c)} = 0\,, \tag{100}$$

and it is invariant under

$$\delta e_{a_1\cdots a_{10}|b,c} = \partial_{[a_1}\lambda_{a_2\cdots a_{10}]|b,c} - \alpha_{a_1\cdots a_{10}(b,c)}\,, \tag{101}$$

$$\delta_\alpha \omega_{a|b_1\cdots b_{11},c} = \partial_a \alpha_{b_1\cdots b_{11},c}\,. \tag{102}$$

We decompose the fields and parameters into irreducible parts

$$e_{a_1\cdots a_{10}|b,c} = B_{a_1\cdots a_{10},b,c} + \widehat{B}_{a_1\cdots a_{10}(b,c)}\,, \qquad \lambda_{a_1\cdots a_9|b,c} = \lambda^{(5)}_{a_1\cdots a_9,b,c} + \lambda^{(6)}_{a_1\cdots a_9(b,c)}\,, \tag{103}$$

with Young tableaux

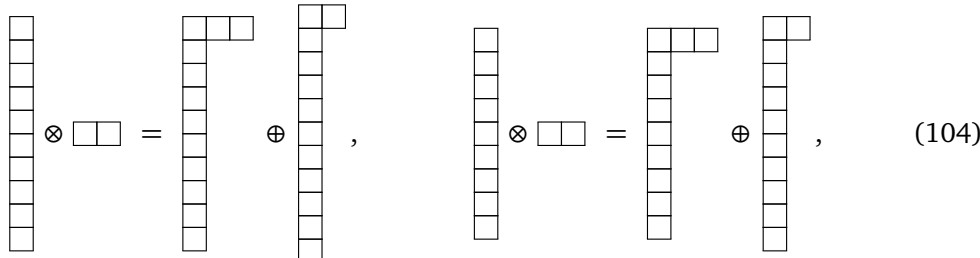

$$\tag{104}$$

where the over-antisymmetrisation constraints are given by

$$B_{[a_1\cdots a_{10},b],c} = \widehat{B}_{[a_1\cdots a_{11},b]} = \lambda^{(5)}_{[a_1\cdots a_9,b],c} = \lambda^{(5)}_{a_1\cdots a_9,[b,c]} = \lambda^{(6)}_{[a_1\cdots a_{10},b]} = 0\,. \tag{105}$$

The transformations of the irreducible fields are given by

$$\delta B_{a_1\cdots a_{10},b,c} = \partial_{[a_1}\lambda^{(5)}_{a_2\cdots a_{10}],b,c} - \frac{11}{60}\left(\partial_{(b|}\lambda^{(6)}_{a_1\cdots a_{10},|c)} - \frac{10}{11}\partial_{[a_1}\lambda^{(6)}_{a_2\cdots a_{10}](b,c)}\right), \tag{106a}$$

$$\delta\widehat{B}_{a_1\cdots a_{11},b} = \frac{121}{60}\partial_{[a_1}\lambda^{(6)}_{a_2\cdots a_{11}],b} - \alpha_{a_1\cdots a_{11},b}\,. \tag{106b}$$

Unfolding $B_{10,1,1}$ has introduced an extra field $\widehat{B}_{11,1}$ that we can eliminate using the $\alpha_{11,1}$ part of (106b), leaving a residual gauge symmetry where the parameters are related by

$$\alpha_{a_1\cdots a_{11},b} = \frac{121}{60}\partial_{[a_1}\lambda^{(6)}_{a_1\cdots a_{11}],b}\,. \tag{107}$$

The field $B_{10,1,1}$ occurs at level four in the $E_{11}$ non-linear realisation and (106a) matches its gauge transformation [12]. Its unfolded formulation features the extra field $\widehat{B}_{11,1}$, and there is a field with precisely this symmetry type in the non-linear realisation at level four, namely the field $C_{11,1}$ in equation (6).

We can solve for $\omega_{[1]}{}^{11,1}$ in equation (100). It will be useful to rewrite it in the form

$$\partial_{[a_1}e_{a_2\cdots a_{11}]|b,c} + \frac{1}{11}\omega_{(b||a_1\cdots a_{11},|c)} = 0\,. \tag{108}$$

Note that only the part of the connection that is symmetric in $b$ and $c$ appears, and one finds that it is given by

$$\omega_{(a||b_1\cdots b_{11},|c)} = -11\,\partial_{[b_1}e_{b_2\cdots b_{11}]|a,c} = -11\,\partial_{[b_1}B_{b_2\cdots b_{11}],a,c} - 2\,\partial_{(a|}\widehat{B}_{b_1\cdots b_{11},|c)}\,. \tag{109}$$

Looking at the gauge transformations (97b) and (97c), we observe that $\omega_{[a||b_1\cdots b_{11},|c]}$ is pure gauge, so we can use

$$\delta_\beta\,\omega_{[a||b_1\cdots b_{11},|c]} = \beta_{b_1\cdots b_{11},ac}\,, \tag{110}$$

to set this component to zero. Solving (96b) for $X_{[1]}{}^{11,2}$ in terms of $\omega_{[1]}{}^{11,1}$ leads to

$$X_{a|b_1\cdots b_{11},c_1c_2} = -22\,\partial_{[c_1}\partial_{[b_1}B_{b_2\cdots b_{11}],c_2],a} - 2\,\partial_a\partial_{[c_1|}\widehat{B}_{b_1\cdots b_{11},|c_2]}\,. \tag{111}$$

Then, solving (96c) implies that $C_{11,2,2}$ can be expressed as the curvature tensor

$$C_{a_1\cdots a_{11},b_1b_2,c_1c_2} = \partial_{[c_1}\partial_{[b_1}\partial_{[a_1}B_{a_2\cdots a_{11}],b_2],c_2]}\,, \tag{112}$$

up to a factor.

If we unfold on-shell, then the primary zero-form $C_{11,2,2}$ will obey a trace constraint. As we discussed in Section 2, the $B_{10,1,1}$ field at level four in $E_{11}$ is related to the Romans field and proagates no additional degrees of freedom. The equation of motion that follows from $E_{11}$ symmetry [12] can be expressed as the complete tracelessness of the curvature tensor:

$$\mathrm{Tr}(C_{11,2,2}) = \mathrm{Tr}(\partial_{[c_1}\partial_{[b_1}\partial_{[a_1}B_{a_2\cdots a_{11}],b_2],c_2]}) = 0\,. \tag{113}$$

This is equivalent to $C_{11,2,2}$ vanishing since it is now an irreducible Lorentz representation.

Although the primary zero-form $C_{10,4}$ of $A_{9,3}$ is equivalent to $F_{4,1} \in \mathcal{T}(A_3)$, we do not have such an equivalence for $C_{11,2,2}$ because $B_{10,1,1}$ is not dual to any field at lower levels. Therefore, $C_{11,2,2}$ does not satisfy an unusual higher trace constraint analogous to (93).

## 4.3 The higher dual three-form duality relation

The non-linear realisation of $E_{11}$ contains three infinite families of higher dual fields and the degrees of freedom of the theory are those of the graviton and the three-form. An infinite set of duality relations that is invariant under the symmetries of the non-linear realisation ensures that the degrees of freedom are preserved. The first higher dual field that we encounter is $A_{9,3}$ at level four. Its unfolded description was given in [23] and we have built upon it by working out the gauge transformations of the extra fields $\widehat{A}_{10,2}$ and $\widehat{A}_{11,1}$ in Section 4.1. These extra fields appear explicitly in the connection $\omega_{[3]}{}^{10}$ after solving the first unfolded equation (72a). However, after solving the second unfolded equation (72b) for the primary zero-form $C_{10,4}$, we find that the two extra fields no longer appear and that $C_{10,4}$ can be written as the curvature tensor of the $A_{9,3}$ field in equation (90).

The first-order duality relation in the non-linear realisation between $A_{9,3}$ at level four and $A_3$ at level one was found [12, 23] to take the form[11]

$$\omega_{a_1 a_2 a_3 | b_1 \cdots b_{10}} \propto \varepsilon_{b_1 \cdots b_{10}}{}^c F_{c\, a_1 a_2 a_3}. \tag{114}$$

Note that we are working at the linearised level, so the coefficient in (114) can be absorbed in a redefinition of the variables. At the full non-linear level, the coefficients would be fixed by $E_{11}$ symmetry since, as we explained in the introduction, $E_{11}$ symmetry determines the tensor structure and the precise combination of terms in all the equations of the theory.

In the $E_{11}$ non-linear realisation, this duality relation between the three-form and the higher dual field $A_{9,3}$ held up to some pure gauge terms, and in our proposed duality relation (114) this gauge freedom has been soaked up by the two extra fields $\widehat{A}_{10,2}$ and $\widehat{A}_{11,1}$. Using equation (89) and taking a curl on the $a[3]$ indices, we obtain the gauge-invariant relation

$$\partial_{[b_1} \partial_{[a_1} A_{a_2 \cdots a_{10}], b_2 b_3 b_4]} \propto \varepsilon_{a_1 \cdots a_{10}}{}^c \partial_c \partial_{[b_1} A_{b_2 b_3 b_4]}. \tag{115}$$

Then taking the fourth trace of both sides leads to the linearised equation of motion for the $A_{9,3}$ field in the non-linear realisation:

$$\partial^{[b_1} \partial_{[a_1} A_{a_2 \cdots a_6 b_1 \cdots b_4],}{}^{b_2 b_3 b_4]} = 0. \tag{116}$$

Similarly, contracting both sides of (115) with $\varepsilon^{a_1 \cdots a_{10} b_1}$ leads directly to the Maxwell equation $\partial^a \partial_{[a} A_{b_1 b_2 b_3]} = 0$. Thus the equations of motion for the three-form and the higher dual field follow directly from the duality relation (114) which is now gauge-invariant as a result of our choice to include extra fields.

Equation (115) can also be written as a relation between (curvature) zero-forms (91), ensuring that $C_{10,4} \in \mathcal{T}(A_{9,3})$ and $F_{4,1} \in \mathcal{T}(A_3)$ are equivalent Lorentz tensors. Following [23], solving the unfolded equations allows us to express the zero-forms in terms of their respective fields, and then the zero-form relation (91) takes the form of (115).

Working backwards, we can integrate (115) to obtain a first-order relation

$$\partial_{[a_1} A_{a_2 \cdots a_{10}], b_1 b_2 b_3} + \partial_{[b_1} \Xi_{b_2 b_3]| a_1 \cdots a_{10}} \propto \varepsilon_{a_1 \cdots a_{10}}{}^c \partial_c A_{b_1 b_2 b_3}, \tag{117}$$

up to an arbitrary $\Xi_{2|10}$ term. It is useful to impose the shift

$$\Xi_{a_1 a_2 | b_1 \cdots b_{10}} \longmapsto \Xi_{a_1 a_2 | b_1 \cdots b_{10}} + 3\, \varepsilon_{b_1 \cdots b_{10}}{}^c A_{a_1 a_2 c}, \tag{118}$$

so that we can rewrite (117) as

$$\partial_{[a_1} A_{a_2 \cdots a_{10}], b_1 b_2 b_3} + \partial_{[b_1} \Xi_{b_2 b_3]| a_1 \cdots a_{10}} \propto \varepsilon_{a_1 \cdots a_{10}}{}^c F_{c\, b_1 b_2 b_3}. \tag{119}$$

---

[11]Whenever we unfold an irreducible field with more than one block of antisymmetric indices, the first-order variable is labelled $\omega$. If it has only one block, then the first-order variable is just the primary zero-form and it is labelled $F$. Thus all the first-order duality relations in this paper take the form "$\omega \propto *\omega$" or "$\omega \propto *F$".

Labelling the irreducible components of $\Xi_{2|10}$ as $\Xi^{(1)}_{10,2}$ and $\Xi^{(2)}_{11,1}$, this relation becomes

$$\partial_{[a_1} A_{a_2 \cdots a_{10}],b_1 b_2 b_3} + \partial_{[b_1|} \Xi^{(1)}_{a_1 \cdots a_{10},|b_2 b_3]} + \partial_{[b_1|} \Xi^{(2)}_{a_1 \cdots a_{10}|b_2,b_3]} \propto \varepsilon_{a_1 \cdots a_{10}}{}^c F_{c\, b_1 b_2 b_3}. \tag{120}$$

As worked out in [23], taking a curl on the $a[10]$ indices leads to

$$\partial_{[a_1} \partial_{[b_1} \Xi_{b_2 b_3]|a_2 \cdots a_{11}]} \propto \varepsilon_{a_1 \cdots a_{11}} \partial^c F_{c\, b_1 b_2 b_3}, \tag{121}$$

which vanishes on-shell due to the Maxwell equation. As a result, we find

$$\partial_{[b_1} \Xi_{b_2 b_3]|a_1 \cdots a_{10}} = \partial_{[a_1} \partial_{[b_1} \xi_{b_2 b_3]|a_2 \cdots a_{10}]}, \tag{122}$$

for some tensor $\xi_{2|9}$ whose components have the same tableaux as the differential parameters of (81b) and (81c). The duality relation can now be written as

$$\partial_{[a_1} A_{a_2 \cdots a_{10}],b_1 b_2 b_3} + \partial_{[a_1} \partial_{[b_1} \xi_{b_2 b_3]|a_2 \cdots a_{10}]} \propto \varepsilon_{a_1 \cdots a_{10}}{}^c F_{c\, b_1 b_2 b_3}. \tag{123}$$

Choosing for the moment not to express the $\Xi$ fields in terms of the smaller $\xi$ fields, the left-hand side of (120) will be proportional to $\omega_{[3]}{}^{10}$ in (89) if we set $\Xi^{(1)}_{10,2} = -\frac{1}{10}\widehat{A}_{10,2}$ and $\Xi^{(2)}_{11,1} = \frac{1}{40}\widehat{A}_{11,1}$. This justifies *a posteriori* our proposed duality relation (114) featuring extra fields. It transforms only with the parameters $\alpha^{(1)}_{10,2}$ and $\alpha^{(2)}_{11,1}$ as

$$\delta \omega_{a_1 a_2 a_3|b_1 \cdots b_{10}} = 12\, \partial_{[a_1|} \alpha^{(1)}_{b_1 \cdots b_{10},|a_2 a_3]} - 3\, \partial_{[a_1|} \alpha^{(2)}_{b_1 \cdots b_{10}|a_2,a_3]} = \partial_{[a_1} \alpha_{a_2 a_3]|b_1 \cdots b_{10}}, \tag{124}$$

where $\alpha_{[2]}{}^{10}$ is the reducible gauge parameter in (76). The right-hand side of (120) is gauge-invariant, and hence so is the left-hand side. We see that $\partial_{[a_1} \alpha_{a_2 a_3]|b_1 \cdots b_{10}} = 0$ is solved by

$$\alpha_{[2]}{}^{a[10]} = \mathrm{d}\alpha_{[1]}{}^{a[10]} \qquad \Longleftrightarrow \qquad \alpha_{b_1 b_2|a_1 \cdots a_{10}} = 2\, \partial_{[b_1} \alpha_{b_2]|a_1 \cdots a_{10}}, \tag{125}$$

where $\alpha_{[1]}{}^{10}$ is a gauge-for-gauge parameter. It was expected that we would need a constraint on our algebraic parameter. For example, the duality relation (46) between gravity and dual gravity holds if the two-form parameter $\alpha_2$ and the nine-form parameter $\alpha_9$ are Hodge dual to each other as in equation (47). However, it is interesting that (120) forces $\alpha_{[2]}{}^{10}$ to be pure gauge-for-gauge. Notice that $e_{[9]}{}^3$ no longer transforms with algebraic shift symmetries as in (78) and the extra fields are necessary to make sense of (120). There must be more freedom at the level of the fields when there are constraints on the parameters. The extra fields in the duality relation (123) emerge in a way that is compatible with this freedom.

To be precise, the duality relation (123) features two dual fields, $A_3$ and $A_{9,3}$, as well as some extra fields: $\xi_{9,2}$, $\xi_{10,1}$, and $\xi_{11}$. Demanding that our duality relation is gauge invariant, we found that our gauge parameter $\alpha_{2|10}$ is built from two smaller parameters: $\alpha_{10,1}$ and $\alpha_{11}$. Thus we are only able to eliminate two of the three extra fields, and our final gauge-invariant duality relation is given in terms of $A_3$, $A_{9,3}$, and the last extra field $\xi_{9,2}$:

$$\partial_{[a_1} A_{a_2 \cdots a_{10}],b_1 b_2 b_3} + \partial_{[b_1} \partial_{[a_1} \xi_{a_2 \cdots a_{10}],b_2 b_3]} \propto \varepsilon_{a_1 \cdots a_{10}}{}^c F_{c\, b_1 b_2 b_3}. \tag{126}$$

This is exactly the duality relation in equation (3.2.12) of [23] that was found using a different procedure. At higher levels, one should in principle be able to obtain the same kind of duality relations with the $\alpha$ parameters constrained and some of the extra $\xi$ fields left intact.

In Section 3.5 we gave the duality relation (68) between $F_4$ and $F_7$. Then in Section 4.1 we wrote down the relation (91) between $F_{4,1}$ into $C_{10,4}$. After expressing $F_{4,1}$ in terms of the three-form and $C_{10,4}$ in terms of the $A_{9,3}$ field, integrating (91) led to a duality relation (114) between these fields featuring a pair of extra fields $\widehat{A}_{10,2}$ and $\widehat{A}_{11,1}$ that we identify as those at level one in the $\ell_2$ representation of $E_{11}$. Once again we find that extra fields are necessary for our first-order duality relations to hold exactly and not as equivalence equations.

# 5 Duality relations at higher levels

In this section we propose an infinite number of linearised first-order duality relations for all higher dual fields in the $E_{11}$ non-linear realisation. For each higher dual field, we will follow the same procedure: (1) introducing a set of variables, i.e. connections and zero-forms; (2) writing down the first few unfolded equations and their gauge transformations; and (3) proposing gauge-covariant duality relations between the dual fields in terms of the first-order variables.

Taking derivatives of our duality relations will lead to relations between primary zero-forms (written as curvature tensors), and taking traces leads to the linearised equations of motion. The equations of motion are expressed as constraints on the curvature. For any pair of dual fields considered here, we see that the curvature tensors are related algebraically, and so the constraints on one curvature directly lead to constraints on the other curvature, i.e. dual equations of motion. Integrating back, we find the pure gauge terms up to which the $E_{11}$ duality relations are expected to hold if the extra fields had not been included in our proposed relations. At low levels where they have already been worked out, the duality relations and equations of motion that we propose here match those of the non-linear realisation at the linearised level.

## 5.1 Unfolding the field $A_{9,6}$ at level five

At level five the idea is essentially the same as at level four, except now there are four fields in the non-linear realisation: $A_{9,6}$, $B_{10,4,1}$, $C_{11,3,1}$, and $C_{11,4}$. Only the higher dual six-form field $A_{9,6}$ will be unfolded here. We introduce an infinite set of variables

$$e_{[9]}{}^{a_1\cdots a_6}, \quad \omega_{[6]}{}^{a_1\cdots a_{10}}, \quad C^{a_1\cdots a_{10},b_1\cdots b_7}, \quad \ldots, \tag{127}$$

where the primary zero-form $C_{10,7}$ is the first zero-form in the module

$$\mathcal{T}(A_{9,6}) = \left\{ C^{(n)}_{10,7,1^n} \,\middle|\, n \in \mathbb{N} \right\} = \left\{ C^{(0)}_{10,7}, C^{(1)}_{10,7,1}, C^{(2)}_{10,7,1,1}, \ldots \right\}. \tag{128}$$

The first two unfolded equations are

$$\mathrm{d}e_{[9]}{}^{a[6]} + h_{b[4]}\,\omega_{[6]}{}^{a[6]b[4]} = 0, \tag{129a}$$

$$\mathrm{d}\omega_{[6]}{}^{a[10]} + h_{b[7]}\,C^{a[10],b[7]} = 0, \tag{129b}$$

with gauge transformations

$$\delta e_{[9]}{}^{a[6]} = \mathrm{d}\lambda_{[8]}{}^{a[6]} - h_{b[4]}\,\alpha_{[5]}{}^{a[6]b[4]}, \qquad \delta\omega_{[6]}{}^{a[10]} = \mathrm{d}\alpha_{[5]}{}^{a[10]}. \tag{130}$$

In components, the unfolded equations are

$$\partial_{[a_1}e_{a_2\cdots a_{10}]|b_1\cdots b_6} + \omega_{[a_1\cdots a_6|a_7\cdots a_{10}]b_1\cdots b_6} = 0, \tag{131a}$$

$$\partial_{[a_1}\omega_{a_2\cdots a_7]|b_1\cdots b_{10}} + C_{b_1\cdots b_{10},a_1\cdots a_7} = 0, \tag{131b}$$

and the gauge transformations are

$$\delta e_{a_1\cdots a_9|b_1\cdots b_6} = \partial_{[a_1}\lambda_{a_2\cdots a_9]|b_1\cdots b_6} - \alpha_{[a_1\cdots a_5|a_6\cdots a_9]b_1\cdots b_6}, \tag{132a}$$

$$\delta\omega_{a_1\cdots a_6|b_1\cdots b_{10}} = \partial_{[a_1}\alpha_{a_2\cdots a_6]|b_1\cdots b_{10}}. \tag{132b}$$

We can decompose $e_{[9]}{}^6$, $\lambda_{[8]}{}^6$ and $\alpha_{[5]}{}^{10}$ in terms of irreducible components as

$$e_{a_1\cdots a_9|b_1\cdots b_6} = A_{a_1\cdots a_9,b_1\cdots b_6} + \widehat{A}_{a_1\cdots a_9[b_1,b_2\cdots b_6]} + \widehat{A}_{a_1\cdots a_9[b_1b_2,b_3\cdots b_6]}, \tag{133}$$

$$\lambda_{a_1\cdots a_8|b_1\cdots b_6} = \lambda^{(1)}_{a_1\cdots a_8,b_1\cdots b_6} + \lambda^{(2)}_{a_1\cdots a_8[b_1,b_2\cdots b_6]} + \lambda^{(3)}_{a_1\cdots a_8[b_1b_2,b_3\cdots b_6]} + \lambda^{(4)}_{a_1\cdots a_8[b_1b_2b_3,b_4b_5b_6]}, \tag{134}$$

$$\alpha_{a_1\cdots a_{10}|b_1\cdots b_5} = \alpha^{(1)}_{a_1\cdots a_{10},b_1\cdots b_5} + \alpha^{(2)}_{a_1\cdots a_{10}[b_1,b_2\cdots b_5]}. \tag{135}$$

The higher dual field $A_{9,6}$ is contained inside the $e_{[9]}{}^6$ variable alongside two extra fields: $\widehat{A}_{10,5}$ and $\widehat{A}_{11,4}$. The parameters $\alpha^{(1)}_{10,5}$ and $\alpha^{(2)}_{11,4}$ can be used to shift away the two extra fields, and some residual gauge symmetry would remain wherein the $\alpha$ gauge parameters would be related to derivatives of the $\lambda$ parameters.

We propose a first-order on-shell duality relation between the six-form $A_6$ at level two and the higher dual six-form $A_{9,6}$ at level five in terms of the field strength $F_7 \in \mathcal{T}(A_6)$ and the first-order connection $\omega_{[6]}{}^{10}$ in (129a). This relation takes the form

$$\omega_{a_1\cdots a_6|b_1\cdots b_{10}} \propto \varepsilon_{b_1\cdots b_{10}}{}^c F_{c\,a_1\cdots a_6}. \tag{136}$$

Again, as with (114), in a consistent extension of E theory featuring all the extra fields, $E_{11}$ symmetry would fix the precise combination of terms in the non-linear duality relations. Here we are working at the linearised level, so the factor in (136) can be absorbed by a redefinition of the fields. On the left-hand side, $A_{9,6}$ appears inside $\omega_{[6]}{}^{10}$ with extra fields $\widehat{A}_{10,5}$ and $\widehat{A}_{11,4}$. The duality relation (136) is gauge-invariant when $\alpha_{[5]}{}^{10}$ is subject to some constraint that is analogous to (125) which forces $\alpha_{[5]}{}^{10}$ to be pure gauge-for-gauge.

Taking a curl of (136) on the $a[6]$ indices leads to the gauge-invariant relation

$$\partial_{[b_1}\partial_{[a_1}A_{a_2\cdots a_{10}],b_2\cdots b_7]} \propto \varepsilon_{a_1\cdots a_{10}}{}^c \partial_c \partial_{[b_1}A_{b_2\cdots b_7]}. \tag{137}$$

Taking the seventh trace of both sides leads to the equation of motion for the $A_{9,6}$ field

$$\partial^{[b_1}\partial_{[a_1}A_{a_2a_3b_1\cdots b_7],}{}^{b_2\cdots b_7]} = 0. \tag{138}$$

Similarly, as with (115), contracting both sides with $\varepsilon^{a_1\cdots a_{10}b_1}$ directly leads to the expected Maxwell equation $\partial^a \partial_{[a}A_{b_1\cdots b_6]} = 0$, so the two equations of motion follow from (136). It is also possible to integrate (137) to obtain the duality relation (136) in a form analogous to (123) at level four, this time featuring a reducible tensor $\xi_{5|9}$.

Alternatively, the equation of motion for $A_{9,6}$ can be described as a higher trace constraint for the primary zero-form $C_{10,7} \in \mathcal{T}(A_{9,6})$. Solving (129a) and (129b) for $C_{10,7}$ in terms of the irreducible fields, we find that it can be expressed as the curvature tensor

$$C_{a_1\cdots a_{10},b_1\cdots b_7} = \partial_{[b_1}\partial_{[a_1}A_{a_2\cdots a_{10}],b_2\cdots b_7]}, \tag{139}$$

up to a factor. Similarly, we can solve the unfolded equations of the six-form field to express $F_{7,1} \in \mathcal{T}(A_6)$ as $F_{a[7],b} = 7\,\partial_b\,\partial_{[a_1}A_{a_2\dots a_7]} = 0$. Thus we see that (137) can be rewritten as the zero-form relation

$$C_{a_1\cdots a_{10},b_1\cdots b_7} \propto \varepsilon_{a_1\cdots a_{10}}{}^c F_{b_1\cdots b_7,c}. \tag{140}$$

This is analogous to the zero-form relation (91) between $C_{10,4} \in \mathcal{T}(A_{9,3})$ and $F_{4,1} \in \mathcal{T}(A_3)$, and it means that $C_{10,7}$ and $F_{7,1}$ are equivalent Lorentz tensors. As always, we really have an infinite number of equivalences between zero-forms $C_{10,7,1,\ldots,1} \in \mathcal{T}(A_{9,6})$ and $F_{7,1,1,\ldots,1} \in \mathcal{T}(A_6)$ but for our purposes it will suffice to consider only (140).

On-shell, the (Lorentz) irreducibility properties of $F_{7,1}$ are exchanged under (140) with the analogous constraints on $C_{10,7}$ as

$$\left.\begin{array}{l} \text{Tr}^7(C_{10,7}) = 0\,, \\ C_{10,7} \text{ is GL(11) irreducible} \end{array}\right\} \quad \Longleftrightarrow \quad \left\{\begin{array}{l} F_{7,1} \text{ is GL(11) irreducible,} \\ \text{Tr}(F_{7,1}) = 0\,. \end{array}\right. \tag{141}$$

This is essentially the same as (92). As a result, the equation of motion for $A_{9,6}$ is equivalent to the higher trace constraint

$$\text{Tr}^7(C_{10,7}) = C_{a_1 \cdots a_4 b_1 \cdots b_7,}{}^{b_1 \cdots b_7} = \partial^{[b_1} \partial_{[a_1} A_{a_2 a_3 a_4 b_1 \cdots b_7],}{}^{b_2 \cdots b_7]} = 0\,. \tag{142}$$

**Duality relation between $A_{9,3}$ and $A_{9,6}$.** Using the zero-form relations (69), (91), and (140), we find that the primary zero-forms $C_{10,4}$ and $C_{10,7}$ are related by

$$C_{a[10],b[7]} \propto \varepsilon_{b[7]}{}^{c[4]} C_{a[10],c[4]}\,. \tag{143}$$

Their on-shell properties are exchanged under (143) as

$$\left.\begin{array}{l} \text{Tr}^4(C^{10,4}) = 0\,, \\ C^{10,4} \text{ is GL(11) irreducible} \end{array}\right\} \quad \Longleftrightarrow \quad \left\{\begin{array}{l} C^{10,7} \text{ is GL(11) irreducible,} \\ \text{Tr}^7(C^{10,7}) = 0\,. \end{array}\right. \tag{144}$$

We can combine the three first-order duality relations equations (68), (114) and (136) into a single relation between $A_{9,3}$ and $A_{9,6}$ that takes the form

$$\varepsilon^{c_1 \cdots c_{10}}{}_{[a_1} \omega_{a_2 a_3 a_4]|c_1 \cdots c_{10}} \propto \omega^{b_1 \cdots b_6|}{}_{a_1 \cdots a_4 b_1 \cdots b_6}\,. \tag{145}$$

It is useful to write this relation in the form

$$\widetilde{\omega}_{[a_1 a_2 a_3|a_4]} \propto \varepsilon_{a_1 \cdots a_4}{}^{b_1 \cdots b_7} \widetilde{\omega}_{[b_1 \cdots b_6|b_7]}\,, \tag{146}$$

where $\widetilde{\omega}_{[3]}{}^1$ and $\widetilde{\omega}_{[6]}{}^1$ are defined in terms of $\omega_{[3]}{}^{10}$ and $\omega_{[6]}{}^{10}$ by

$$\widetilde{\omega}_{a_1 a_2 a_3|b} := \varepsilon_b{}^{c_1 \cdots c_{10}} \omega_{a_1 a_2 a_3|c_1 \cdots c_{10}}\,, \qquad \widetilde{\omega}_{a_1 \cdots a_6|b} := \varepsilon_b{}^{c_1 \cdots c_{10}} \omega_{a_1 \cdots a_6|c_1 \cdots c_{10}}\,. \tag{147}$$

Equations (68), (114), (136) and (146) populate the following array of duality relations:

$$\begin{array}{ccc} F_4 & \longleftrightarrow & \omega_{[3]}{}^{10} \\ \updownarrow & & \updownarrow \\ F_7 & \longleftrightarrow & \omega_{[6]}{}^{10} \end{array} \tag{148}$$

This will be extended infinitely in Section 5.4 where first-order on-shell duality relations for all higher dual fields in the three-form and six-form sectors will be worked out.

## 5.2 Unfolding the field $h_{9,8,1}$ at level six

There are nine fields in the $E_{11}$ non-linear realisation at level six: $h_{9,8,1}$, $B_{10,6,2}$, $B_{10,7,1}$, $B_{10,8}$, $C_{11,4,3}$, $C_{11,5,1,1}$, two copies of $C_{11,6,1}$, and $C_{11,7}$. Here we will obtain the unfolded formulation of the higher dual gravity field $h_{9,8,1}$. In order to do so, we introduce a set of variables

$$e_{[9]}{}^{a_1 \cdots a_8, b}\,, \quad \omega_{[8]}{}^{a_1 \cdots a_{10}, b}\,, \quad X_{[1]}{}^{a_1 \cdots a_{10}, b_1 \cdots b_9}\,, \quad C^{a_1 \cdots a_{10}, b_1 \cdots b_9, c_1 c_2}\,, \quad \dots\,, \tag{149}$$

where the primary zero-form $C_{10,9,2}$ is the first zero-form in the module

$$\mathcal{T}(h_{9,8,1}) = \left\{ C_{10,9,2,1^n}^{(n)} \mid n \in \mathbb{N} \right\} = \left\{ C_{10,9,2}^{(0)}, C_{10,9,2,1}^{(1)}, C_{10,9,2,1,1}^{(2)}, \dots \right\}. \tag{150}$$

The first three unfolded equations are

$$\mathrm{d}e_{[9]}{}^{a[8],b} + h_{c[2]}\,\omega_{[8]}{}^{c[2]\langle a[8],b\rangle} = 0\,, \tag{151a}$$

$$\mathrm{d}\omega_{[8]}{}^{a[10],b} + h_{c[8]}X_{[1]}{}^{a[10],c[8]b} = 0\,, \tag{151b}$$

$$\mathrm{d}X_{[1]}{}^{a[10],b[9]} + h_{c[2]}C^{a[10],b[9],c[2]} = 0\,. \tag{151c}$$

In (151a) the angled brackets denote the projection of the final nine indices of $\omega_{[8]}{}^{10,1}$ onto the GL(11) irreducible $\mathbb{Y}[8,1]$ tableau. It may be clearer to rewrite (151a) as

$$\mathrm{d}e_{[9]}{}^{a_1\cdots a_8,b} + h_{c_1 c_2}\left(\omega_{[8]}{}^{c_1 c_2 a_1\cdots a_8,b} - \omega_{[8]}{}^{c_1 c_2 [a_1\cdots a_8,b]}\right) = 0\,. \tag{152}$$

The gauge transformations of the above equations are given by

$$\delta e_{[9]}{}^{a[8],b} = \mathrm{d}\lambda_{[8]}{}^{a[8],b} - h_{c[2]}\alpha_{[7]}{}^{c[2]\langle a[8],b\rangle}\,, \tag{153a}$$

$$\delta\omega_{[8]}{}^{a[10],b} = \mathrm{d}\alpha_{[7]}{}^{a[10],b} - h_{c[8]}\beta^{a[10],c[8]b}\,, \tag{153b}$$

$$\delta X_{[1]}{}^{a[10],b[9]} = \mathrm{d}\beta^{a[10],b[9]}\,. \tag{153c}$$

Schematically, we once again decompose everything in terms of irreducible components:

$$e_{[9]}{}^{8,1} = h_{9,8,1}^{(1)} + \widehat{A}_{10,7,1} + \widehat{A}_{10,8} + \widehat{A}_{11,6,1} + \widehat{A}_{11,7}\,, \tag{154a}$$

$$\lambda_{[8]}{}^{8,1} = \lambda_{8,8,1}^{(1)} + \lambda_{9,7,1}^{(2)} + \lambda_{9,8}^{(3)} + \lambda_{10,6,1}^{(4)} + \lambda_{10,7}^{(5)} + \lambda_{11,5,1}^{(6)} + \lambda_{11,6}^{(7)}\,, \tag{154b}$$

$$\alpha_{[7]}{}^{10,1} = \alpha_{10,7,1}^{(1)} + \alpha_{10,8}^{(2)} + \alpha_{11,6,1}^{(3)} + \alpha_{11,7}^{(4)}\,. \tag{154c}$$

The variable $e_{[9]}{}^{8,1}$ contains the irreducible field $h_{9,8,1}$ alongside four extra fields: $\widehat{A}_{10,7,1}$, $\widehat{A}_{10,8}$, $\widehat{A}_{11,6,1}$, and $\widehat{A}_{11,7}$. These extra fields can be set to zero using the components of $\alpha_{[7]}{}^{10,1}$ so that only $h_{9,8,1}$ remains, whereafter there will remain some residual gauge symmetry and the $\alpha$ parameters will be related to first derivatives of the $\lambda$ parameters.

It is useful to denote the number of higher dualisations with a superscript, distinguishing the first higher dual graviton $h_{9,8,1}^{(1)}$ at level six from the dual graviton $h_{8,1}$ at level three. Here we propose a first-order on-shell duality relation between the fields $h_{9,8,1}^{(1)}$ and $h_{8,1}$ in terms of their first-order connections:

$$\omega_{a_1\cdots a_8|b_1\cdots b_{10},c}^{(1)} \propto \varepsilon_{b_1\cdots b_{10}}{}^d\,\omega_{c|a_1\cdots a_8 d}\,. \tag{155}$$

As for all the duality relations that we propose in this paper, we are working at the linearised level so the constant of proportionality can be absorbed by a redefinition of the fields. However, the tensor structure and the precise coefficients in the full non-linear relations would be fixed by $E_{11}$ symmetry, as explained in the introduction.

In the same way that (114) and (136) hold exactly when the parameters are constrained to be pure gauge-for-gauge, our higher duality relation (155) holds exactly in a gauge where $\beta^{10,9}$ is related to $\alpha_{[7]}{}^{10,1}$ and $\alpha^9$ via the constraint[12]

$$\partial_{[a_1}\alpha_{a_2\cdots a_8]|b_1\cdots b_{10},c} - \beta_{b_1\cdots b_{10},a_1\cdots a_8 c} \propto \varepsilon_{b_1\cdots b_{10}}{}^d\partial_c\alpha_{a_1\cdots a_8 d}\,, \tag{156}$$

where the constant of proportionality is the same as that of (155). This constraint is invariant under gauge-for-gauge transformations. As at previous levels, the constraints on the gauge parameters lead to extra freedom at the level of the fields, hence the extra fields in (155).

Taking derivatives of (155) leads to a gauge-invariant relation

$$\partial_{[c_1}\partial_{[b_1}\partial_{[a_1}h_{a_2\cdots a_{10}],b_2\cdots b_9],c_2]}^{(1)} \propto \varepsilon_{a_1\cdots a_{10}}{}^d\partial_d\partial_{[c_1}\partial_{[b_1}h_{b_2\cdots b_9],c_2]}\,. \tag{157}$$

---

[12]As in previous sections, we rescale $p$-forms by factors $p!$ when writing equations in components.

The equation of motion for $h_{8,1}$ was given in (45) and under (157) it is equivalent to

$$\partial^{[c}\partial_{[b_1}\partial_{[a_1}h^{(1)}_{a_2\cdots a_{10}],b_2\cdots b_8 d],}{}^{d]} = 0\,.\tag{158}$$

Moreover, going back to (157), antisymmetring $d$ with $b[9]$ or $c[2]$ causes the right-hand side to vanish, leading to two on-shell constraints for the $h_{9,8,1}$ field:

$$\partial_{[c_1}\partial^{[b_1}\partial_{[a_1}h^{(1)}_{b_1\cdots b_9],}{}^{b_2\cdots b_9],}{}_{c_2]} = 0\,,\qquad \partial^{[c_1}\partial_{[b_1}\partial_{[a_1}h^{(1)}_{a_2\cdots a_8 c_1 c_2],b_2\cdots b_9],}{}^{c_2]} = 0\,.\tag{159}$$

Together, (158) and (159) are the equations of motion for the higher dual graviton $h^{(1)}_{9,8,1}$ whose Young tableau contains more than two columns. As such, we have more than one equation of motion for the $h_{9,8,1}$ field, each with three derivatives, and they are independent of each other.

Solving equations (151a), (151b) and (151c) for the primary zero-form $C_{10,9,2} \in \mathcal{T}(h_{9,8,1})$ in terms of $h^{(1)}_{9,8,1}$ and the extra fields, we find that it is given by the curvature tensor

$$C_{a_1\cdots a_{10},b_1\cdots b_9,c_1 c_2} = \partial_{[c_1}\partial_{[b_1}\partial_{[a_1}h_{a_2\cdots a_{10}],b_2\cdots b_9],c_2]}\,,\tag{160}$$

up to a factor. Since we are unfolding on-shell, $C_{10,9,2}$ obeys higher trace constraints that will be equivalent to the $h^{(1)}_{9,8,1}$ equations of motion. Moreover, $C_{9,2}$ and its descendents $C_{9,2,1,\ldots,1}$ are irreducible Lorentz representations, so they are traceless and satisfy over-antisymmetrisation constraints for GL(11) irreducible tensors. It is useful to rewrite (157) as a relation between $C_{10,9,2} \in \mathcal{T}(h_{9,8,1})$ and $C_{9,2,1} \in \mathcal{T}(h_{8,1})$ where we recall that $C_{9,2,1}$ is really a projection of the gradient of the primary zero-form $C_{9,2}$:

$$C_{a_1\cdots a_{10},b_1\cdots b_9,c_1 c_2} = \varepsilon_{a_1\cdots a_{10}}{}^{d}\, C_{b_1\cdots b_9,c_1 c_2,d}\,.\tag{161}$$

Under this relation, the on-shell properties of the zero-forms are exchanged as

$$\left.\begin{array}{l} \mathrm{Tr}_{2,3}(C_{10,9,2}) = 0\,,\\ \sigma_{2,3}(C_{10,9,2}) = 0\,,\\ (\mathrm{Tr}_{1,3})^2(C_{10,9,2}) = 0\,,\\ (\mathrm{Tr}_{1,2})^9(C_{10,9,2}) = 0\,,\\ \sigma_{1,2}(C_{10,9,2}) = 0\,,\\ \sigma_{1,3}(C_{10,9,2}) = 0 \end{array}\right\} \iff \left\{\begin{array}{l} \mathrm{Tr}_{1,2}(C_{9,2,1}) = 0\,,\\ \sigma_{1,2}(C_{9,2,1}) = 0\,,\\ \sigma_{2,3}(C_{9,2,1}) = 0\,,\\ \sigma_{1,3}(C_{9,2,1}) = 0\,,\\ \mathrm{Tr}_{1,3}(C_{9,2,1}) = 0\,,\\ \mathrm{Tr}_{2,3}(C_{9,2,1}) = 0\,. \end{array}\right.\tag{162}$$

We use $\mathrm{Tr}_{i,j}$ to denote a trace on columns $i$ and $j$, and $\sigma_{i,j}$ denotes over-antisymmetrisation of column $i$ with one index in column $j > i$. For example, one can write $\sigma_{1,3}(h_{9,8,1})$ in place of $h_{[a_1\cdots a_9|,b_1\cdots b_8,|c]}$. A mixed-symmetry field $\phi$ is GL(11) irreducible if and only if $\sigma_{i,j}(\phi) = 0$ for all $i$ and $j$ with $i < j$. Thus the higher trace constraints

$$(\mathrm{Tr}_{1,2})^9(C_{10,9,2}) = 0\,,\qquad (\mathrm{Tr}_{1,3})^2(C_{10,9,2}) = 0\,,\qquad \mathrm{Tr}_{2,3}(C_{10,9,2}) = 0\,,\tag{163}$$

are equivalent to the linearised equations of motion (158) and (159) of the higher dual field $h^{(1)}_{9,8,1}$ when the primary zero-form $C_{10,9,2}$ is expressed as the curvature tensor (160).

Before moving on, we will clarify the linearised equations of motion (163). It may seem strange that we have three independent third-order equations rather than just one second-order equation, but this is unsurprising from the perspective of reference [90]. The idea is that the higher dual field $h^{(1)}_{9,8,1}$ propagates the correct degrees of freedom when its curvature obeys all three equations. Two of the equations arise from the Bianchi identities for the dual graviton $h_{8,1}$ and the third equation appears when taking a gradient of the dual gravity equation (45).

Note that $B_{10,1,1}$ at level four also has three blocks of indices and hence more than one equation of motion, i.e. the complete tracelessness of its curvature $C_{11,2,2}$ in (113).

Note that a Lagrangian formulation would be different. Higher dual action principles require extra fields in addition to the irreducible higher dual field alone, so instead of one equation of motion for the higher dual field, there are several. An action in four dimensions for the higher dual graviton that is second-order in derivatives was found in [74]. We found that an extra field is present inside this action, and neither of the two could be eliminated. There were two standard second-order equations of motion, one for each of the two field.

Working backwards, we can integrate the $b[9]$ column of (157) and use the Poincaré lemma again on $c[2]$ to obtain

$$\partial_{[c_1|}\partial_{[a_1}h^{(1)}_{a_2\cdots a_{10}],b_1\cdots b_8,|c_2]} + \partial_{[c_1|}\partial_{[b_1}\Xi_{b_2\cdots b_8]|a_1\cdots a_{10},|c_2]} \propto \varepsilon_{a_1\cdots a_{10}}{}^d \partial_d \partial_{[c_1|}h_{b_1\cdots b_8,|c_2]}, \tag{164}$$

up to some arbitrary $\Xi_{7|10,1}$ tensor field. Imposing the shift

$$\Xi_{b_1\cdots b_7|a_1\cdots a_{10},c} \longrightarrow \Xi_{b_1\cdots b_7|a_1\cdots a_{10},c} - 8\,\varepsilon_{a_1\cdots a_{10}}{}^d h_{b_1\cdots b_7 d,c}, \tag{165}$$

allows us to write equation (164) as

$$\partial_{[c_1|}\partial_{[a_1}h^{(1)}_{a_2\cdots a_{10}],b_1\cdots b_8,|c_2]} + \partial_{[c_1|}\partial_{[b_1}\Xi_{b_2\cdots b_8]|a_1\cdots a_{10},|c_2]} \propto \varepsilon_{a_1\cdots a_{10}}{}^d \left(9\,\partial_{[c_1}\partial_{[d}h_{b_1\cdots b_8],c_2]}\right). \tag{166}$$

The $c[2]$ column can now be integrated, leading to a first-order relation

$$\partial_{[a_1}h^{(1)}_{a_2\cdots a_{10}],b_1\cdots b_8,c} + \partial_{[b_1}\Xi_{b_2\cdots b_8]|a_1\cdots a_{10},c} \propto \varepsilon_{a_1\cdots a_{10}}{}^d \left(9\,\partial_{[d}h_{b_1\cdots b_8],c} + \partial_c \Theta_{d b_1\cdots b_8}\right). \tag{167}$$

Comparing this with the previous duality relation in the gravity sector (46), we identify the terms in the parentheses with the connection $\omega_{[1]}{}^8$ in (36a). Moreover, $\Theta_9$ is identified with the extra field $\widehat{A}_9$ in (40) and the irreducible components of $\Xi_{7|10,1}$ are identified with the extra fields in (154a). Solving (151a) for $\omega_{[8]}{}^{10,1} = \omega^{(1)}{}_{[8]}{}^{10,1}$ (again using a superscript to denote the number of higher dualisations) leads to

$$\omega^{(1)}_{a_1\cdots a_8|b_1\cdots b_{10},c} = \partial_{[b_1}h^{(1)}_{b_2\cdots b_{10}],a_1\cdots a_8,c} + \partial_{[a_1}\Xi_{a_2\cdots a_8]|b_1\cdots b_{10},c}, \tag{168}$$

on the left-hand side of (167) up to certain factors, while the quantity on the right-hand side in parentheses is the solution (43) of equation (36a). Thus we have integrated up from the equations of motion to obtain the duality relations.

Looking back at equation (166), one can antisymmetrise $a[10]$ with $c_1$ to obtain

$$\partial_{[a_1}\partial_{[b_1}\Xi_{b_2\cdots b_8]|a_2\cdots a_{11}],c} \propto \varepsilon_{a_1\cdots a_{11}}\eta^{c_1 d}\partial_{[c_1}\partial_{[d}h_{b_1\cdots b_8],c_2]}, \tag{169}$$

which vanishes on-shell due to the dual gravity equation of motion (45). This implies that

$$\partial_{[b_1}\Xi_{b_2\cdots b_8]|a_1\cdots a_{10},c} = \partial_{[a_1}\partial_{[b_1}\xi_{b_2\cdots b_8]|a_2\cdots a_{10}],c}, \tag{170}$$

for some tensor $\xi_{7|9,1}$ whose irreducible components have the same Young tableaux as all but one of the irreducible differential gauge parameters in (154b) and (153a). Importantly, these are the only parameters in the gauge transformations of the extra fields (154a), and (167) can now be written in the form

$$\partial_{[a_1}h^{(1)}_{a_2\cdots a_{10}],b_1\cdots b_8,c} + \partial_{[a_1}\partial_{[b_1}\xi_{b_2\cdots b_8]|a_2\cdots a_{10}],c} \propto \varepsilon_{a_1\cdots a_{10}}{}^d \left(9\,\partial_{[d}h_{b_1\cdots b_8],c} + \partial_c \Theta_{d b_1\cdots b_8}\right). \tag{171}$$

We propose that the gravity sector of the non-linear realisation of $E_{11}$ should be extended to include the on-shell duality relations (46) and (155) that are summarised as follows:

$$\omega_{[1]}{}^2 \quad \longleftrightarrow \quad \omega_{[1]}{}^9 \quad \longleftrightarrow \quad \omega^{(1)}{}_{[8]}{}^{10,1}. \tag{172}$$

We will extend this chain of dualities infinitely to higher levels in Section 5.4. In the non-linear realisation, the duality relations are equivalence relations that only hold up to pure gauge terms. So far, we have worked out these gauge terms for the duality relations up to level six. These relations were written using the unfolded variables that are associated with each $E_{11}$ field, and they hold exactly in the sense that they are gauge-covariant. The difference between the duality relations found here and those of the non-linear realisation is that our proposed duality relations necessarily include extra fields that absorb all the gauge freedom.

## 5.3   Unfolding the field $A_{9,9,3}$ at level seven

In this section we will unfold the second higher dual three-form $A_{9,9,3}$ at level seven. This will allow us to work out a first-order duality relation between this field and the first higher dual three-form $A_{9,3}$ at level four. The unfolded variables are

$$e_{[9]}{}^{a[9],b[3]}, \quad \omega_{[9]}{}^{a[10],b[3]}, \quad X_{[3]}{}^{a[10],b[10]}, \quad C^{a[10],b[10],c[4]}, \quad \ldots \tag{173}$$

The first three unfolded equations are

$$\mathrm{d}e_{[9]}{}^{a[9],b[3]} + h_c\, \omega_{[9]}{}^{c\langle a[9],b[3]\rangle} = 0, \tag{174a}$$

$$\mathrm{d}\omega_{[9]}{}^{a[10],b[3]} + h_{c[7]} X_{[3]}{}^{a[10],c[7]b[3]} = 0, \tag{174b}$$

$$\mathrm{d}X_{[3]}{}^{a[10],b[10]} + h_{c[4]} C^{a[10],b[10],c[4]} = 0, \tag{174c}$$

where the angled brackets denote the projection of the final twelve indices of $\omega_{[9]}{}^{10,3}$ onto the irreducible $\mathbb{Y}[9,3]$ tableau. As usual, the primary zero-form $C_{10,10,4}$ belongs to the tower

$$\mathcal{T}(A_{9,9,3}) = \left\{ C^{(n)}_{10,10,4,1^n} \,\middle|\, n \in \mathbb{N} \right\} = \left\{ C^{(0)}_{10,10,4}, C^{(1)}_{10,10,4,1}, C^{(2)}_{10,10,4,1,1}, \ldots \right\}. \tag{175}$$

The unfolded equations are invariant under the gauge transformations

$$\delta e_{[9]}{}^{a[9],b[3]} = \mathrm{d}\lambda_{[8]}{}^{a[9],b[3]} + h_c\, \alpha_{[8]}{}^{c\langle a[9],b[3]\rangle}, \tag{176a}$$

$$\delta \omega_{[9]}{}^{a[10],b[3]} = \mathrm{d}\alpha_{[8]}{}^{a[10],b[3]} + h_{c[7]} \beta_{[2]}{}^{a[10],c[7]b[3]}, \tag{176b}$$

$$\delta X_{[3]}{}^{a[10],b[10]} = \mathrm{d}\beta_{[2]}{}^{a[10],b[10]}. \tag{176c}$$

As for any set of unfolded equations involving higher-degree forms, for each parameter there is a family of reducibility (gauge-for-gauge) transformations

$$\delta \lambda_{[8-k]}{}^{a[9],b[3]} = \mathrm{d}\lambda_{[7-k]}{}^{a[9],b[3]} + h_c\, \alpha_{[7-k]}{}^{c\langle a[9],b[3]\rangle}, \tag{177a}$$

$$\delta \alpha_{[8-k]}{}^{a[10],b[3]} = \mathrm{d}\alpha_{[7-k]}{}^{a[10],b[3]} + h_{c[7]} \beta_{[1-k]}{}^{a[10],b[3]c[7]}, \tag{177b}$$

$$\delta \beta_{[2-k]}{}^{a[10],b[10]} = \mathrm{d}\beta_{[1-k]}{}^{a[10],b[10]}, \tag{177c}$$

where $k = 0, 1, \ldots, 7$ in (177a) and (177b), and $k = 0, 1$ in (177c). It is understood that a $p$-form with negative form degree is identically zero.

The irreducible fields and connections are given by

$$e_{[9]}{}^{9,3} = A_{9,9,3} + \widehat{A}_{10,8,3} + \widehat{A}_{10,9,2} + \widehat{A}_{11,7,3} + \widehat{A}_{11,8,2} + \widehat{A}_{11,9,1}, \tag{178a}$$

$$\omega_{[9]}{}^{10,3} = \omega_{10,9,3} + \omega_{10,10,2} + \omega_{11,8,3} + \omega_{11,9,2} + \omega_{11,10,1}, \tag{178b}$$

$$X_{[3]}{}^{10,10} = X_{10,10,3} + X_{11,10,2}. \tag{178c}$$

The variable $e_{[9]}{}^{9,3}$ contains the irreducible field $A_{9,9,3}$ alongside five extra fields $\widehat{A}_{10,8,3}$, $\widehat{A}_{10,9,2}$, $\widehat{A}_{11,7,3}$, $\widehat{A}_{11,8,2}$, and $\widehat{A}_{11,9,1}$. The irreducible gauge parameters are

$$\lambda_{[8]}{}^{9,3} = \lambda_{9,8,3}^{(1)} + \lambda_{9,9,2}^{(2)} + \lambda_{10,7,3}^{(3)} + \lambda_{10,8,2}^{(4)} + \lambda_{10,9,1}^{(5)} + \lambda_{11,6,3}^{(6)} + \lambda_{11,7,2}^{(7)} + \lambda_{11,8,1}^{(8)} + \lambda_{11,9}^{(9)},$$
(179a)

$$\alpha_{[8]}{}^{10,3} = \alpha_{10,8,3}^{(1)} + \alpha_{10,9,2}^{(2)} + \alpha_{10,10,1}^{(3)} + \alpha_{11,7,3}^{(4)} + \alpha_{11,8,2}^{(5)} + \alpha_{11,9,1}^{(6)} + \alpha_{11,10}^{(7)},$$
(179b)

$$\beta_{[2]}{}^{10,10} = \beta_{10,10,2}^{(1)} + \beta_{11,10,1}^{(2)}.$$
(179c)

Now we will explain the role of each of each component. The five extra fields in (178a) can be set to zero using the gauge parameters

$$\alpha_{10,8,3}^{(1)}, \quad \alpha_{10,9,3}^{(2)}, \quad \alpha_{11,7,3}^{(4)}, \quad \alpha_{11,8,2}^{(5)}, \quad \alpha_{11,9,1}^{(6)}.$$
(180)

After all the extra fields are eliminated, there will still exist some gauge symmetry in terms of the $\alpha_{10,10,1}^{(3)}$ and $\alpha_{11,10}^{(7)}$ parameters. It seems that we are trying to gauge away two fields that do not exist. However, the reducibility transformation

$$\delta\alpha_{[8]}{}^{a[10],b[3]} = d\alpha_{[7]}{}^{a[10],b[3]} + h_{c[7]}\beta_{[1]}{}^{a[10],b[3]c[7]},$$
(181)

in (177b) tells us that $\alpha_{10,10,1}^{(3)}$ and $\alpha_{11,10}^{(7)}$ can both be shifted away using the components of the gauge-for-gauge parameter $\beta_{[1]}{}^{10,10}$ in (179c).

In equation (178b) the connection $\omega_{[9]}{}^{10,3}$ is decomposed into five irreducible components. Two of them can be set to zero using the parameter $\beta_{[2]}{}^{10,10}$ in (176b) and the other three are used to express $\omega_{[9]}{}^{10,3}$ in terms of $(de)_{[10]}{}^{9,3}$. Now notice that $(d\omega)_{[10]}{}^{10,3}$ has three components. One of them vanishes as a result of the Bianchi identity $(d^2 e)_{[11]}{}^{9,3} = 0$ and the other two are used when $X_{[3]}{}^{10,10}$ is expressed in terms of $(d\omega)_{[10]}{}^{10,3}$. In exactly the same way, one of the two components of $(dX)_{[4]}{}^{10,10}$ vanishes due to the Bianchi identity $(d^2\omega)_{[11]}{}^{9,3} = 0$, while the other one is used to express $C_{10,10,4}$ in terms of $(dX)_{[4]}{}^{10,10}$. Consequently, after solving the unfolded equations and using the Bianchi identities and gauge symmetries, the primary zero-form $C_{10,10,4}$ in (174c) can be expressed entirely in terms of the $A_{9,9,3}$ field.

Recall that the components of $\alpha_{[8]}{}^{10,3}$ either shift away the extra fields or are shifted away themselves using the reducibility parameter $\beta_{[1]}{}^{10,10}$. Therefore, we find that all the gauge and gauge-for-gauge parameters account for each other except for the field $A_{9,9,3}$ itself, the gauge parameters $\{\lambda_{9,8,3}^{(1)}, \lambda_{9,9,2}^{(2)}\}$, and a small set of reducibility parameters $\{\lambda_{9,7,3}, \lambda_{9,8,2}, \lambda_{9,9,1}, \dots\}$.

We now propose a first-order duality relation between the first higher dual three-form $A_{9,3}^{(1)}$ at level four and the second higher dual three-form $A_{9,9,3}^{(2)}$ at level seven. As previously explained, the superscripts denote the number of higher dualisations. Our duality relation takes the form

$$\omega_{a_1 \cdots a_9 | b_1 \cdots b_{10}, c_1 c_2 c_3}^{(2)} \propto \varepsilon_{b_1 \cdots b_{10}}{}^{d}\, \omega_{c_1 c_2 c_3 | d a_1 \cdots a_9}^{(1)}.$$
(182)

All the irreducible components of $e_{[9]}{}^{9,3}$ in (178a) appear inside equation (182). Importantly, this on-shell duality relation holds exactly. Equation (182) is gauge-invariant when the gauge parameters $\beta_{[2]}{}^{10,10}$, $\alpha_{[8]}{}^{10,3}$, and $\alpha_{[2]}{}^{10}$ for the connections in (182) are related by

$$\partial_{[a_1}\alpha_{a_2 \cdots a_9]|b_1 \cdots b_{10}, c_1 c_2 c_3} + \beta_{[a_1 a_2 || b_1 \cdots b_{10}, |a_3 \cdots a_9]c_1 c_2 c_3} \propto \varepsilon_{b_1 \cdots b_{10}}{}^{d}\partial_{c_1}\alpha_{c_2 c_3 | d a_1 \cdots a_9},$$
(183)

which is analogous to (156). The constant of proportionality is the same as (182). Recall that the previous duality relation (114) is gauge-invariant under the constraint (125) which forces $\alpha_{[2]}{}^{10}$ to be pure gauge-for-gauge. Under (183), this constraint now implies a further

constraint on $\alpha_{[8]}{}^{10,3}$ and $\beta_{[2]}{}^{10,10}$ which leads to the first-order connection $\omega^{(2)}{}_{[9]}{}^{10,3}$ being gauge-invariant:

$$\partial_{[a_1}\alpha_{a_2\cdots a_9]|b_1\cdots b_{10},c_1c_2c_3} + \beta_{[a_1a_2||b_1\cdots b_{10},|a_3\cdots a_9]c_1c_2c_3} = 0\,. \tag{184}$$

The gauge parameter constraints at higher levels will continue to enforce the gauge invariance of all the first-order on-shell duality relations.

Taking derivatives of (182) leads to the relation

$$\partial_{[c_1}\partial_{[b_1}\partial_{[a_1}A^{(2)}_{a_2\cdots a_{10}],b_2\cdots b_{10}],c_2c_3c_4]} \propto \varepsilon_{a_1\cdots a_{10}}{}^d\partial_d\partial_{[c_1}\partial_{[b_1}A^{(1)}_{b_2\cdots b_{10}],c_2c_3c_4]}\,. \tag{185}$$

Working on-shell, the $A_{9,3}$ equation of motion (93) is equivalent under (185) to

$$\partial^{[c_1}\partial_{[b_1}\partial_{[a_1}A^{(2)}_{a_2\cdots a_{10}],b_2\cdots b_6c_1\cdots c_4],}{}^{c_2c_3c_4]} = 0\,. \tag{186}$$

In addition, antisymmetrising $d$ with $b[10]$ or $c[4]$ causes the right-hand side of (185) to vanish, so the left-hand side is subject to some further on-shell constraints:

$$\partial_{[c_1}\partial^{[b_1}\partial_{[b_1}A^{(2)}_{b_2\cdots b_{10}],}{}^{b_2\cdots b_{10}],}{}_{c_2c_3c_4]} = 0\,, \qquad \partial^{[c_1}\partial_{[b_1}\partial_{[a_1}A^{(2)}_{a_2\cdots a_6c_1\cdots c_4],b_2\cdots b_9],}{}^{c_2c_3c_4]} = 0\,. \tag{187}$$

Equations (186) and (187) are the equations of motion for the $A^{(2)}_{9,9,3}$ field.

Solving the unfolded equations (174a), (174b) and (174c) for the primary zero-form, we find that $C_{10,10,4}$ can be expressed up to a factor as the curvature tensor

$$C_{a_1\cdots a_{10},b_1\cdots b_{10},c_1\cdots c_4} = \partial_{[c_1}\partial_{[b_1}\partial_{[a_1}A_{a_2\cdots a_{10}],b_2\cdots b_{10}],c_2c_3c_4]}\,. \tag{188}$$

We can now rewrite (185) as a relation between $C_{10,10,4} \in \mathcal{T}(A^{(2)}_{9,9,3})$ and $C_{10,4,1} \in \mathcal{T}(A^{(1)}_{9,3})$:

$$C_{a_1\cdots a_{10},b_1\cdots b_{10},c_1c_2c_3c_4} \propto \varepsilon_{a_1\cdots a_{10}}{}^d C_{b_1\cdots b_{10},c_1c_2c_3c_4,d}\,. \tag{189}$$

As a result, $C_{10,10,4}$ inherits the constraints

$$(\text{Tr}_{2,3})^4(C_{10,10,4}) = 0\,, \qquad \sigma_{2,3}(C_{10,10,4}) = 0\,, \tag{190}$$

and the remaining constraints are exchanged under (189) as

$$\left.\begin{array}{l}(\text{Tr}_{1,2})^{10}(C_{10,10,4}) = 0\,, \\ (\text{Tr}_{1,3})^4(C_{10,10,4}) = 0\,, \\ \sigma_{1,2}(C_{10,10,4}) = 0\,, \\ \sigma_{1,3}(C_{10,10,4}) = 0\end{array}\right\} \iff \left\{\begin{array}{l}\sigma_{1,3}(C_{10,4,1}) = 0\,, \\ \sigma_{2,3}(C_{10,4,1}) = 0\,, \\ \text{Tr}_{1,3}(C_{10,4,1}) = 0\,, \\ \text{Tr}_{2,3}(C_{10,4,1}) = 0\,.\end{array}\right. \tag{191}$$

We can combine the zero-form relations (189) and (91) to obtain a new relation between $C_{10,10,4} \in \mathcal{T}(A_{9,9,3})$ and $F^{(2)}_{4,1,1} \in \mathcal{T}(A_3)$:

$$C_{a_1\cdots a_{10},b_1\cdots b_{10},c_1c_2c_3c_4} = \varepsilon_{a_1\cdots a_{10}}{}^{d_1}\varepsilon_{b_1\cdots b_{10}}{}^{d_2}F^{(2)}_{c_1c_2c_3c_4,d_1,d_2}\,. \tag{192}$$

Taking a curl on the $c[4]$ indices gives

$$\partial_{[e|}C_{a_1\cdots a_{10},b_1\cdots b_{10},|c_1\cdots c_4]} = \varepsilon_{a_1\cdots a_{10}}{}^{d_1}\varepsilon_{b_1\cdots b_{10}}{}^{d_2}\partial_{[e}F^{(2)}_{c_1\cdots c_4],d_1,d_2}\,. \tag{193}$$

Equation (61) tells us that $F^{(3)}_{4,1,1,1} \in \mathcal{T}(A_3)$ is really the GL(11) irreducible projection of the gradient of the adjacent zero-form $F^{(2)}_{4,1,1} \in \mathcal{T}(A_3)$, so equation (193) becomes

$$\partial_{[e|}C_{a_1\cdots a_{10},b_1\cdots b_{10},|c_1\cdots c_4]} = \varepsilon_{a_1\cdots a_{10}}{}^{d_1}\varepsilon_{b_1\cdots b_{10}}{}^{d_2}F^{(3)}_{[c_1\cdots c_4|,d_1,d_2,|e]} = 0\,. \tag{194}$$

The generalised Poincaré lemma [90] applied to (194) implies that $C_{10,10,4}$ can be expressed as the curvature tensor (188). This method will be useful when we consider fields at all higher levels since it allows us to express primary zero-forms and their gradients in terms of $E_{11}$ fields without needing to solve an arbitrary number of unfolded equations.

Now we will present an equivalent method of obtaining the linearised equations of motion, i.e. the higher trace constraints. On-shell, all the zero-forms $F^{(n)}_{4,1,\dots,1} \in \mathcal{T}(A_3)$ are irreducible Lorentz representations, and as a result the properties of $F^{(2)}_{4,1,1}$ are exchanged under (192) with constraints on the curvature tensor $C_{10,10,4}$ as follows:

$$
\left.
\begin{aligned}
(\mathrm{Tr}_{1,2})^{10}(C_{10,10,4}) &= 0\,, \\
(\mathrm{Tr}_{1,3})^{4}(C_{10,10,4}) &= 0\,, \\
(\mathrm{Tr}_{2,3})^{4}(C_{10,10,4}) &= 0\,, \\
\sigma_{1,2}(C_{10,10,4}) &= 0\,, \\
\sigma_{1,3}(C_{10,10,4}) &= 0\,, \\
\sigma_{2,3}(C_{10,10,4}) &= 0
\end{aligned}
\right\}
\quad \Longleftrightarrow \quad
\left\{
\begin{aligned}
\mathrm{Tr}_{2,3}(F_{4,1,1}) &= 0\,, \\
\sigma_{1,2}(F_{4,1,1}) &= 0\,, \\
\sigma_{1,3}(F_{4,1,1}) &= 0\,, \\
\sigma_{2,3}(F_{4,1,1}) &= 0\,, \\
\mathrm{Tr}_{1,2}(F_{4,1,1}) &= 0\,, \\
\mathrm{Tr}_{1,3}(F_{4,1,1}) &= 0\,.
\end{aligned}
\right.
\tag{195}
$$

Our notation $\mathrm{Tr}_{i,j}$ and $\sigma_{i,j}$ is the same as in (162). Thus the postulated relation (192) leads to the higher trace constraints:

$$
(\mathrm{Tr}_{1,2})^{10}(C_{10,10,4}) = 0\,, \qquad (\mathrm{Tr}_{1,3})^{4}(C_{10,10,4}) = 0\,, \qquad (\mathrm{Tr}_{2,3})^{4}(C_{10,10,4}) = 0\,.
\tag{196}
$$

When $C_{10,10,4}$ is expressed as the curvature tensor (188), these trace constraints are equivalent to the equations of motion (186) and (187) for $A^{(2)}_{9,9,3}$. Even if we unfold off-shell without these constraints, it is immediate to see that $C_{10,10,4}$ is invariant under the gauge transformation

$$
\delta A_{a_1 \cdots a_9, b_1 \cdots b_9, c_1 c_2 c_3} = \left[ \partial_{[b_1} \lambda^{(1)}_{a_1 \cdots a_9, |b_2 \cdots b_9], c_1 c_2 c_3} + \partial_{[c_1} \lambda^{(2)}_{a_1 \cdots a_9, b_1 \cdots b_9, |c_2 c_3]} \right]_{9,9,3}\,,
\tag{197}
$$

where $[\cdots]_{9,9,3}$ denotes a projection onto the GL(11) irreducible $\mathbb{Y}[9,9,3]$ tableau.

Working backwards from the third-order curvature relation (185), we can integrate $b[10]$ to introduce an arbitrary $\Xi_{8|10,3}$ tensor, and then we can shift it as

$$
\Xi_{b_1 \cdots b_8 | a_1 \cdots a_{10}, c_1 c_2 c_3} \longmapsto \Xi_{b_1 \cdots b_8 | a_1 \cdots a_{10}, c_1 c_2 c_3} + 9\, \varepsilon_{a_1 \cdots a_{10}}{}^{d} A^{(1)}_{b_1 \cdots b_8 d, c_1 c_2 c_3}\,,
\tag{198}
$$

leading to a second-order relation

$$
\partial_{[c_1|} \partial_{[a_1} A^{(2)}_{a_2 \cdots a_{10}], b_1 \cdots b_9, |c_2 c_3 c_4]} + \partial_{[c_1|} \partial_{[b_1} \Xi_{b_2 \cdots b_9]|a_1 \cdots a_{10}, |c_2 c_3 c_4]} \propto \varepsilon_{a_1 \cdots a_{10}}{}^{d} \left( 10\, \partial_{[c_1} \partial_{[d} A^{(1)}_{b_1 \cdots b_9], c_2 c_3 c_4]} \right),
\tag{199}
$$

where we have used the Poincaré lemma again on $c[4]$ to make the curl on these indices explicit in every term. Integrating on $c[4]$ now gives us the first-order relation (182) in the form

$$
\partial_{[a_1} A^{(2)}_{a_2 \cdots a_{10}], b_1 \cdots b_9, c_1 c_2 c_3} + \partial_{[b_1} \Xi_{b_2 \cdots b_9]|a_1 \cdots a_{10}, c_1 c_2 c_3} \propto \varepsilon_{a_1 \cdots a_{10}}{}^{d} \left( 10\, \partial_{[d} A^{(1)}_{b_1 \cdots b_9], c_1 c_2 c_3} + \partial_{[c_1} \Theta_{c_2 c_3]|d b_1 \cdots b_9} \right).
\tag{200}
$$

The irreducible components of $\Theta_{2|10}$ are identified with the two extra fields in (79a) and the irreducible components of $\Xi_{8|10,3}$ are identified either with the set of extra fields in (178a) or with the components of $\alpha_{[8]}{}^{10,3}$ that can be set to zero using the gauge-for-gauge parameter $\beta_{[1]}{}^{10,10}$ in (177c).

The duality relation (182) holds exactly. However, it can be written as an equivalence relation between the first terms on the left-hand and right-hand sides of (200). The precise meaning of this equivalence relation is explained in (200) which is found by integrating either the equation of motion of $A^{(1)}_{9,3}$ or that of $A^{(2)}_{9,9,3}$.

### 5.4 Unfolding and duality relations at higher levels

In this section, we restrict our attention to the irreducible fields $\{A^{(n)}_{9,\dots,9,3}, A^{(n)}_{9,\dots,9,6}, h^{(n)}_{9,\dots,9,8,1}\}$ in (9) whose blocks of antisymmetric indices are no larger than nine, and we propose first-order duality relations between them at arbitrarily high levels. As discussed at lower levels, we found that they are relations between the first-order connections associated with each field.

**Unfolding higher dual three-forms.** In order to unfold the $n^{\text{th}}$ higher dual $A^{(n)}_{9^n,3}$ in $E_{11}$ at level $3n+1$, we introduce the following variables:

$$e_{[9]}{}^{9^{n-1},3}, \quad \omega_{[9]}{}^{10,9^{n-2},3}, \quad X_{[9]}{}^{10^2,9^{n-3},3}, \quad \dots, \quad X_{[9]}{}^{10^{n-1},3}, \quad X_{[3]}{}^{10^n}, \quad C^{10^n,4}, \quad \dots \quad (201)$$

Schematically, the first two unfolded equations can be written as

$$\mathrm{d}e_{[9]}{}^{9^{n-1},3} + h_1\,\omega_{[9]}{}^{10,9^{n-2},3} = 0\,, \tag{202a}$$

$$\mathrm{d}\omega_{[9]}{}^{10,9^{n-2},3} + h_1 X_{[9]}{}^{10^2,9^{n-3},3} = 0\,, \tag{202b}$$

and they are invariant under the gauge symmetries

$$\delta e_{[9]}{}^{9^{n-1},3} = \mathrm{d}\lambda_{[8]}{}^{9^{n-1},3} + h_1\,\alpha_{[8]}{}^{10,9^{n-2},3}\,, \tag{203a}$$

$$\delta \omega_{[9]}{}^{10,9^{n-2},3} = \mathrm{d}\alpha_{[8]}{}^{10,9^{n-2},3} + h_1\,\beta_{[8]}{}^{10^2,9^{n-3},3}\,, \tag{203b}$$

$$\delta X_{[9]}{}^{10^2,9^{n-3},3} = \mathrm{d}\beta_{[8]}{}^{10^2,9^{n-3},3}\,. \tag{203c}$$

The primary zero-form $C_{10^n,4}$ is the first in the tower

$$\mathcal{T}(A^{(n)}_{9^n,3}) = \{C^{(m)}_{10^n,4,1^m} \mid m \in \mathbb{N}\} = \{C^{(0)}_{10^n,4}, C^{(1)}_{10^n,4,1}, C^{(2)}_{10^n,4,1,1}, \dots\}\,. \tag{204}$$

The first variable in the tower $e_{[9]}{}^{9^{n-1},3}$ decomposes into irreducible components as

$$e_{[9]}{}^{9^{n-1},3} = A^{(n)}_{9^n,3} + \widehat{A}_{10,9^{n-1},8,3} + \widehat{A}_{10,9^{n-1},2} + \widehat{A}_{11,9^{n-2},7,3} + \widehat{A}_{11,9^{n-2},8,2} + \widehat{A}_{11,9^{n-1},1}\,, \tag{205}$$

where we can see the higher dual field $A^{(n)}_{9^n,3}$ alongside five extra fields. Upon solving the first unfolded equation, the second variable $\omega_{[9]}{}^{10,9^{n-2},3}$ will be given in terms of derivatives of these six fields.

When two fields are related by electromagnetic duality, there must be a bijection between their zero-form modules. For example, the six-form in $E_{11}$ at level two is the magnetic dual of the three-form at level one, and the zero-forms $F^{(n)}_{4,1,\dots,1} \in \mathcal{T}(A_3)$ are related to the zero-forms $F^{(n)}_{7,1,\dots,1} \in \mathcal{T}(A_6)$ via (68) and (69) that we reproduce here:

$$F^{(n)}_{a_1\cdots a_7, c_1,\dots,c_n} = \varepsilon_{a_1\cdots a_7}{}^{b_1\cdots b_4} F^{(n)}_{b_1\cdots b_4, c_1,\dots,c_n}\,, \qquad n = 0, 1, 2, \dots \tag{206}$$

These relations are different in the case of higher (gradient) dualisations. For example, $A_{9,3}$ is the first higher dual three-form field, and the zero-forms $C^{(n)}_{10,4,1^n} \in \mathcal{T}(A_{9,3})$ are related to the zero-forms $F^{(n+1)}_{4,1^{n+1}} \in \mathcal{T}(A_3)$ by the shifted relations

$$C^{(n)}_{a_1\cdots a_{10}, b_1\cdots b_4, c^1,\dots,c^n} = \varepsilon_{a_1\cdots a_{10}}{}^{d} F^{(n+1)}_{b_1\cdots b_4, c_1,\dots,c_n, d}\,. \tag{207}$$

This is not a bijection since $F_4 \equiv F^{(0)}_4$ does not correspond to any zero-form in $\mathcal{T}(A_{9,3})$.

As explained in [23], considering only the three-form sector for the sake of definiteness, we need all the zero-forms in $\mathcal{T}(A_3)$ at a point in space-time $x_0$ together with the infinite

tower of unfolded equations (56), (59) and (60) in order to reconstruct an on-shell dynamical three-form field in some open neighbourhood around $x_0$ using the Taylor expansion

$$A_{a[3]}(x) = A_{a[3]}(x_0) + \sum_{n=1}^{\infty} \frac{1}{n!}(x-x_0)^{b_1}\cdots(x-x_0)^{b_n} F^{(n-1)}_{a_1 a_2 a_3(b_1, b_2, \ldots, b_n)}(x_0). \tag{208}$$

If we were to write down a Taylor expansion for $A_{9,3}$ analogous to (208), the coefficients that are usually given in terms of the tensors $\{C^{(0)}_{10,4}, C^{(1)}_{10,4,1}, C^{(2)}_{10,4,1,1}, \ldots\}$ would instead be given in terms of $\{F^{(1)}_{4,1}, F^{(2)}_{4,1,1}, F^{(3)}_{4,1,1,1}, \ldots\}$. Notice that the zero-form $F^{(0)}_4$ in the linear term of the three-form expansion is no longer present. Thus the first higher dual $A_{9,3}$ describes the on-shell dynamical three-form beyond first-order, i.e. at long distances. This truncation only omits one of the zero-forms in $\mathcal{T}(A_3)$ and the field equations can still be reconstructed by integrating Bianchi identities.

**Duality relations for higher dual three-forms.** We have already found a duality relation between $A^{(2)}_{9,9,3}$ and $A^{(1)}_{9,3}$ in equation (182) and now we propose, in the context of the unfolded formalism, an infinite number of first-order on-shell duality relations for the entire three-form sector. In particular, we relate pairs of adjacent higher dual fields $A^{(n)}_{9^n,3}$ and $A^{(n-1)}_{9^{n-1},3}$ for $n > 2$. These higher relations have a different form to (182) between the first and second higher dual three-forms. The duality relations at all higher levels are

$$\omega^{(n)}{}_{a[9]|b[10],c[9],d^1[9],\ldots,d^{n-3}[9],e[3]} \propto \varepsilon_{b[10]}{}^p \omega^{(n-1)}{}_{a[9]|pc[9],d^1[9],\ldots,d^{n-3}[9],e[3]}, \tag{209}$$

where $\omega^{(n)}{}_{[9]}{}^{10,9^{n-2},3}$ are the first-order connections associated with $A^{(n)}_{9^n,3}$ in (202a).

We require that our duality relations are gauge-covariant, so taking the gauge transformation of both sides leads to a relation between $\alpha_{[8]}{}^{10,9,3}$, $\beta_{[8]}{}^{10,10,3}$, $\alpha_{[8]}{}^{10,3}$, and $\beta_{[2]}{}^{10,10}$ for $n = 3$:

$$\partial_{[a_1}\alpha_{a_2\cdots a_9]|b[10],c[9],d[3]} + \beta_{[a_1\cdots a_8||b[10],|a_9]c[9],d[3]}$$
$$\propto \varepsilon_{b[10]}{}^p \left( \partial_{[a_1}\alpha_{a_2\cdots a_9]|pc[9],d[3]} + \beta_{[a_1 a_2||pc[9],|a_3\cdots a_9]d[3]} \right). \tag{210}$$

For the higher duality relations with $n > 3$, we have the constraints

$$\partial_{[a_1}\alpha_{a_2\cdots a_9]|b[10],c[9],d^1[9],\ldots,d^{n-3}[9],e[3]} + \beta_{[a_1\cdots a_8||b[10],|a_9]c[9],d^1[9],\ldots,d^{n-3}[9],e[3]}$$
$$\propto \varepsilon_{b[10]}{}^p \left( \partial_{[a_1}\alpha_{a_2\cdots a_9]|pc[9],d^1[9],\ldots,d^{n-3}[9],e[3]} + \beta_{[a_1\cdots a_8||pc[9],|a_9]d^1[9],d^2[9],\ldots,d^{n-3}[9],e[3]} \right). \tag{211}$$

Thus for all our duality relations to be gauge-covariant, we need to impose an infinite tower of gauge parameter constraints for $n \geq 3$, each of which follows from the previous one:

$$\partial_{[a_1}\alpha_{a_2\cdots a_9]|b[10],c[9],d^1[9],\ldots,d^{n-3}[9],e[3]} + \beta_{[a_1\cdots a_8||b[10],|a_9]c[9],d^1[9],\ldots,d^{n-3}[9],e[3]} = 0. \tag{212}$$

These constraints create more field degrees of freedom, and they force every connection $\omega^{(n)}$ to be gauge-invariant. Consequently, we have an infinite set of extra fields that appear explicitly in the tower of duality relations.

Taking derivatives of (209) leads to the gauge-invariant relation

$$\partial_{[b_1}\partial_{[a_1^n|}\cdots\partial_{[a_1^1}A^{(n)}_{a_2^1\cdots a_{10}^1],\ldots,|a_2^n\cdots a_{10}^n],b_2 b_3 b_4]}$$
$$\propto \varepsilon_{a^1[10]}{}^c \partial_c \partial_{[b_1}\partial_{[a_1^n|}\cdots\partial_{[a_1^2}A^{(n-1)}_{a_2^2\cdots a_{10}^2],\ldots,|a_2^n\cdots a_{10}^n],b_2 b_3 b_4]}. \tag{213}$$

Now we want to show that taking appropriate traces leads to the equations of motion for each field. For this we suppose that the equations of motion for $A^{(n-1)}_{9^{n-1},3}$ are

$$\eta^{a_1^i a_1^j} \cdots \eta^{a_{10}^i a_{10}^j} \partial_{[b_1} \partial_{[a_1^{n-1}|} \cdots \partial_{[a_1^1} A^{(n-1)}_{a_2^1 \cdots a_{10}^1],\ldots,|a_2^{n-1} \cdots a_{10}^{n-1}],b_2 b_3 b_4]} = 0, \tag{214}$$

$$\eta^{a_1^i b_1} \cdots \eta^{a_4^i b_4} \partial_{[b_1} \partial_{[a_1^{n-1}|} \cdots \partial_{[a_1^1} A^{(n-1)}_{a_2^1 \cdots a_{10}^1],\ldots,|a_2^{n-1} \cdots a_{10}^{n-1}],b_2 b_3 b_4]} = 0, \tag{215}$$

for all $i$ and $j$ with $1 \le i < j \le n-1$. These equations generalise (116), (186), and (187) that we found earlier for low values of $n$. From this, $A^{(n)}_{9^n,3}$ inherits

$$\eta^{a_1^i a_1^j} \cdots \eta^{a_{10}^i a_{10}^j} \partial_{[b_1} \partial_{[a_1^n|} \cdots \partial_{[a_1^1} A^{(n)}_{a_2^1 \cdots a_{10}^1],\ldots,|a_2^n \cdots a_{10}^n],b_2 b_3 b_4]} = 0, \tag{216}$$

$$\eta^{a_1^i b_1} \cdots \eta^{a_4^i b_4} \partial_{[b_1} \partial_{[a_1^n|} \cdots \partial_{[a_1^1} A^{(n)}_{a_2^1 \cdots a_{10}^1],\ldots,|a_2^n \cdots a_{10}^n],b_2 b_3 b_4]} = 0, \tag{217}$$

for $2 \le i < j \le n$. Antisymmetrising $c$ with $a^i[10]$ in (213) for $2 \le i \le n$ leads to

$$\eta^{a_1^1 a_1^j} \cdots \eta^{a_{10}^1 a_{10}^j} \partial_{[b_1} \partial_{[a_1^n|} \cdots \partial_{[a_1^1} A^{(n)}_{a_2^1 \cdots a_{10}^1],\ldots,|a_2^n \cdots a_{10}^n],b_2 b_3 b_4]} = 0, \tag{218}$$

while antisymmetrising $c$ with $b[4]$ leads to

$$\eta^{a_1^1 b_1} \cdots \eta^{a_4^1 b_4} \partial_{[b_1} \partial_{[a_1^n|} \cdots \partial_{[a_1^1} A^{(n)}_{a_2^1 \cdots a_{10}^1],\ldots,|a_2^n \cdots a_{10}^n],b_2 b_3 b_4]} = 0. \tag{219}$$

Thus for $n > 2$ we have shown inductively that the equations of motion of $A^{(n)}_{9^n,3}$ are (216) and (217) for $1 \le i < j \le n$, and that they all follow from the infinite chain of dualities (209).

**Reformulation in terms of zero-forms.** The discussion above is quite cumbersome. Here we will express everything in terms of the zero-forms in the unfolded formalism, and this will once again give us extremely compact forms of the curvature relations and equations of motion. We introduce a zero-form relation between $F^{(n)}_{4,1^n} \in \mathcal{T}(A_3)$ and $C^{(0)}_{10^n,4} \in \mathcal{T}(A^{(n)}_{9^n,3})$ analogous to equations (91) and (192) at lower levels:

$$C^{(0)}_{a^1[10],\ldots,a^n[10],b[4]} = \varepsilon_{a^1[10]}{}^{d_1} \cdots \varepsilon_{a^n[10]}{}^{d_n} F^{(n)}_{b[4],d_1,\ldots,d_n}. \tag{220}$$

This is one of an infinite number of shifted zero-form relations

$$C^{(m)}_{a^1[10],\ldots,a^n[10],b[4],c_1,\ldots,c_m} = \varepsilon_{a^1[10]}{}^{d_1} \cdots \varepsilon_{a^n[10]}{}^{d_n} F^{(n+m)}_{b[4],d_1,\ldots,d_n,c_1,\ldots,c_m}. \tag{221}$$

As a result, if we write down a Taylor expansion for $A^{(n)}_{9^n,3}$ analogous to (208), the coefficients that are usually given in terms of $\{C^{(0)}_{10^n,4}, C^{(1)}_{10^n,4,1}, C^{(2)}_{10^n,4,1,1}, \ldots\}$ will instead be given in terms of $\{F^{(n)}_{4,1^n}, F^{(n+1)}_{4,1^{n+1}}, F^{(n+2)}_{4,1^{n+2}}, \ldots\}$. The first $n$ zero-forms $\{F^{(0)}_4, F^{(1)}_{4,1}, \ldots, F^{(n-1)}_{4,1^{n-1}}\}$ do not appear in the Taylor expansion of $A^{(n)}_{9^n,3}$ around a point in space-time. Therefore, higher dual fields $A^{(n)}_{9^n,3}$ for increasing $n$ describe the original three-form at higher and higher orders, meaning at longer and longer distances. The same is true for the higher dual six-forms $A^{(n)}_{9^n,6}$ and gravitons $h^{(n)}_{9^n,8,1}$. As before, only a finite set of zero-forms is omitted, and integrating the Bianchi identities leads to all the original equations of motion.

Returning to the zero-form relation (220), taking a curl on the $b[4]$ indices gives

$$\partial_{[e|} C^{(0)}_{a^1[10],\ldots,a^n[10],|b_1 \cdots b_4]} = \varepsilon_{a^1[10]}{}^{d_1} \cdots \varepsilon_{a^n[10]}{}^{d_n} \partial_{[e} F^{(n)}_{b_1 \cdots b_4],d_1,\ldots,d_n}, \tag{222}$$

but the zero-form $F^{(n+1)}_{4,1^{n+1}} \in \mathcal{T}(A_3)$ is irreducible, so (222) becomes

$$\partial_{[e|} C^{(0)}_{a^1[10],\ldots,a^n[10],|b_1 \cdots b_4]} = \varepsilon_{a^1[10]}{}^{d_1} \cdots \varepsilon_{a^n[10]}{}^{d_n} F^{(n+1)}_{[b_1 \cdots b_4|,d_1,\ldots,d_n,|e]} = 0. \tag{223}$$

The generalised Poincaré lemma implies that $C^{(0)}_{10^n,4}$ can be expressed as the curvature tensor

$$C^{(0)}{}_{a^1[10],\dots,a^n[10],b[4]} = \partial_{[b_1}\partial_{[a^n_1|}\cdots\partial_{[a^2_1}\partial_{[a^1_1}A^{(n)}{}_{a^1_2\cdots a^1_{10}],a^2_2\cdots a^2_{10}],\dots,|a^n_2\cdots a^n_{10}],b_2 b_3 b_4]}\,,\tag{224}$$

for the $n^{\text{th}}$ higher dual three-form $A^{(n)}_{9^n,3}$. This is precisely what one would find by solving the first $n+1$ unfolded equations, but here we have finished in one step. It is immediate to see that this curvature is invariant under

$$\delta A^{(n)}{}_{a^1[9],\dots,a^n[9],b[3]} = \Big[\partial_{[a^n_1|}\lambda^{(1)}{}_{a^1[9],\dots,a^{n-1}[9],|a^n_2\cdots a^n_9],b[3]} + \partial_{[b_1|}\lambda^{(2)}{}_{a^1[9],\dots,a^n[9],|b_2 b_3]}\Big]_{9^n,3}\,,\tag{225}$$

where $[\cdots]_{9^n,3}$ denotes a projection onto the GL(11) irreducible $\mathbb{Y}[9^n,3]$ tableau.

Working on-shell, the zero-forms $F^{(n)}_{4,1^n}$ are all irreducible Lorentz tensors. The irreducibility properties of $F^{(n)}_{4,1^n}$ are exchanged under equation (220) as follows:

$$\left.\begin{array}{l}(\text{Tr}_{i,j})^{10}(C_{10^n,4}) = 0\,,\\[2pt](\text{Tr}_{i,n+1})^4(C_{10^n,4}) = 0\,,\\[2pt]\sigma_{i,j}(C_{10^n,4}) = 0\,,\\[2pt]\sigma_{i,n+1}(C_{10^n,4}) = 0\end{array}\right\} \iff \left\{\begin{array}{l}\text{Tr}_{i+1,j+1}(F_{4,1^n}) = 0\,,\\[2pt]\sigma_{1,i+1}(F_{4,1^n}) = 0\,,\\[2pt]\sigma_{i+1,j+1}(F_{4,1^n}) = 0\,,\\[2pt]\text{Tr}_{1,i+1}(F_{4,1^n}) = 0\,.\end{array}\right.\tag{226}$$

where $1 \leqslant i < j \leqslant n$. The primary zero-form $C_{10^n,4}$ now obeys the higher trace constraints

$$(\text{Tr}_{i,j})^{10}(C_{10^n,4}) = 0\,, \qquad (\text{Tr}_{i,n+1})^4(C_{10^n,4}) = 0\,, \qquad 1 \leq i < j \leq n\,.\tag{227}$$

Thus the irreducibility properties of $F^{(n)}_{4,1^n} \in \mathcal{T}(A_3)$ led to an extremely compact form (226) of the linearised equations of motion (216) and (217), where $C_{10^n,4}$ is the curvature (224).

The zero-form relations (220) for adjacent values of $n$ imply a new relation between the primary zero-forms $C^{(0)}_{10^n,4} \in \mathcal{T}(A^{(n)}_{9^n,3})$ and $C^{(1)}_{10^{n-1},4,1} \in \mathcal{T}(A^{(n-1)}_{9^{n-1},3})$:

$$C^{(0)}{}_{a^1[10],a^2[10],\dots,a^n[10],b[4]} = \varepsilon_{a^1[10]}{}^d\, C^{(1)}{}_{a^2[10],\dots,a^n[10],b[4],d}\,.\tag{228}$$

When these zero-forms are expressed in terms of the original fields using (224), we find that (228) reproduces the curvature relation (213). Under (228), the zero-form $C^{(0)}_{10^n,4}$ inherits from $C^{(1)}_{10^{n-1},4,1}$ the constraints

$$(\text{Tr}_{i,j})^{10}(C_{10^n,4}) = (\text{Tr}_{i,n+1})^4(C_{10^n,4}) = \sigma_{i,j}(C_{10^n,4}) = \sigma_{i,n+1}(C_{10^n,4}) = 0\,,\tag{229}$$

for $2 \leq i < j \leq n$. The remaining constraints are exchanged as

$$\left.\begin{array}{l}(\text{Tr}_{1,i})^{10}(C_{10^n,4}) = 0\,,\\[2pt](\text{Tr}_{1,n+1})^4(C_{10^n,4}) = 0\,,\\[2pt]\sigma_{1,i}(C_{10^n,4}) = 0\,,\\[2pt]\sigma_{1,n+1}(C_{10^n,4}) = 0\end{array}\right\} \iff \left\{\begin{array}{l}\sigma_{i-1,n+1}(C_{10^{n-1},4,1}) = 0\,,\\[2pt]\sigma_{n,n+1}(C_{10^{n-1},4,1}) = 0\,,\\[2pt]\text{Tr}_{i-1,n+1}(C_{10^{n-1},4,1}) = 0\,,\\[2pt]\text{Tr}_{n,n+1}(C_{10^{n-1},4,1}) = 0\,.\end{array}\right.\tag{230}$$

**Integrating curvature relations.** Working backwards from the higher curvature relations (213), we can integrate $a^2[10]$, introduce an arbitrary tensor $\Xi_{8|10,9^{n-2},3}$ and impose the shift

$$\Xi_{a^2[8]|a^1[10],a^3[9],\dots,a^n[9],b[3]} \longmapsto \Xi_{a^2[8]|a^1[10],a^3[9],\dots,a^n[9],b[3]} + 9\,\varepsilon_{a^1[10]}{}^c A^{(n-1)}_{a^2[8]c,a^3[9],\dots,a^n[9],b[3]}\,,\tag{231}$$

to obtain

$$\partial_{[b}\partial_{[a^n|}\cdots\partial_{[a^3|}\partial_{[a^1}A^{(n)}_{a^1[9]],a^2[9],|a^3[9]],\dots,|a^n[9]],b[3]]} + \partial_{[b}\partial_{[a^n|}\cdots\partial_{[a^3|}\partial_{[a^2}\Xi_{a^2[8]]|a^1[10],|a^3[9]],\dots,|a^n[9]],b[3]]}$$
$$\propto \varepsilon_{a^1[10]}{}^c\Big(10\,\partial_{[b}\partial_{[a^n|}\cdots\partial_{[a^3}\partial_{[c}A^{(n-1)}_{a^2[9]],a^3[9]],\dots,|a^n[9]],b[3]]}\Big)\,.\tag{232}$$

Integrating $a^3[10]$, we introduce an arbitrary tensor $\Theta_{8|10,9^{n-3},3}$ on the right-hand side:

$$\partial_{[b}\partial_{[a^n|}\cdots\partial_{[a^4|}\partial_{[a^1}A^{(n)}_{a^1[9]],a^2[9],a^3[9],|a^4[9]],...,|a^n[9]],b[3]]}$$
$$+\partial_{[b}\partial_{[a^n|}\cdots\partial_{[a^4|}\partial_{[a^2}\Xi_{a^2[8]]|a^1[10],a^3[9],|a^4[9]],...,|a^n[9]],b[3]]}$$
$$\propto\varepsilon_{a^1[10]}{}^c\Big(10\,\partial_{[b}\partial_{[a^n|}\cdots\partial_{[a^4|}\partial_{[c}A^{(n-1)}_{a^2[9]],a^3[9],|a^4[9]],...,|a^n[9]],b[3]]}$$
$$+\partial_{[b}\partial_{[a^n|}\cdots\partial_{[a^4|}\partial_{[a^3}\Theta_{a^3[8]]|ca^2[9],|a^4[9]],...,|a^n[9]],b[3]]}\Big).\quad(233)$$

The reducible tensor $\Xi_{8|10,9^{n-2},3}$ contains the extra fields

$$\{\widehat{A}_{10,9^{n-2},8,3},\widehat{A}_{10,9^{n-1},2},\widehat{A}_{11,9^{n-2},7,3},\widehat{A}_{11,9^{n-2},8,2},\widehat{A}_{11,9^{n-1},1}\},\quad(234)$$

that are associated with the $A^{(n)}_{9^n,3}$ field, while $\Theta_{8|10,9^{n-3},3}$ contains those that are associated with the $A^{(n-1)}_{9^{n-1},3}$ field.

Integrating the $a^4[10],\ldots,a^n[10]$ columns produces a sequence of tensors, each of which is absorbed into the previous one since we can swap all these columns with each other and also with $a^2[10]$ and $a^3[10]$. The result of this repeated integration is

$$\partial_{[b|}\partial_{[a^1}A^{(n)}_{a^1[9]],a^2[9],...,a^n[9],|b[3]]}+\partial_{[b}\partial_{[a^2}\Xi_{a^2[8]]|a^1[10],a^3[9],...,a^n[9],|b[3]]}$$
$$\propto\varepsilon_{a^1[10]}{}^c\Big(10\,\partial_{[b|}\partial_{[c}A^{(n-1)}_{a^2[9]],a^3[9],...,a^n[9],|b[3]]}+\partial_{[b|}\partial_{[a^3}\Theta_{a^3[8]]|ca^2[9],a^4[9],...,a^n[9],|b[3]]}\Big).\quad(235)$$

Integrating one final time and introducing an arbitrary $\Upsilon_{2|10,9^{n-1}}$ tensor, we obtain

$$\partial_{[a^1}A^{(n)}_{a^1[9]],a^2[9],...,a^n[9],b[3]}+\partial_{[a^2}\Xi_{a^2[8]]|a^1[10],a^3[9],...,a^n[9],b[3]}+\partial_{[b}\Upsilon_{b[2]]|a^1[10],a^2[9],...,...,a^n[9]}$$
$$\propto\varepsilon_{a^1[10]}{}^c\Big(10\,\partial_{[c}A^{(n-1)}_{a^2[9]],a^3[9],...,a^n[9],b[3]}+\partial_{[a^3}\Theta_{a^3[8]]|ca^2[9],a^4[9],...,a^n[9],b[3]}\Big).\quad(236)$$

These duality relations would have been equivalence equations in the $E_{11}$ non-linear realisation meaning that they would only hold up to certain pure gauge terms. By integrating the equations of motion, we have found relations that hold exactly when the gauge parameters are subject to certain constraints. The gauge freedom is absorbed by extra fields. This is an elaboration of (both the computation and the result of) the duality relations in equation (3.5.14) of reference [23], but now the extra fields $\Theta_{8|10,9^{n-3},3}$ are explicit. Every term in (236) needs to be projected onto the GL(11)-irreducible representation associated with the $\mathbb{Y}[10,9,\ldots,9,3]$ diagram.

The first-order connections $\omega^{(n)}{}_{[9]}{}^{10,9^{n-2},3}$ and $\omega^{(n-1)}{}_{[9]}{}^{10,9^{n-3},3}$ in the duality relations (209) are variables that come from the unfolded formalism. Notice that $\Xi_{8|10,9^{n-2},3}$ in (236) has the same structure as the gauge parameter $\alpha_{[8]}{}^{10,9^{n-2},3}$ in (203a). Some components of $\Xi_{8|10,9^{n-2},3}$ are identified with the extra fields in $e_{[9]}{}^{9^{n-1},3}$ and the others are identified with the components of $\alpha_{[8]}{}^{10,9^{n-2},3}$ that can be shifted away with the gauge-for-gauge parameter $\beta_{[7]}{}^{10^2,9^{n-3},2}$. Some components of $\beta_{[7]}{}^{10^2,9^{n-3},2}$ may subsequently be shifted away using a gauge-for-gauge-for-gauge parameter $\gamma_{[6]}{}^{10^3,9^{n-4},3}$, and so on. Looking back at (236), it is unclear where (or if) $\Upsilon_{2|10,9^{n-2}}$ originates in the unfolded formalism, so it is not included in the duality relation (209).

In summary, equations (182) and (209) extend the set of duality relations (146) to include the entire three-form sector.

$$
\begin{array}{ccccccccc}
F_4 & \longleftrightarrow & \omega^{(1)}{}_{[3]}{}^{10} & \longleftrightarrow & \omega^{(2)}{}_{[9]}{}^{10,3} & \longleftrightarrow & \omega^{(3)}{}_{[9]}{}^{10,9,3} & \longleftrightarrow & \omega^{(4)}{}_{[9]}{}^{10,9,9,3} & \longleftrightarrow & \cdots \\
\updownarrow & & \updownarrow & & & & & & \\
F_7 & \longleftrightarrow & \omega^{(1)}{}_{[6]}{}^{10} & & & & & &
\end{array}
\quad(237)
$$

**Unfolding higher dual six-forms.** We will now consider the six-form sector of the theory. Our analysis will be similar to that of the three-form sector, so we will not dwell on all details. To unfold the $A^{(n)}_{9^n,6}$ field in $E_{11}$ at level $3n + 2$, we introduce the following variables:

$$e_{[9]}{}^{9^{n-1},6}, \quad \omega_{[9]}{}^{10,9^{n-2},6}, \quad X_{[9]}{}^{10^2,9^{n-3},6}, \quad \ldots, \quad X_{[9]}{}^{10^{n-1},6}, \quad X_{[6]}{}^{10^n}, \quad C^{10^n,7}, \quad \ldots \quad (238)$$

Schematically, the first two unfolded equations are

$$\mathrm{d}e_{[9]}{}^{9^{n-1},6} + h_1 \, \omega_{[9]}{}^{10,9^{n-2},6} = 0, \tag{239a}$$

$$\mathrm{d}\omega_{[9]}{}^{10,9^{n-2},6} + h_1 X_{[9]}{}^{10^2,9^{n-3},6} = 0, \tag{239b}$$

and they are invariant under the gauge symmetries

$$\delta e_{[9]}{}^{9^{n-1},6} = \mathrm{d}\lambda_{[8]}{}^{9^{n-1},6} + h_1 \, \alpha_{[8]}{}^{10,9^{n-2},6}, \tag{240a}$$

$$\delta \omega_{[9]}{}^{10,9^{n-2},6} = \mathrm{d}\alpha_{[8]}{}^{10,9^{n-2},6} + h_1 \, \beta_{[8]}{}^{10^2,9^{n-3},6}, \tag{240b}$$

$$\delta X_{[9]}{}^{10^2,9^{n-3},6} = \mathrm{d}\beta_{[8]}{}^{10^2,9^{n-3},6}. \tag{240c}$$

The primary zero-form $C_{10^n,7}$ is the first in the tower

$$\mathcal{T}(A^{(n)}_{9^n,6}) = \left\{ C^{(m)}_{10^n,7,1^m} \mid m \in \mathbb{N} \right\} = \left\{ C^{(0)}_{10^n,7}, C^{(1)}_{10^n,7,1}, C^{(2)}_{10^n,7,1,1}, \ldots \right\}. \tag{241}$$

**Duality relation between $A^{(2)}_{9,9,6}$ and $A^{(1)}_{9,6}$.** Before proceeding to arbitrarily high levels, it is useful to consider the first-order duality relation between the first and second higher dual six-forms in terms of their first-order connections:

$$\omega^{(2)}_{a_1 \cdots a_9 | b_1 \cdots b_{10}, c_1 \cdots c_6} \propto \varepsilon_{b_1 \cdots b_{10}}{}^d \, \omega^{(1)}_{c_1 \cdots c_6 | d a_1 \cdots a_9}. \tag{242}$$

Requiring this to be gauge-invariant leads to a gauge parameter relation analogous to (183). The constraint associated with the previous duality relation (136) told us that the parameter $\alpha_{[5]}{}^{10}$ is pure gauge-for-gauge, leading to the next constraint associated with (242):

$$\partial_{[a_1} \alpha_{a_2 \cdots a_9] | b_1 \cdots b_{10}, c_1 \cdots c_6} + \beta_{[a_1 \cdots a_5 || b_1 \cdots b_{10}, | a_6 \cdots a_9] c_1 \cdots c_6} = 0. \tag{243}$$

This gauge parameter constraint allows the extra fields to appear explicitly in (242).

Taking derivatives of the duality relation (242) leads to

$$\partial_{[c_1} \partial_{[b_1} \partial_{[a_1} A^{(2)}_{a_2 \cdots a_{10}], b_2 \cdots b_{10}], c_2 \cdots c_7]} \propto \varepsilon_{a_1 \cdots a_{10}}{}^d \partial_d \partial_{[c_1} \partial_{[b_1} A^{(1)}_{b_2 \cdots b_{10}], c_2 \cdots c_7]}, \tag{244}$$

and taking appropriate traces leads either to the equations of motion for $A^{(1)}_{9,6}$ that were found to be (138), or to the following equations of motion for the $A^{(2)}_{9,9,6}$ field:

$$\partial_{[b_1} \partial^{[a_1} \partial_{[a_1} A^{(2)}_{a_2 \cdots a_{10}],}{}^{a_2 \cdots a_{10}],}{}_{b_2 \cdots b_7]} = 0, \tag{245}$$

$$\partial^{[c_1} \partial_{[b_1} \partial_{[a_1} A^{(2)}_{a_2 \cdots a_{10}], b_2 b_3 c_1 \cdots c_7],}{}^{c_2 \cdots c_7]} = 0. \tag{246}$$

We can also reformulate these equations in terms of the zero-forms in the unfolded formalism. The techniques used to do this for the three-form sector show that $C^{(0)}_{10,10,7}$ can be expressed as the curvature of $A^{(2)}_{9,9,6}$. The curvature relation (244) then becomes

$$C^{(0)}_{a_1 \cdots a_{10}, b_1 \cdots b_{10}, c_1 \cdots c_7} = \varepsilon_{a_1 \cdots a_{10}}{}^d C^{(1)}_{b_1 \cdots b_{10}, c_1 \cdots c_7, d}, \tag{247}$$

and the higher trace constraints

$$(\text{Tr}_{1,2})^{10}(C_{10,10,7}) = 0, \qquad (\text{Tr}_{1,3})^{4}(C_{10,10,7}) = 0, \qquad (\text{Tr}_{2,3})^{4}(C_{10,10,7}) = 0, \tag{248}$$

are equivalent to the linearised equations of motion for $A^{(2)}_{9,9,6}$.

As always, we can integrate up the equations of motion to obtain first-order relations with extra fields appearing explicitly. Writing (247) in terms of the gauge potentials, integrating this equation twice leads to

$$\partial_{[a_1} A^{(2)}_{a_2\cdots a_{10}],b_1\cdots b_9,c_1\cdots c_6} + \partial_{[b_1} \Xi_{b_2\cdots b_9]|a_1\cdots a_{10},c_1\cdots c_6}$$
$$\propto \varepsilon_{a_1\cdots a_{10}}{}^{d}\left(10\,\partial_{[d} A^{(1)}_{b_1\cdots b_9],c_1\cdots c_6} + \partial_{[c_1} \Theta_{c_2\cdots c_6]|d\,b_1\cdots b_9}\right). \tag{249}$$

The irreducible components of $\Xi_{8|10,6}$ and $\Theta_{5|10}$ include the extra fields associated with $A^{(2)}_{9,9,6}$ and $A^{(1)}_{9,6}$, respectively.

**Duality relations for higher dual six-forms.**   In exactly the same way that we were led to (209) in the three-form sector, here we propose first-order on-shell duality relations between adjacent higher dual six-form fields $A^{(n)}_{9^n,6}$ and $A^{(n-1)}_{9^{n-1},6}$ for $n > 2$:

$$\omega^{(n)}{}_{a[9]|b[10],c[9],d^1[9],\ldots,d^{n-3}[9],e[6]} \propto \varepsilon_{b[10]}{}^{p}\,\omega^{(n-1)}{}_{a[9]|pc[9],d^1[9],\ldots,d^{n-3}[9],e[6]}. \tag{250}$$

The first-order connections $\{\omega^{(n)}{}_{[9]}{}^{10,9^{n-2},6}\}$ are the unfolded variables that appear in (239a). For these duality relations to be gauge-invariant, we need to impose the constraints

$$\partial_{[a_1} \alpha_{a_2\cdots a_9]|b[10],c[9],d^1[9],\ldots,d^{n-3}[9],e[6]} + \beta_{[a_1\cdots a_8||b[10],|a_9]c[9],d^1[9],\ldots,d^{n-3}[9],e[6]} = 0. \tag{251}$$

These are essentially the same as the constraints for the three-form sector.

Taking derivatives leads to the gauge-invariant relations

$$\partial_{[b_1} \partial_{[a_1^n|} \cdots \partial_{[a_1^1} A^{(n)}_{a_2^1\cdots a_{10}^1],\ldots,|a_2^n\cdots a_{10}^n],b_2\cdots b_7]} \propto \varepsilon_{a^1[10]}{}^{c}\partial_c\partial_{[b_1}\partial_{[a_1^n|}\cdots\partial_{[a_1^2} A^{(n-1)}_{a_2^2\cdots a_{10}^2],\ldots,|a_2^n\cdots a_{10}^n],b_2\cdots b_7]}, \tag{252}$$

and taking appropriate traces leads to the equations of motion for the $A^{(n)}_{9^n,6}$ field

$$\eta^{a_1^i a_1^j} \cdots \eta^{a_{10}^i a_{10}^j} \partial_{[b_1}\partial_{[a_1^n|}\cdots\partial_{[a_1^1} A^{(n)}_{a_2^1\cdots a_{10}^1],\ldots,|a_2^n\cdots a_{10}^n],b_2\cdots b_7]} = 0, \tag{253}$$

$$\eta^{a_1^i b_1} \cdots \eta^{a_7^i b_7} \partial_{[b_1}\partial_{[a_1^n|}\cdots\partial_{[a_1^1} A^{(n)}_{a_2^1\cdots a_{10}^1],\ldots,|a_2^n\cdots a_{10}^n],b_2\cdots b_7]} = 0, \tag{254}$$

for $1 \le i < j \le n$.

**Reformulation in terms of zero-forms.**   As we did in the three-form sector, we relate the primary zero-form $C^{(0)}_{10^n,4} \in \mathcal{T}(A_{9^n,3})$ to the zero-form $F^{(n)}_{7,1^n} \in \mathcal{T}(A_6)$ through the relation

$$C^{(0)}{}_{a^1[10],\ldots,a^n[10],b[7]} = \varepsilon_{a^1[10]}{}^{d_1} \cdots \varepsilon_{a^n[10]}{}^{d_n} F^{(n)}{}_{b[7],d_1,\ldots,d_n}, \tag{255}$$

which mirrors (220). This is one of an infinite number of shifted zero-form relations

$$C^{(m)}{}_{a^1[10],\ldots,a^n[10],b[7],c_1,\ldots,c_m} = \varepsilon_{a^1[10]}{}^{d_1} \cdots \varepsilon_{a^n[10]}{}^{d_n} F^{(n+m)}{}_{b[7],d_1,\ldots,d_n,c_1,\ldots,c_m}. \tag{256}$$

Taking a curl of (255) on the $b[7]$ indices, using Lorentz irreducibility of $F^{(n+1)}_{7,1^{n+1}} \in \mathcal{T}(A_6)$, and applying the generalised Poincaré lemma, we find that $C^{(0)}_{10^n,4}$ is the curvature tensor

$$C^{(0)}{}_{a^1[10],\ldots,a^n[10],b[7]} = \partial_{[b_1}\partial_{[a_1^n|}\cdots\partial_{[a_1^2}\partial_{[a_1^1} A^{(n)}_{a_2^1\cdots a_{10}^1],a_2^2\cdots a_{10}^2],\ldots,|a_2^n\cdots a_{10}^n],b_2\cdots b_7]}, \tag{257}$$

for the $n^{\text{th}}$ higher dual six-form $A^{(n)}_{9^n,6}$. It is immediate to see that this is invariant under

$$\delta A^{(n)}{}_{a^1[9],\dots,a^n[9],b[6]} = \Big[\partial_{[a^n_1|}\lambda^{(1)}{}_{a^1[9],\dots,a^{n-1}[9],|a^n_2\cdots a^n_9],b[6]} + \partial_{[b_1|}\lambda^{(2)}{}_{a^1[9],\dots,a^n[9],|b_2\cdots b_6]}\Big]_{9^n,6}, \quad (258)$$

where $[\cdots]_{9^n,6}$ denotes a projection onto the GL(11) irreducible $\mathbb{Y}[9^n,6]$ tableau.

Working on-shell, the zero-forms $F^{(n)}_{7,1^n}$ are all irreducible Lorentz tensors. The properties of $F^{(n)}_{7,1^n}$ are exchanged with constraints on $C^{(0)}_{10^n,7}$ under (255) as in (226) where all the fours are replaced by sevens. Therefore, $C_{10^n,7}$ obeys higher trace constraints

$$(\mathrm{Tr}_{i,j})^{10}(C_{10^n,7}) = 0, \qquad (\mathrm{Tr}_{i,n+1})^7(C_{10^n,7}) = 0, \qquad 1 \le i < j \le n, \quad (259)$$

which are equivalent to the linearised equations of motion for all higher $A^{(n)}_{9^n,6}$ fields.

Considering equation (255) for adjacent values of $n$, we find a zero-form relation between $C^{(0)}_{10^n,7} \in \mathcal{T}(A^{(n)}_{9^n,6})$ and $C^{(1)}_{10^{n-1},7,1} \in \mathcal{T}(A^{(n-1)}_{9^{n-1},6})$ which takes the form

$$C^{(0)}{}_{a^1[10],a^2[10],\dots,a^n[10],b[7]} = \varepsilon_{a^1[10]}{}^d C^{(1)}{}_{a^2[10],\dots,a^n[10],b[7],d}. \quad (260)$$

Under this relation, $C^{(0)}_{10^n,7}$ inherits from $C^{(1)}_{10^{n-1},7,1}$ the constraints

$$(\mathrm{Tr}_{i,j})^{10}(C_{10^n,7}) = (\mathrm{Tr}_{i,n+1})^7(C_{10^n,7}) = \sigma_{i,j}(C_{10^n,7}) = \sigma_{i,n+1}(C_{10^n,7}) = 0, \quad (261)$$

for $2 \le i < j \le n$, and the remaining constraints are exchanged as in (230) where all the fours are once again replaced by sevens.

**Integrating curvature relations.** Repeated integration of the curvature relation (252) leads to a first-order on-shell duality relation

$$\partial_{[a^1}A^{(n)}_{a^1[9]],a^2[9],\dots,a^n[9],b[6]} + \partial_{[a^2}\Xi_{a^2[8]|]a^1[10],a^3[9],\dots,a^n[9],b[6]} + \partial_{[b}\Upsilon_{b[5]]|a^1[10],a^2[9],\dots,\dots,a^n[9]}$$
$$\propto \varepsilon_{a^1[10]}{}^c\Big(10\,\partial_{[c}A^{(n-1)}_{a^2[9]],a^3[9],\dots,a^n[9],|b[6]} + \partial_{[a^3}\Theta_{a^3[8]]|ca^2[9],a^4[9],\dots,a^n[9],b[6]}\Big), \quad (262)$$

featuring arbitrary tensors $\Xi_{8|10,9^{n-2},6}$, $\Theta_{8|10,9^{n-3},6}$, and $\Upsilon_{5|10,9^{n-1}}$. Note that (262) must be projected onto the GL(11) irreducible $\mathbb{Y}[9,\dots,9,6]$ tableau. The tensors $\Xi$ and $\Theta$ are clearly identified with extra fields since they have the same symmetry types as the $\alpha$ parameters in the unfolded equations, but once again it is not known if the irreducible fields in $\Upsilon_{5|10,9^n}$ originate from the unfolded equations and gauge symmetries.

**Duality relations between $A^{(n)}_{9^n,3}$ and $A^{(n)}_{9^n,6}$.** It is now straightforward to obtain relations between the $n^{\text{th}}$ higher dual three-form and six-form fields for $n \ge 2$. They constitute the rungs of the ladder in the diagram below.

$$\begin{array}{ccccc} \cdots & \longleftrightarrow & \omega^{(n)}{}_{[9]}{}^{10,9^{n-2},3} & \longleftrightarrow & \cdots \\ & & \Big\updownarrow & & \\ \cdots & \longleftrightarrow & \omega^{(n)}{}_{[9]}{}^{10,9^{n-2},6} & \longleftrightarrow & \cdots \end{array} \quad (263)$$

These duality relations take the form

$$\widetilde{\omega}^{(n)}{}_{[a_1|a_2a_3a_4]} \propto \varepsilon_{a_1\cdots a_4}{}^{b_1\cdots b_7}\widetilde{\omega}^{(n)}{}_{[b_1|b_2\cdots b_7]}, \quad (264)$$

where the definitions of $\widetilde{\omega}_{1|3}^{(n)}$ and $\widetilde{\omega}_{1|6}^{(n)}$ depend on the parity of $n$:

$$\widetilde{\omega}^{(2m)}{}_{a|b[k]} := \omega^{(2m)}{}_{d[9]|c[9]a,}{}^{c[9],d[9],}{}_{e^1[9],}{}^{e^1[9],}{}_{\dots,e^{m-2}[9],}{}^{e^{m-2}[9],}{}_{b[k]}, \tag{265}$$

$$\widetilde{\omega}^{(2m-1)}{}_{a|b[k]} := \varepsilon^{e[10]}{}_a \, \omega^{(2m-1)}{}_{c[9]|e[10],}{}^{c[9],}{}_{d^1[9],}{}^{d^1[9],}{}_{\dots,d^{m-2}[9],}{}^{d^{m-2}[9],}{}_{b[k]}. \tag{266}$$

Setting $k = 3$ or $k = 6$ gives the appropriate definition for each sector.

We now have an infinite ladder of first-order on-shell duality relations. One of the rails of the ladder is populated by higher duality relations for the three-form sector: (114), (182) and (209). The other rail of the ladder is populated by those of the six-form sector: (136), (242) and (250). Lastly, the rungs of the ladder are populated by electromagnetic dualities: (68), (146), and (264). This is summarised as follows:

$$
\begin{array}{ccccccccccc}
F_4 & \longleftrightarrow & \omega^{(1)}{}_{[3]}{}^{10} & \longleftrightarrow & \omega^{(2)}{}_{[9]}{}^{10,3} & \longleftrightarrow & \omega^{(3)}{}_{[9]}{}^{10,9,3} & \longleftrightarrow & \omega^{(4)}{}_{[9]}{}^{10,9,9,3} & \longleftrightarrow & \cdots \\[4mm]
\updownarrow & & \updownarrow & & \updownarrow & & \updownarrow & & \updownarrow & & \\[4mm]
F_7 & \longleftrightarrow & \omega^{(1)}{}_{[6]}{}^{10} & \longleftrightarrow & \omega^{(2)}{}_{[9]}{}^{10,6} & \longleftrightarrow & \omega^{(3)}{}_{[9]}{}^{10,9,6} & \longleftrightarrow & \omega^{(4)}{}_{[9]}{}^{10,9,9,6} & \longleftrightarrow & \cdots
\end{array}
\tag{267}
$$

**Unfolding higher dual gravitons.** Lastly, we will sketch the unfolding of the gravitational sector of the theory at all levels. In order to unfold the second higher dual graviton $h_{9,9,8,1}^{(2)}$ in $E_{11}$ at level nine, we introduce a tower of variables

$$e_{[9]}{}^{9,8,1}, \quad \omega_{[9]}{}^{10,8,1}, \quad X_{[8]}{}^{10,10,1}, \quad Y_{[1]}{}^{10,10,9}, \quad C^{10,10,9,2}, \quad \dots \tag{268}$$

The first four unfolded equations are

$$\mathrm{d}e_{[9]}{}^{a[9],b[8],c} + h_d \, \omega_{[9]}{}^{d\langle a[9],b[8],c\rangle} = 0, \tag{269a}$$

$$\mathrm{d}\omega_{[9]}{}^{a[10],b[8],c} + h_{d[2]} X_{[8]}{}^{a[10],d[2]\langle b[8],c\rangle} = 0, \tag{269b}$$

$$\mathrm{d}X_{[8]}{}^{a[10],b[10],c} + h_{d[8]} Y_{[1]}{}^{a[10],b[10],d[8]c} = 0, \tag{269c}$$

$$\mathrm{d}Y_{[1]}{}^{a[10],b[10],c[9]} + h_{d[2]} C^{a[10],b[10],c[9],d[2]} = 0, \tag{269d}$$

where angled brackets denote projection onto the obvious irreducible Young tableaux, and these equations are invariant under the gauge symmetries

$$\delta e_{[9]}{}^{a[9],b[8],c} = \mathrm{d}\lambda_{[8]}{}^{a[9],b[8],c} + h_d \, \alpha_{[8]}{}^{d\langle a[9],b[8],c\rangle}, \tag{270a}$$

$$\delta \omega_{[9]}{}^{a[10],b[8],c} = \mathrm{d}\alpha_{[8]}{}^{a[10],b[8],c} - h_{d[2]} \beta_{[7]}{}^{a[10],d[2]\langle b[8],c\rangle}, \tag{270b}$$

$$\delta X_{[8]}{}^{a[10],b[10],c} = \mathrm{d}\beta_{[7]}{}^{a[10],b[10],c} - h_{d[8]} \gamma^{a[10],b[10],d[8]c}, \tag{270c}$$

$$\delta Y_{[1]}{}^{a[10],b[10],c[9]} = \mathrm{d}\gamma^{a[10],b[10],c[9]}. \tag{270d}$$

Moreover, in order to unfold the $n^{\text{th}}$ higher dual graviton $h_{9^n,8,1}^{(n)}$ in $E_{11}$ at level $3n + 3$, where $n \geq 3$, we introduce a tower of variables beginning with

$$e_{[9]}{}^{9^{n-1},8,1}, \quad \omega_{[9]}{}^{10,9^{n-2},8,1}, \quad X_{[9]}{}^{10^2,9^{n-3},8,1}, \quad X_{[9]}{}^{10^3,9^{n-4},8,1}, \quad \dots \tag{271}$$

Schematically, the first two unfolded equations are

$$\mathrm{d}e_{[9]}{}^{9^{n-1},8,1} + h_1 \, \omega_{[9]}{}^{10,9^{n-2},8,1} = 0, \tag{272a}$$

$$\mathrm{d}\omega_{[9]}{}^{10,9^{n-2},8,1} + h_1 X_{[9]}{}^{10^2,9^{n-3},8,1} = 0, \tag{272b}$$

and they are invariant under

$$\delta e_{[9]}{}^{9^{n-1},8,1} = \mathrm{d}\lambda_{[8]}{}^{9^{n-1},8,1} + h_1\,\alpha_{[8]}{}^{10,9^{n-2},8,1}\,, \tag{273a}$$

$$\delta \omega_{[9]}{}^{10,9^{n-2},8,1} = \mathrm{d}\alpha_{[8]}{}^{10,9^{n-2},8,1} + h_1\,\beta_{[8]}{}^{10^2,9^{n-3},8,1}\,, \tag{273b}$$

$$\delta X_{[9]}{}^{10^2,9^{n-3},8,1} = \mathrm{d}\beta_{[8]}{}^{10^2,9^{n-3},8,1}\,. \tag{273c}$$

The tower (271) continues as

$$\ldots,\quad X_{[9]}{}^{10^{n-2},9,8,1},\quad X_{[9]}{}^{10^{n-1},8,1},\quad X_{[8]}{}^{10^n,1},\quad X_{[1]}{}^{10^n,9},\quad C^{10^n,9,2},\quad \ldots,\tag{274}$$

where the primary zero-form $C_{10^n,9,2}$ is the first in the tower

$$\mathcal{T}(h^{(n)}_{9^n,8,1}) = \{C^{(m)}_{10^n,9,2,1^m} \mid m \in \mathbb{N}\} = \{C^{(0)}_{10^n,9,2}, C^{(1)}_{10^n,9,2,1}, C^{(2)}_{10^n,9,2,1,1}, \ldots\}\,. \tag{275}$$

In order not to repeat the details of the three-form and six-form sectors, we simply state that the primary zero-form $C^{(0)}_{10^n,9,2}$ can be expressed as the curvature tensor

$$C^{(0)}{}_{a^1[10],\ldots,a^n[10],b[9],c[2]} = \partial_{[c_1}\partial_{[b_1}\partial_{[a_1^n|}\cdots\partial_{[a_1^2}\partial_{[a_1^1}h^{(n)}{}_{a_2^1\cdots a_{10}^1],a_2^2\cdots a_{10}^2],\ldots,|a_2^n\cdots a_{10}^n],b_2\cdots b_9],c_2]}\,, \tag{276}$$

for the higher dual graviton $h^{(n)}_{9^n,8,1}$. This can be shown either by solving the unfolded equations for $C^{(0)}_{10^n,9,2}$ in terms of $h^{(n)}_{9^n,8,1}$ or by using the generalised Poincaré lemma. It is immediate to see that this curvature is invariant under

$$\delta h^{(n)}{}_{a^1[9],\ldots,a^n[9],b[8],c} = \left[\begin{array}{c} \partial_{[a_1^n|}\lambda^{(1)}_{a^1[9],\ldots,a^{n-1}[9],|a_2^n\ldots a_9^n],b[8],c} + \partial_{[b_1}\lambda^{(2)}_{a^1[9],\ldots,a^n[9],|b_2\ldots b_8],c} \\ + \partial_c\,\lambda^{(3)}_{a^1[9],\ldots,a^n[9],b[8]} \end{array}\right]_{9^n,8,1}\,, \tag{277}$$

where $[\cdots]_{9^n,8,1}$ denotes a projection onto the GL(11) irreducible $\mathbb{Y}[9^n,8,1]$ tableau.

**Duality relation between $h^{(2)}_{9,9,8,1}$ and $h^{(1)}_{9,8,1}$.** As before, we will first propose the duality relation between $h^{(1)}_{9,8,1}$ and $h^{(2)}_{9,9,8,1}$ since it has a different form to the duality relations at higher levels. This duality relation takes the form

$$\omega^{(2)}{}_{a_1\cdots a_9|b_1\cdots b_{10},c_1\cdots c_8,d} \propto \varepsilon_{b_1\cdots b_{10}}{}^p\,\omega^{(1)}{}_{c_1\cdots c_8|pa_1\cdots a_9,d}\,. \tag{278}$$

Similar to what we found in the three-form and six-form sectors, this relation is gauge-covariant when the parameters obey the constraint

$$\partial_{[a_1}\alpha_{a_2\cdots a_8]|b_1\cdots b_{10},c_1\cdots c_8,d} - \beta_{[a_1\cdots a_7||b_1\cdots b_{10},|a_8 a_9]c_1\cdots c_8,d} \propto \delta_{a_1\cdots a_9 p,b_1\cdots b_{10}}\partial_d\,\alpha_{c_1\cdots c_8}{}^p\,, \tag{279}$$

where $\delta_{p[n],q[n]}$ denotes $\delta^{q[n]}_{p[n]}$ with all the indices lowered. We have used the previous constraint (156) to obtain the new constraint (279), which does not tell us that $\omega^{(2)}{}_{[9]}{}^{10,8,1}$ needs to be gauge-invariant, but only that its gauge transformation is related to the dual gravity nine-form parameter $\alpha^9$ that we introduced in Section 3.3.

Taking derivatives of the duality relation (278) leads to a gauge-invariant relation

$$\partial_{[d_1}\partial_{[c_1}\partial_{[b_1}\partial_{[a_1}h^{(2)}{}_{a_2\cdots a_{10}],b_2\cdots b_{10}],c_2\cdots c_9],d_2]} \propto \varepsilon_{a_1\cdots a_{10}}{}^e\,\partial_e\partial_{[d_1}\partial_{[c_1}\partial_{[b_1}h^{(1)}{}_{b_2\cdots b_{10}],c_2\cdots c_9],d_2]}\,, \tag{280}$$

and taking appropriate traces leads to the equations of motion for each field. In terms of the curvature tensor $C^{(0)}_{10,10,9,2}$, the equations of motion for $h^{(2)}_{9,9,8,1}$ that follow from (278) are

$$\begin{aligned} (\mathrm{Tr}_{1,2})^{10}(C_{10,10,9,2}) &= 0\,, & \mathrm{Tr}_{3,4}(C_{10,10,9,2}) &= 0\,, \\ (\mathrm{Tr}_{1,3})^9(C_{10,10,9,2}) &= 0\,, & (\mathrm{Tr}_{1,4})^2(C_{10,10,9,2}) &= 0\,, & 1 \le i < j \le n\,. \\ (\mathrm{Tr}_{2,3})^9(C_{10,10,9,2}) &= 0\,, & (\mathrm{Tr}_{2,4})^2(C_{10,10,9,2}) &= 0\,, \end{aligned} \tag{281}$$

Equation (280) can be expressed in terms of $C^{(0)}_{10,10,9,2} \in \mathcal{T}(h^{(2)}_{9,9,8,1})$ and $C^{(1)}_{10,9,2,1} \in \mathcal{T}(h^{(1)}_{9,8,1})$:

$$C^{(0)}{}_{a_1 \cdots a_{10}, b_1 \cdots b_{10}, c_1 \cdots c_9, d_1 d_2} = \varepsilon_{a_1 \cdots a_{10}}{}^e C^{(1)}{}_{b_1 \cdots b_{10}, c_1 \cdots c_9, d_1 d_2, e} \,. \tag{282}$$

Integrating this three times leads to the first-order relation

$$\partial_{[a_1} h^{(2)}{}_{a_2 \cdots a_{10}], b_1 \cdots b_9, c_1 \cdots c_8, d} + \partial_{[b_1} \Xi_{b_2 \cdots b_9]|a_1 \cdots a_{10}, c_1 \cdots c_8, d} + \partial_d \Upsilon_{a_1 \cdots a_{10}, b_1 \cdots b_9, c_1 \cdots c_8}$$
$$\propto \varepsilon_{a_1 \cdots a_{10}}{}^e \left( 10\, \partial_{[e} h^{(1)}{}_{b_1 \cdots b_9], c_1 \cdots c_8, d} + \partial_{[c_1} \Theta_{c_2 \cdots c_8]|e b_1 \cdots b_9, d} \right). \tag{283}$$

The arbitrary tensors $\Xi_{8|10,8,1}$ and $\Theta_{7|10,1}$ have the same structure as $\alpha_{[8]}{}^{10,8,1}$ in (273a) and $\alpha_{[7]}{}^{10,1}$ in (153a), respectively, so their components are interpreted either as the extra fields that are associated with $h^{(2)}_{9,9,8,1}$ and $h^{(1)}_{9,8,1}$ or as the components of $\alpha_{[8]}{}^{10,8,1}$ and $\alpha_{[7]}{}^{10,1}$ that can be shifted away with gauge-for-gauge symmetries. The field $\Upsilon_{10,9,8}$ does not seem to originate from the unfolded equations or gauge symmetries.

**Duality relations for higher dual gravitons.** We now propose first-order on-shell duality relations between $h^{(n)}_{9^n,8,1}$ and $h^{(n-1)}_{9^{n-1},8,1}$ that take the form

$$\omega^{(n)}{}_{a[9]|b[10],c[9],d^1[9],\ldots,d^{n-3}[9],e[8],f} \propto \varepsilon_{b[10]}{}^p \omega^{(n-1)}{}_{a[9]|pc[9],d^1[9],\ldots,d^{n-3}[9],e[8],f} \,. \tag{284}$$

The gauge parameter constraints for these higher duality relations are given by

$$\partial_{[a_1} \alpha_{a_2 \cdots a_9]|b[10],c[9],d^1[9],\ldots,d^{n-3}[9],e[8],f} - \beta_{[a_1 \cdots a_8||b[10],|a_9]c[9],d^1[9],\ldots,d^{n-3}[9],e[8],f}$$
$$\propto \varepsilon_{b[10]}{}^{p_1} \varepsilon_{p_1 c[9]}{}^{p_2} \varepsilon_{p_2 d^1[9]}{}^{p_3} \cdots \varepsilon_{p_{n-2} d^{n-3}[9]}{}^{p_{n-1}} \varepsilon_{p_{n-1} a[9]}{}^{p_n} \partial_f \alpha_{e[8] p_n} \,. \tag{285}$$

In contrast to the three-form and six-form sectors of the theory where gauge-invariance of the duality relations forces the gauge-invariance of all first-order connections $\omega$ for the higher dual fields, here we find that the gauge parameter constraints do not force the first-order connections in the gravity sector to be gauge-invariant, but instead their gauge variations are all related to the dual gravity parameter $\alpha^9$ in Section 3.3.

Taking derivatives leads to

$$\partial_{[c_1} \partial_{[b_1} \partial_{[a_1^n|} \cdots \partial_{[a_1^1} h^{(n)}{}_{a_2^1 \cdots a_{10}^1],\ldots,|a_2^n \cdots a_{10}^n],b_2 \cdots b_9],c_2]}$$
$$\propto \varepsilon_{a^1[10]}{}^d \partial_d \partial_{[c_1} \partial_{[b_1} \partial_{[a_1^n|} \cdots \partial_{[a_1^2} h^{(n-1)}{}_{a_2^2 \cdots a_{10}^2],\ldots,|a_2^n \cdots a_{10}^n],b_2 \cdots b_9],c_2]} \,, \tag{286}$$

and taking traces leads to the equations of motion of each field. Expressing $C^{(0)}_{10^n,9,2} \in \mathcal{T}(h^{(n)}_{9\cdot 8,1})$ and $C^{(1)}_{10^{n-1},9,2,1} \in \mathcal{T}(h^{(n-1)}_{9^{n-1},8,1})$ as curvatures, the equations of motion for $h^{(n)}_{9^n,8,1}$ can be written in the compact form

$$\begin{aligned}
(\mathrm{Tr}_{i,j})^{10}(C_{10^n,9,2}) = 0\,, && (\mathrm{Tr}_{i,n+1})^9(C_{10^n,9,2}) = 0\,, && \\
(\mathrm{Tr}_{i,n+2})^2(C_{10^n,9,2}) = 0\,, && \mathrm{Tr}_{n+1,n+2}(C_{10^n,9,2}) = 0\,, && 1 \le i < j \le n\,.
\end{aligned} \tag{287}$$

Some of these are inherited from the equations of motion of $h^{(n-1)}_{9^{n-1},8,1}$ and others are due to the irreducibility properties of the curvature tensors.

**Reformulation in terms of zero-forms.** The primary zero-form $C^{(n)}_{10^n,9,2} \in \mathcal{T}(h^{(n)}_{9^n,8,1})$ is related to $C^{(n)}_{9,2,1^n} \in \mathcal{T}(h_{8,1})$ through the zero-form relation

$$C^{(0)}{}_{a^1[10],\ldots,a^n[10],b[9],c[2]} = \varepsilon_{a^1[10]}{}^{d_1} \cdots \varepsilon_{a^n[10]}{}^{d_n} C^{(n)}{}_{b[9],c[2],d_1,\ldots,d_n} \,. \tag{288}$$

This mirrors (220) and (255) in the three-form and six-form sectors, and generalises (161) to higher levels. As before, (288) is one of an infinite number of shifted relations

$$C^{(m)}{}_{a^1[10],\ldots,a^n[10],b[9],c[2],e_1,\ldots,e_m} = \varepsilon_{a^1[10]}{}^{d_1}\cdots\varepsilon_{a^n[10]}{}^{d_n}C^{(n+m)}{}_{b[9],c[2],d_1,\ldots,d_n,e_1,\ldots,e_m}. \tag{289}$$

Working on-shell, $C^{(n)}_{9,2,1^n} \in \mathcal{T}(h_{8,1})$ are all irreducible Lorentz tensors, and their properties are exchanged under (288) with constraints on $C^{(0)}_{10^n,9,2} \in \mathcal{T}(h^{(n)}_{9^n,8,1})$ as follows:

$$\left.\begin{aligned} (\mathrm{Tr}_{i,j})^{10}(C_{10^n,9,2}) &= 0,\\ (\mathrm{Tr}_{i,n+1})^{9}(C_{10^n,9,2}) &= 0,\\ (\mathrm{Tr}_{i,n+2})^{2}(C_{10^n,9,2}) &= 0,\\ \mathrm{Tr}_{n+1,n+2}(C_{10^n,9,2}) &= 0,\\ \sigma_{i,j}(C_{10^n,9,2}) &= 0,\\ \sigma_{i,n+1}(C_{10^n,9,2}) &= 0,\\ \sigma_{i,n+2}(C_{10^n,9,2}) &= 0,\\ \sigma_{n+1,n+2}(C_{10^n,9,2}) &= 0 \end{aligned}\right\} \iff \left\{\begin{aligned} \mathrm{Tr}_{i+2,j+2}(C_{9,2,1^n}) &= 0,\\ \sigma_{1,i+2}(C_{9,2,1^n}) &= 0,\\ \sigma_{2,i+2}(C_{9,2,1^n}) &= 0,\\ \mathrm{Tr}_{1,2}(C_{9,2,1^n}) &= 0,\\ \sigma_{i+2,j+2}(C_{9,2,1^n}) &= 0,\\ \mathrm{Tr}_{1,i+2}(C_{9,2,1^n}) &= 0,\\ \mathrm{Tr}_{2,i+2}(C_{9,2,1^n}) &= 0,\\ \sigma_{1,2}(C_{9,2,1^n}) &= 0, \end{aligned}\right. \tag{290}$$

where $1 \le i < j \le n$. Therefore, $C_{10^n,9,2}$ obeys higher trace constraints (287) which are the linearised equations of motion for the $h^{(n)}_{9^n,8,1}$ field.

Another way to proceed would have been to notice that the zero-form relation (288) for the higher duals $h^{(n)}_{9^n,8,1}$ and $h^{(n-1)}_{9^{n-1},8,1}$ imply a new relation between $C^{(0)}_{10^n,9,2} \in \mathcal{T}(h^{(n)}_{9^n,8,1})$ and $C^{(1)}_{10^{n-1},9,2,1} \in \mathcal{T}(h^{(n-1)}_{9^{n-1},8,1})$ of the form

$$C^{(0)}{}_{a^1[10],a^2[10],\ldots,a^n[10],b[9],c[2]} = \varepsilon_{a^1[10]}{}^{d}C^{(1)}{}_{a^2[10],\ldots,a^n[10],b[9],c[2],d}. \tag{291}$$

The primary zero-form $C^{(0)}_{10^n,9,2}$ inherits from $C^{(1)}_{10^{n-1},9,2,1}$ the constraints

$$(\mathrm{Tr}_{i,j})^{10}(C_{10^n,9,2}) = 0, \qquad \sigma_{i,j}(C_{10^n,9,2}) = 0, \tag{292}$$

$$(\mathrm{Tr}_{i,n+1})^{9}(C_{10^n,9,2}) = 0, \qquad \sigma_{i,n+1}(C_{10^n,9,2}) = 0, \tag{293}$$

$$(\mathrm{Tr}_{i,n+2})^{2}(C_{10^n,9,2}) = 0, \qquad \sigma_{i,n+2}(C_{10^n,9,2}) = 0, \tag{294}$$

$$\mathrm{Tr}_{n+1,n+2}(C_{10^n,9,2}) = 0, \qquad \sigma_{n+1,n+2}(C_{10^n,9,2}) = 0, \tag{295}$$

for $2 \le i < j \le n$, and the remaining constraints are exchanged as follows:

$$\left.\begin{aligned} (\mathrm{Tr}_{1,i})^{10}(C_{10^n,9,2}) &= 0,\\ (\mathrm{Tr}_{1,n+1})^{9}(C_{10^n,9,2}) &= 0,\\ (\mathrm{Tr}_{1,n+2})^{2}(C_{10^n,9,2}) &= 0,\\ \sigma_{1,i}(C_{10^n,9,2}) &= 0,\\ \sigma_{1,n+1}(C_{10^n,9,2}) &= 0,\\ \sigma_{1,n+2}(C_{10^n,9,2}) &= 0 \end{aligned}\right\} \iff \left\{\begin{aligned} \sigma_{i-1,n+2}(C_{10^{n-1},9,2,1}) &= 0,\\ \sigma_{n,n+2}(C_{10^{n-1},9,2,1}) &= 0,\\ \sigma_{n+1,n+2}(C_{10^{n-1},9,2,1}) &= 0,\\ \mathrm{Tr}_{i-1,n+2}(C_{10^{n-1},9,2,1}) &= 0,\\ \mathrm{Tr}_{n,n+2}(C_{10^{n-1},9,2,1}) &= 0,\\ \mathrm{Tr}_{n+1,n+2}(C_{10^{n-1},9,2,1}) &= 0. \end{aligned}\right. \tag{296}$$

**Integrating curvature relations.** Repeatedly integrating equation (286) and applying an appropriate shift leads to a set of first-order duality relations with extra fields made explicit:

$$\partial_{[a^1}h^{(n)}{}_{a^1[9]],a^2[9],\ldots,a^n[9],b[8],c} + \partial_{[a^2}\Xi_{a^2[8]]|a^1[10],a^3[9],\ldots,a^n[9],b[8],c}$$
$$+ \partial_c \Upsilon_{a^1[10],a^2[9],\ldots,\ldots,a^n[9],b[8]} + \partial_{[b}\Pi_{b[7]]|a^1[10],a^2[9],\ldots,\ldots,a^n[9],c}$$
$$\propto \varepsilon_{a^1[10]}{}^{d}\left(10\,\partial_{[d}h^{(n-1)}{}_{a^2[9]],a^3[9],\ldots,a^n[9],|b[6]} + \partial_{[a^3}\Theta_{a^3[8]]|da^2[9],a^4[9],\ldots,a^n[9],b[8],c}\right). \tag{297}$$

Some of the components of the arbitrary tensors $\Xi_{8|10,9^{n-2},8,1}$ and $\Theta_{8|10,9^{n-3},8,1}$ are interpreted as the extra fields associated with $h^{(n)}_{9^n,8,1}$ and $h^{(n-1)}_{9^{n-1},8,1}$, respectively, and the others are interpreted as components of the parameters $\alpha_{[8]}{}^{10,9^{n-2},8,1}$ and $\alpha_{[8]}{}^{10,9^{n-3},8,1}$ in (273a) that can be shifted away with a gauge-for-gauge symmetry. The other arbitrary tensors $\Upsilon_{10,9^{n-1},8}$ and $\Pi_{7|10,9^{n-1},1}$ once again have no obvious origin in the unfolded equations and symmetries, so they are not featured in (284). All these higher duality relations for $n > 2$ are depicted as follows:

$$\cdots \quad \longleftrightarrow \quad \omega^{(n-1)}{}_{[9]}{}^{10,9^n,8,1} \quad \longleftrightarrow \quad \omega^{(n)}{}_{[9]}{}^{10,9^{n+1},8,1} \quad \longleftrightarrow \quad \cdots \tag{298}$$

Finally, the infinite tower of duality relations given by (46), (155), (278), and (284) can be glued together in the same diagram:

$$\omega_{[1]}{}^2 \quad \longleftrightarrow \quad \omega_{[1]}{}^9 \quad \longleftrightarrow \quad \omega^{(1)}{}_{[8]}{}^{10,1} \quad \longleftrightarrow \quad \omega^{(2)}{}_{[9]}{}^{10,8,1} \quad \longleftrightarrow \quad \omega^{(3)}{}_{[9]}{}^{10,9,8,1} \quad \longleftrightarrow \quad \cdots \tag{299}$$

**Summary.**    In this section we have proposed an infinite number of duality relations between all the higher dual fields in the $E_{11}$ non-linear realisation. By taking derivatives and traces we have obtained all their linearised equations of motion. These duality relations and equations of motion match those of the non-linear realisation up to the level where they have been worked out. The presence of extra fields and constrained gauge parameters ensures that these duality relations all hold exactly and not as equivalence relations up to pure gauge terms. Integrating the equations of motion has led to first-order duality relations with extra fields explicit.

Of course, the non-linear realisation contains much more than the higher dual three-forms, six-forms, and gravitons. The first field beyond these three families is the Romans field $B_{10,1,1}$ at level four. There is also a field $B_{10,9,1,1}$ at level seven, and we speculate that this should be interpreted as a higher dual Romans field. Examining $E_{11}$ level-by-level, it seems that every field either (1) belongs to an infinite family of higher dual fields associated with a field at lower levels, or (2) starts a family of its own with all higher dual counterparts appearing at higher levels. It may be possible to derive duality relations analogous to those summarised in (267) and (299) for the Romans field $B_{10,1,1}$ and all other fields in $E_{11}$ with columns of height ten. It is less clear how to construct higher duality relations for fields with columns of height eleven, such as a relation between $C_{11,1}$ at level four and one of the $C_{11,9,1}$ fields at level seven.

## 5.5   Counting extra fields in representations of $E_{11}$

So far we have worked out the unfolded formulation of every dual field in the $E_{11}$ non-linear realisation, i.e. the fields with at most nine antisymmetric indices in each block. In Section 4, we unfolded the $A_{9,3}$ and $B_{10,1,1}$ fields at level four, and we will now briefly sketch the unfolded formulation of the fields with at most ten indices in each block up to level seven. We will find that the fields required to unfold these fields are not all contained in $E_{11}$ itself.

We calculated the linearised equations of motion for all dual fields by taking derivatives and traces of the infinite set of first-order duality relations that we proposed earlier in this section. These equations of motion hold exactly and they are only given in terms of the irreducible $E_{11}$ fields. Hence if one is only concerned with the equations of motion then $E_{11}$ contains all the required fields. The duality relations in the $E_{11}$ non-linear realisation are equivalence relations in the sense that they only hold up to pure gauge terms, and this contrasts with the duality relations that we proposed here in terms of the unfolded variables since these relations all hold exactly. This difference is due to the extra fields appearing in our proposed duality relations. For example, the duality relation (46) between the graviton $h_{1,1}$ and the dual graviton $h_{8,1}$ features an extra two-form $\widehat{A}_2$ and nine-form $\widehat{A}_9$ which soak up the gauge freedom of (51a). We have also integrated up the equations of motion to obtain first-order duality relations which

relate all the higher dual fields up to generic gauge transformation terms. Thus we have found the precise meaning of the equivalence equations in non-linear realisation.

In this section we will catalogue all the extra fields that appear in the unfolded formalism compared with those in the $E_{11}$ non-linear realisation up to level seven. We proceed level by level, listing the extra fields in each case.

Unfolding the graviton at level zero led to an extra two-form field that can be eliminated using the $I_c(E_{11})$ transformation at level zero, i.e. local Lorentz symmetry. At levels one and two we find the three-form and six-form fields, and their unfolded formulations introduce no extra fields. In Section 3.3 we unfolded the dual graviton at level three, and this introduced an extra nine-form field. A field of precisely this type features in the duality relation (46).

In Table 1, we summarise the unfolded spectrum associated with the graviton, three-form, six-form, and dual graviton in the $E_{11}$ non-linear realisation at levels zero, one, two, and three, respectively. The first column contains the first unfolded variable $e_{[p_\alpha]}{}^\alpha$ for each $E_{11}$ field that we unfold, and the second column lists the symmetry types of all their irreducible components. The $E_{11}$ column counts the number of fields of each symmetry type contained inside $E_{11}$ itself. It may be possible for $E_{11}$ fields to play (at least partially) the role of extra fields. The number of fields that we have after unfolding is given in the unfolding column. The net column gives the number of extra fields, i.e. the deficit of $E_{11}$ fields compared with the new unfolded spectrum. In other words, it counts how many more fields there are inside the unfolded spectrum compared with the non-linear realisation. A negative number $-n$ in the net column tells us that we need to add $n$ fields to the non-linear realisation. It might be the case that these extra fields are really just other $E_{11}$ fields, but they also may belong to highest weight representations of $E_{11}$ that need to be added to the theory in a consistent way. The last column describes the content of the $\ell_2$ representation. We note that the $I_c(E_{11})$ symmetry at level zero can be used to shift away the antisymmetric part $\widehat{A}_2$ at level zero. This corresponds to the local transformation of the vielbein. At level three we see that the $\ell_2$ representation begins with a nine-form that matches the symmetry type of the extra field associated with the dual graviton.

**Analysis up to level six.**    In the $E_{11}$ non-linear realisation there are three fields at level four: the higher dual field $A_{9,3}$ which is dual to the three-form at level one, the Romans field $B_{10,1,1}$, and $C_{11,1}$. In Section 4 we found that unfolding $A_{9,3}$ led to a pair of extra fields $\widehat{A}_{10,2}$ and $\widehat{A}_{11,1}$, while unfolding $B_{10,1,1}$ led to one extra field $\widehat{B}_{11,1}$. Thus we find three extra fields beyond the original fields in the non-linear realisation at level four: $\widehat{A}_{10,2}$, $\widehat{A}_{11,1}$, and $\widehat{B}_{11,1}$. Notice that we also have a third field $C_{11,1}$ in the non-linear realisation, and it has the same GL(11) symmetry type as two of the extra fields at this level. It is possible that $C_{11,1}$ plays a role in unfolding the other two fields $A_{9,3}$ and $B_{10,1,1}$ at level four, and to see if this is true one would need to compute the non-linear realisation up to level four and see how $C_{11,1}$ occurs.

At level five there are four fields the non-linear realisation: $A_{9,6}$, $B_{10,4,1}$, $C_{11,3,1}$, and $C_{11,4}$. In Section 5.1 we found that the higher dual six-form $A_{9,6}$ is accompanied by $\widehat{A}_{10,5}$ and $\widehat{A}_{11,4}$ in its unfolded formulation. If we were to unfold the second field $B_{10,4,1}$ then this would lead to another pair of extra fields $\widehat{B}_{11,3,1}$ and $\widehat{B}_{11,4}$. To see this explicitly one can their first unfolded variables into GL(11) irreducible components:

$$e_{[9]}{}^6 = A_{9,6} \oplus \widehat{A}_{10,5} \oplus \widehat{A}_{11,4}, \tag{300a}$$

$$e_{[10]}{}^{4,1} = B_{10,4,1} \oplus \widehat{B}_{11,3,1} \oplus \widehat{B}_{11,4}. \tag{300b}$$

In total, then, there are four extra fields at level five: $\widehat{A}_{10,5}$, $\widehat{A}_{11,4}$, $\widehat{B}_{11,3,1}$, and $\widehat{B}_{11,4}$.

There are nine fields in $E_{11}$ at level six: $h_{9,8,1}$, $B_{10,6,2}$, $B_{10,7,1}$, $B_{10,8}$, and five other fields with blocks of eleven indices. In order to unfold the fields of height ten or less, we introduce

Table 1: Counting extra fields up to level three.

| $e_{[p_\alpha]}{}^\alpha$ | fields | $E_{11}$ | unfolding | net | $\ell_2$ |
|---|---|---|---|---|---|
| $e_{[1]}{}^1$ | $h_{1,1}$ | 1 | 1 | 0 | 0 |
| | $\widehat{A}_2$ | 1 | 1 | 0 | 0 |
| $A_{[3]}$ | $A_3$ | 1 | 1 | 0 | 0 |
| $A_{[6]}$ | $A_6$ | 1 | 1 | 0 | 0 |
| $e_{[8]}{}^1$ | $h_{8,1}$ | 1 | 1 | 0 | 0 |
| | $\widehat{A}_9$ | 0 | 1 | $-1$ | 1 |

the following variables:

$$e_{[9]}{}^{8,1} = h_{9,8,1} \oplus \widehat{A}_{10,7,1} \oplus \widehat{A}_{10,8} \oplus \widehat{A}_{11,6,1} \oplus \widehat{A}_{11,7}, \tag{301a}$$

$$e_{[10]}{}^{6,2} = B_{10,6,2} \oplus \widehat{B}_{11,5,2} \oplus \widehat{B}_{11,6,1}, \tag{301b}$$

$$e_{[10]}{}^{7,1} = B_{10,7,1} \oplus \widehat{B}_{11,6,1} \oplus \widehat{B}_{11,7}, \tag{301c}$$

$$e_{[10]}{}^{8} = B_{10,8} \oplus \widehat{B}_{11,7}. \tag{301d}$$

Thus there are nine extra fields at level six: $\widehat{A}_{10,7,1}$, $\widehat{A}_{10,8}$, $\widehat{A}_{11,6,1}$, $\widehat{A}_{11,7}$, $\widehat{B}_{11,5,2}$, $\widehat{B}_{11,6,1}$ (two copies), and $\widehat{B}_{11,7}$ (two copies).

In Table 2 we have summarised the unfolded spectrum associated with different sets of $E_{11}$ fields in the theory at levels four, five and six. The first column denotes each type of field that we encounter in the unfolding procedure with their Young tableaux indicated explicitly as a subscript. The second column tells us the multiplicities of the fields in $E_{11}$. We observe from the table that all fields in the first column of each index structure occur in $E_{11}$ if we include those with multiplicity zero, the first examples of which are $A_9$ at level three and $B_{10,2}$ at level four. We do not list all the fields of multiplicity zero, for example $A_{9,9}$ at level six and $B_{10,10,2,2}$ at level eight, since these ones do not play a role in unfolding. The last column gives us the squared length of the $E_{11}$ root associated with each field.

In the third column $\mathcal{U}_{(9)}$ we list all the fields produced by unfolding all the $E_{11}$ fields which have no blocks of ten or eleven indices. These fields are the graviton, three-form, six-form, dual graviton, and the higher dual fields in (9) which contain more blocks of nine indices, i.e. the fields $A_{9,...,9,3}$, $A_{9,...,9,6}$, and $h_{9,...,9,8,1}$. In the fourth column $\mathcal{U}_{(10)}$ we list all the fields produced by unfolding all the fields which have no blocks of eleven indices, and in the fifth column $\mathcal{U}_{(11)}$ we have those produced by unfolding all the fields in $E_{11}$. Note that unfolding the fields with blocks of eleven indices leads to no extra fields, so the $\mathcal{U}_{(11)}$ column is obtained from the $\mathcal{U}_{(10)}$ column by adding to it the fields in $E_{11}$ with blocks of eleven indices. In the $\mathcal{U}_{(11)}$ case, none of the fields in $E_{11}$ can play the role of an extra field since they are all unfolded. The sixth column $\mathcal{U}_{(\alpha^2=2)}$ counts the fields produced by unfolding the fields in $E_{11}$ associated with real roots of the $E_{11}$ algebra. Lastly, in the seventh and eighth columns, we list the multiplicities of all the fields in the $\ell_2$ and $\ell_{10}$ representations of $E_{11}$.

Since all the degrees of freedom are contained in the fields with blocks of at most nine indices, we find that $\mathcal{U}_{(9)}$ contains all the degrees of freedom and so $\ell_2$ seems to be sufficient to encode the dynamics. Here we are only unfolding dynamical fields. The fields with blocks of ten or eleven indices do not contain the degrees of freedom, but nevertheless they can play an important role, the first example being $B_{10,1,1}$ at level four which is responsible for Romans theory. Unfolding only the fields with blocks of at most nine indices produces extra fields that can all be found in the $\ell_2$ representation, at least up to level six. This holds whether or not we

Table 2: Unfolding different sets of $E_{11}$ fields from levels four to six.

| fields | $E_{11}$ | $\mathcal{U}_{(9)}$ | $\mathcal{U}_{(10)}$ | $\mathcal{U}_{(11)}$ | $\mathcal{U}_{(\alpha^2=2)}$ | $\ell_2$ | $\ell_{10}$ | $\alpha^2$ |
|---|---|---|---|---|---|---|---|---|
| $A_{9,3}$ | 1 | 1 | 1 | 1 | 1 | 0 | 0 | 2 |
| $B_{10,1,1}$ | 1 | 0 | 1 | 1 | 1 | 0 | 0 | 2 |
| $B_{10,2}$ | 0 | 1 | 1 | 1 | 1 | 1 | 0 | 0 |
| $C_{11,1}$ | 1 | 1 | 2 | 3 | 2 | 1 | 1 | −2 |
| $A_{9,6}$ | 1 | 1 | 1 | 1 | 1 | 0 | 0 | 2 |
| $B_{10,4,1}$ | 1 | 0 | 1 | 1 | 1 | 0 | 0 | 2 |
| $B_{10,5}$ | 0 | 1 | 1 | 1 | 1 | 1 | 0 | 0 |
| $C_{11,3,1}$ | 1 | 0 | 1 | 2 | 1 | 1 | 0 | 0 |
| $C_{11,4}$ | 1 | 1 | 2 | 3 | 2 | 1 | 1 | −2 |
| $h_{9,8,1}$ | 1 | 1 | 1 | 1 | 1 | 0 | 0 | 2 |
| $B_{10,6,2}$ | 1 | 0 | 1 | 1 | 1 | 0 | 0 | 2 |
| $B_{10,7,1}$ | 1 | 1 | 2 | 2 | 1 | 1 | 0 | 0 |
| $B_{10,8}$ | 1 | 1 | 2 | 2 | 1 | 1 | 0 | −2 |
| $C_{11,4,3}$ | 1 | 0 | 0 | 1 | 1 | 0 | 0 | 2 |
| $C_{11,5,1,1}$ | 1 | 0 | 0 | 1 | 1 | 0 | 0 | 2 |
| $C_{11,5,2}$ | 0 | 0 | 1 | 1 | 1 | 1 | 0 | 0 |
| $C_{11,6,1}$ | 2 | 1 | 3 | 5 | 2 | 2 | 1 | −2 |
| $C_{11,7}$ | 1 | 1 | 3 | 4 | 1 | 2 | 1 | −4 |

allow some $E_{11}$ fields to play the role of extra fields. In other words, if we include $E_{11}$ fields that are not unfolded, then we do not need $\ell_2$ at all.

We see that all the fields in $\mathcal{U}_{(10)}$ can either be found in the set of fields in $E_{11}$ that are not unfolded, or inside the $\ell_2$ representation. It is slightly tricky now because some of the fields in $E_{11}$ have the same Young tableaux as two of the extra fields that appear when we unfold the $h_{9,8,1}$ field – see equation (301a). As worked out in Section 4, unfolding $A_{9,3}$ leads to two extra fields, $\widehat{A}_{10,2}$ and $\widehat{A}_{11,1}$, one of which has the same symmetry type as the extra field $\widehat{B}_{11,1}$ that appears when unfolding the $B_{10,1,1}$ field. It is possible that $C_{11,1}$ at level four in $E_{11}$ could play the role of one of these extra fields, and the other one could come from the $\ell_2$ representation. If we allow some $E_{11}$ fields to play the role of extra fields, such as $C_{11,1}$ and so on, then $\ell_2$ is once again more than sufficient to account for the unfolded spectrum. If it turns out that the fields in $E_{11}$ cannot be used to unfold other fields then we would need to look beyond $\ell_2$ to find all the extra fields. In this case, our counting shows that the $\ell_{10}$ representation of $E_{11}$ would be a good candidate for a source of extra fields beyond the $\ell_2$ representation.

So far, nothing has required us to unfold every field at every level. Part of the problem is to understand which fields need to be unfolded and which do not. If we unfold every field in $E_{11}$ as in the $\mathcal{U}_{(11)}$ case, then all the extra fields would need to come from additional representations of $E_{11}$. In the $\mathcal{U}_{(11)}$ column we have counted all the fields in this maximal unfolded spectrum, and we notice that there is a perfect match up to level six between the number of extra fields and the $\ell_2$ and $\ell_{10}$ representations. In numbers, this means that the entries of the $\mathcal{U}_{(11)}$ column are equal to the sum of those of the $E_{11}$, $\ell_2$ and $\ell_{10}$ columns. This is somewhat misleading: we will see that this perfect match breaks down at level seven.

In the $\mathcal{U}_{(\alpha^2=2)}$ column we proceed by unfolding only the $E_{11}$ fields that correspond to real $E_{11}$ roots, i.e. the roots $\alpha$ whose squared length is equal to two. The last column tells us the squared length of each root. Once again, we find that the fields in $E_{11}$ that are not unfolded and the fields in $\ell_2$ are more than enough to account for this unfolded spectrum of fields.

**Analysis at level seven.** In the non-linear realisation there are twenty-four fields at level seven: $A_{9,9,3}$, $B_{10,7,4}$, $B_{10,8,2,1}$, $B_{10,8,3}$, $B_{10,9,1,1}$, two copies of $B_{10,9,2}$, $B_{10,10,1}$, and also sixteen fields with columns of height eleven. In order to unfold, we introduce the following variables:

$$e_{[9]}{}^{9,3} = A_{9,9,3} \oplus \widehat{A}_{10,8,3} \oplus \widehat{A}_{10,9,2} \oplus \widehat{A}_{11,7,3} \oplus \widehat{A}_{11,8,2} \oplus \widehat{A}_{11,9,1}, \tag{302a}$$

$$e_{[10]}{}^{7,4} = B_{10,7,4} \oplus \widehat{B}_{11,6,4} \oplus \widehat{B}_{11,7,3}, \tag{302b}$$

$$e_{[10]}{}^{8,2,1} = B_{10,8,2,1} \oplus \widehat{B}_{11,7,2,1} \oplus \widehat{B}_{11,8,1,1} \oplus \widehat{B}_{11,8,2}, \tag{302c}$$

$$e_{[10]}{}^{8,3} = B_{10,8,3} \oplus \widehat{B}_{11,7,3} \oplus \widehat{B}_{11,8,2}, \tag{302d}$$

$$e_{[10]}{}^{9,1,1} = B_{10,9,1,1} \oplus \widehat{B}_{11,8,1,1} \oplus \widehat{B}_{11,9,1}, \tag{302e}$$

$$e_{[10]}{}^{9,2} = B_{10,9,2} \oplus \widehat{B}_{11,8,2} \oplus \widehat{B}_{11,9,1}, \tag{302f}$$

$$e_{[10]}{}^{10,1} = B_{10,10,1} \oplus \widehat{B}_{11,9,1} \oplus \widehat{B}_{11,10}. \tag{302g}$$

Note that two copies of $e_{[10]}{}^{9,2}$ are needed since there are two $B_{10,9,2}$ fields in $E_{11}$.

In Table 3 we summarise our analysis at level seven. We find that unfolding only the higher dual field $A_{9,9,3}$ produces extra fields that can all be taken either from $\ell_2$ or from $E_{11}$. Unfolding fields with blocks of at most ten indices leads to more extra fields, and most but not all of them can be taken from $\ell_2$. The rest can be taken either from $E_{11}$ or from an additional representation like $\ell_{10}$, and in either case there are more than enough fields to account for the unfolded spectrum. If we were to unfold all $E_{11}$ fields, we would find that the perfect match up to level six breaks down. In particular, $\ell_2$ and $\ell_{10}$ contain more than enough fields. Lastly, we consider unfolding only the fields in $E_{11}$ that are associated with real roots. In this case at level seven we do not even need $\ell_2$ and we can take all the extra fields from $E_{11}$ itself.

**Analysis at level eight.** We conclude by examining the unfolded spectrum at level eight in the non-linear realisation. There are sixty-seven fields at this level: $A_{9,9,6}$, $B_{10,7,7}$, $B_{10,8,5,1}$, $B_{10,8,6}$, $B_{10,9,3,2}$, two copies of $B_{10,9,4,1}$, two copies of $B_{10,9,5}$, $B_{10,10,2,1,1}$, two copies of $B_{10,10,3,1}$, two copies of $B_{10,10,4}$, and fifty-three fields with columns of height eleven.

$$e_{[9]}{}^{9,6} = A_{9,9,6} \oplus \widehat{A}_{10,8,6} \oplus \widehat{A}_{10,9,5} \oplus \widehat{A}_{11,7,6} \oplus \widehat{A}_{11,8,5} \oplus \widehat{A}_{11,9,4}, \tag{303a}$$

$$e_{[10]}{}^{7,7} = B_{10,7,7} \oplus \widehat{B}_{11,7,6}, \tag{303b}$$

$$e_{[10]}{}^{8,5,1} = B_{10,8,5,1} \oplus \widehat{B}_{11,7,5,1} \oplus \widehat{B}_{11,8,4,1} \oplus \widehat{B}_{11,8,5}, \tag{303c}$$

$$e_{[10]}{}^{8,6} = B_{10,8,6} \oplus \widehat{B}_{11,7,6} \oplus \widehat{B}_{11,8,5}, \tag{303d}$$

$$e_{[10]}{}^{9,3,2} = B_{10,9,3,2} \oplus \widehat{B}_{11,8,3,2} \oplus \widehat{B}_{11,9,2,2} \oplus \widehat{B}_{11,9,3,1}, \tag{303e}$$

$$e_{[10]}{}^{9,4,1} = B_{10,9,4,1} \oplus \widehat{B}_{11,8,4,1} \oplus \widehat{B}_{11,9,3,1} \oplus \widehat{B}_{11,9,4}, \tag{303f}$$

$$e_{[10]}{}^{9,5} = B_{10,9,5} \oplus \widehat{B}_{11,8,5} \oplus \widehat{B}_{11,9,4}, \tag{303g}$$

$$e_{[10]}{}^{10,2,1,1} = B_{10,10,2,1,1} \oplus \widehat{B}_{11,9,2,1,1} \oplus \widehat{B}_{11,10,1,1,1} \oplus \widehat{B}_{11,10,2,1}, \tag{303h}$$

$$e_{[10]}{}^{10,3,1} = B_{10,10,3,1} \oplus \widehat{B}_{11,9,3,1} \oplus \widehat{B}_{11,10,2,1} \oplus \widehat{B}_{11,10,3}, \tag{303i}$$

$$e_{[10]}{}^{10,4} = B_{10,10,4} \oplus \widehat{B}_{11,9,4} \oplus \widehat{B}_{11,10,3}. \tag{303j}$$

In Table 4 we continue our analysis at level eight. Unfolding only the higher dual field $A_{9,9,6}$ produces five extra fields which can all be taken from either $E_{11}$ or $\ell_2$. If we unfold the fields with blocks of at most ten indices, then we find that all the extra fields can be taken from $E_{11}$ and $\ell_2$. It should be noted for this case that an additional representation such as $\ell_{10}$ needs to be added if $E_{11}$ fields are not allowed to play the role of extra fields.

Table 3: Unfolding different sets of $E_{11}$ fields at level seven.

| fields | $E_{11}$ | $\mathcal{U}_{(9)}$ | $\mathcal{U}_{(10)}$ | $\mathcal{U}_{(11)}$ | $\mathcal{U}_{(\alpha^2=2)}$ | $\ell_2$ | $\ell_{10}$ | $\alpha^2$ |
|---|---|---|---|---|---|---|---|---|
| $A_{9,9,3}$ | 1 | 1 | 1 | 1 | 1 | 0 | 0 | 2 |
| $B_{10,7,4}$ | 1 | 0 | 1 | 1 | 1 | 0 | 0 | 2 |
| $B_{10,8,2,1}$ | 1 | 0 | 1 | 1 | 1 | 0 | 0 | 2 |
| $B_{10,8,3}$ | 1 | 1 | 2 | 2 | 1 | 1 | 0 | 0 |
| $B_{10,9,1,1}$ | 1 | 0 | 1 | 1 | 0 | 1 | 0 | 0 |
| $B_{10,9,2}$ | 2 | 1 | 3 | 3 | 1 | 1 | 0 | −2 |
| $B_{10,10,1}$ | 1 | 0 | 1 | 1 | 0 | 2 | 0 | −4 |
| $C_{11,6,3,1}$ | 1 | 0 | 0 | 1 | 1 | 0 | 0 | 2 |
| $C_{11,6,4}$ | 1 | 0 | 1 | 2 | 1 | 1 | 0 | 0 |
| $C_{11,7,2,1}$ | 1 | 0 | 1 | 2 | 1 | 1 | 0 | 0 |
| $C_{11,7,3}$ | 2 | 1 | 3 | 5 | 2 | 2 | 1 | −2 |
| $C_{11,8,1,1}$ | 3 | 0 | 2 | 5 | 1 | 2 | 1 | −2 |
| $C_{11,8,2}$ | 3 | 1 | 5 | 8 | 2 | 4 | 1 | −4 |
| $C_{11,9,1}$ | 4 | 1 | 5 | 9 | 1 | 5 | 2 | −6 |
| $C_{11,10}$ | 1 | 0 | 1 | 2 | 0 | 3 | 1 | −8 |

Table 4: Unfolding different sets of $E_{11}$ fields at level eight.

| fields | $E_{11}$ | $\mathcal{U}_{(9)}$ | $\mathcal{U}_{(10)}$ | $\mathcal{U}_{(11)}$ | $\mathcal{U}_{(\alpha^2=2)}$ | $\ell_2$ | $\ell_{10}$ | $\alpha^2$ |
|---|---|---|---|---|---|---|---|---|
| $A_{9,9,6}$ | 1 | 1 | 1 | 1 | 1 | 0 | 0 | 2 |
| $B_{10,7,7}$ | 1 | 0 | 1 | 1 | 1 | 0 | 0 | 2 |
| $B_{10,8,5,1}$ | 1 | 0 | 1 | 1 | 1 | 0 | 0 | 2 |
| $B_{10,8,6}$ | 1 | 1 | 2 | 2 | 1 | 1 | 0 | 0 |
| $B_{10,9,3,2}$ | 1 | 0 | 1 | 1 | 1 | 0 | 0 | 2 |
| $B_{10,9,4,1}$ | 2 | 0 | 2 | 2 | 0 | 1 | 0 | 0 |
| $B_{10,9,5}$ | 2 | 1 | 3 | 3 | 1 | 1 | 0 | −2 |
| $B_{10,10,2,1,1}$ | 1 | 0 | 1 | 1 | 1 | 0 | 0 | 2 |
| $B_{10,10,3,1}$ | 2 | 0 | 2 | 2 | 0 | 2 | 0 | −2 |
| $B_{10,10,4}$ | 2 | 0 | 2 | 2 | 0 | 2 | 0 | −4 |
| $C_{11,6,6,1}$ | 1 | 0 | 0 | 1 | 1 | 0 | 0 | 2 |
| $C_{11,7,4,2}$ | 1 | 0 | 0 | 1 | 1 | 0 | 0 | 2 |
| $C_{11,7,5,1}$ | 1 | 0 | 1 | 2 | 1 | 1 | 0 | 0 |
| $C_{11,7,6}$ | 2 | 1 | 3 | 5 | 2 | 2 | 1 | −2 |
| $C_{11,8,3,1,1}$ | 1 | 0 | 0 | 1 | 1 | 0 | 0 | 2 |
| $C_{11,8,3,2}$ | 1 | 0 | 1 | 2 | 1 | 1 | 0 | 0 |
| $C_{11,8,4,1}$ | 4 | 0 | 3 | 7 | 1 | 3 | 1 | −2 |
| $C_{11,8,5}$ | 3 | 1 | 5 | 8 | 2 | 4 | 1 | −4 |
| $C_{11,9,2,1,1}$ | 1 | 0 | 1 | 2 | 1 | 1 | 0 | 0 |
| $C_{11,9,2,2}$ | 2 | 0 | 1 | 3 | 1 | 1 | 0 | −2 |
| $C_{11,9,3,1}$ | 6 | 0 | 5 | 11 | 1 | 6 | 2 | −4 |
| $C_{11,9,4}$ | 7 | 1 | 7 | 14 | 1 | 7 | 2 | −6 |
| $C_{11,10,1,1,1}$ | 2 | 0 | 1 | 3 | 1 | 2 | 1 | −2 |
| $C_{11,10,2,1}$ | 7 | 0 | 3 | 10 | 1 | 8 | 2 | −6 |
| $C_{11,10,3}$ | 6 | 0 | 4 | 10 | 0 | 10 | 3 | −8 |
| $C_{11,11,1,1}$ | 3 | 0 | 0 | 3 | 0 | 5 | 3 | −8 |
| $C_{11,11,2}$ | 5 | 0 | 0 | 5 | 0 | 8 | 3 | −10 |

The perfect match that we noticed when we unfolded all $E_{11}$ fields in the theory up to level six was broken at level seven, and it continues to be broken at level eight. We notice a perfect match between $E_{11} \oplus \ell_2 \oplus \ell_{10}$ and the maximal unfolded spectrum in the $\mathcal{U}_{(11)}$ column for sixteen of the twenty-seven types of field in Table 4. The fields for which $\ell_{10}$ is not needed since $E_{11}$ and $\ell_2$ alone match the unfolded spectrum are $B_{11,8,4,1}$ and $B_{11,9,4}$. Furthermore, the fields for which $E_{11}$ and $\ell_2$ are already larger than needed are $B_{10,9,4,1}$, $B_{10,10,3,1}$, $B_{10,10,4}$, $B_{11,9,3,1}$, $B_{11,10,1,1,1}$, $B_{11,10,2,1}$, $B_{11,10,3}$, $B_{11,11,1,1}$, and $B_{11,11,2}$. As at level seven, it seems that we need to pick and choose which fields to unfold rather than unfolding everything at once. If we unfold only the fields corresponding to real roots, it is clear from the $\mathcal{U}_{(\alpha^2=2)}$ column in Table 4 that all the extra fields can be taken either from $E_{11}$ or from $\ell_2$.

It would be interesting to have a maximal set of $E_{11}$ fields whose unfolded spectrum is a subset of $E_{11}$ and the $\ell_2$ representation. This way one would not need to worry about $\ell_{10}$ or any further highest weight representation that might need to be added at much higher levels. Moreover, even if we unfold all $E_{11}$ fields, the extra fields do not 'fill up' $\ell_2$ and $\ell_{10}$. If we were to include both of these representations while desiring a perfect match then we would need to add even more irreducible tensor fields to the unfolded spectrum, and it is not clear where such fields would even come from.

# 6 Unfolding $A_1^{+++}$ at low levels

The non-linear realisation of $A_1^{+++}$ is known to contain and extend gravity in four dimensions to a theory featuring an infinite number of fields, including all higher dual gravitons [19,74]. In order to make contact with this theory, we will unfold the fields in $A_1^{+++}$ up to level three. The Dynkin diagram of $A_1^{+++}$ is given by

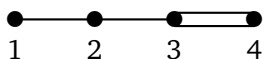

At level zero the only field is the graviton $h_{1,1}$ and its unfolded formulation is identical to that given in Section 3.2, where all the zero-forms are now valued in irreducible representations of GL(4) rather than GL(11). Unfolding on-shell, all the zero-forms will be valued in irreducible representations of the Lorentz group SO(1,3). At level one there is only one field, the dual graviton $h_{1,1}^{(0)}$, and its unfolded formulation is exactly the same as that of the graviton at level zero since they are both symmetric rank-two tensors in four dimensions. At higher levels we find an infinite number of fields, including the family of higher dual fields containing the first higher dual graviton $h_{2,1,1}^{(1)}$ at level two, and more generally we find the $n^{\text{th}}$ higher dual graviton $h_{2^n,1,1}^{(n)}$ at level $n+1$ for arbitrary $n \geq 2$:

$$h_{1,1} \sim \square\square\,, \qquad h_{1,1}^{(0)} \sim \square\square\,, \qquad h_{2^n,1,1}^{(n)} \sim \overset{\cdots}{\underset{\cdots}{\square\square}}\square\square\square\,. \qquad (304)$$

The graviton and the dual graviton both transform with a vector gauge parameter in their unfolded formulations, and an extra two-form is introduced alongside each of them. The first two-form can be eliminated with an $I_c(A_1^{+++})$ transformation at level zero, i.e. a local Lorentz transformation. The second two-form is analogous to the extra nine-form in Section 3.3, and it was shown in any number of space-time dimensions that this extra field can be eliminated from the action for dual gravity using the (Hodge dual of the) Lorentz gauge parameter, leaving an action only in terms of the dual graviton [52].

The zero-forms in the unfolded module of the graviton have the same tableaux as they did in equation (65). In four dimensions the graviton and dual graviton have the same tableau,

so their unfolded equations look the same and the zero-forms in their unfolded modules have the same symmetry types:

$$\mathcal{T}(h_{1,1}) = \left\{ C^{(0)}_{2,2}, C^{(1)}_{2,2,1}, C^{(2)}_{2,2,1,1}, \dots \right\}, \qquad \mathcal{T}(h^{(0)}_{1,1}) = \left\{ \widetilde{C}^{(0)}_{2,2}, \widetilde{C}^{(1)}_{2,2,1}, \widetilde{C}^{(2)}_{2,2,1,1}, \dots \right\}. \tag{305}$$

In the non-linear realisation of $A_1^{+++}$ it was found [19] that the first-order on-shell duality relation between gravity and dual gravity takes the form

$$\omega^{(0)}_{a|b[2]} \propto \varepsilon_{b[2]}{}^{c[2]} \omega_{a|c[2]}, \tag{306}$$

where $\omega_{1|2}$ and $\omega^{(0)}_{1|2}$ are the first-order connections associated with the graviton and the dual graviton, respectively. Taking derivatives leads to the on-shell curvature relation

$$\partial_{[b_1}\partial_{[a_1}h^{(0)}{}_{a_2],b_2]} \propto \varepsilon_{b_1 b_2}{}^{c_1 c_2}\partial_{[c_1}\partial_{[a_1}h_{a_2],c_2]}. \tag{307}$$

This can equivalently be expressed in terms of primary zero-forms in a similar way to equation (49) in eleven dimensions:

$$\widetilde{C}^{(0)}{}_{a_1 a_2, b_2 b_2} \propto \varepsilon_{b_1 b_2}{}^{c_1 c_2} C^{(0)}{}_{a_1 a_2, c_1 c_2}. \tag{308}$$

The unfolded equations allow us to write the primary zero-forms $C^{(0)}_{2,2}$ and $\widetilde{C}^{(0)}_{2,2}$ as proportional to the curvature tensors $\partial_{[b_1}\partial_{[a_1}h_{a_2],b_2]}$ and $\partial_{[b_1}\partial_{[a_1}h^{(0)}{}_{a_2],b_2]}$, respectively. Taking traces leads to the linearised equations of motion for gravity and dual gravity:

$$\partial^{[c}\partial_{[c}h_{a]}{}^{b]} = 0, \qquad \partial^{[c}\partial_{[c}h^{(0)}{}_{a]}{}^{b]} = 0. \tag{309}$$

The first higher dual graviton $h^{(1)}_{2,1,1} \sim$ ▯ is the only $A_1^{+++}$ field at level two. The first three unfolded equations are given by

$$\mathrm{d}e_{[2]}{}^{a,b} + h_{c[2]}\,\omega_{[1]}{}^{c[2](a,b)} = 0, \tag{310a}$$

$$\mathrm{d}\omega_{[1]}{}^{a[3],b} + h_c\,X_{[1]}{}^{a[3],bc} = 0, \tag{310b}$$

$$\mathrm{d}X_{[1]}{}^{a[3],b[2]} + h_{c[2]}\,C^{a[3],b[2],c[2]} = 0, \tag{310c}$$

and their gauge transformations are

$$\delta e_{[2]}{}^{a,b} = \mathrm{d}\lambda_{[1]}{}^{a,b} - h_{c[2]}\,\alpha^{c[2](a,b)}, \tag{311a}$$

$$\delta \omega_{[1]}{}^{a[3],b} = \mathrm{d}\alpha^{a[3],b} + h_c\,\beta^{a[3],bc}, \tag{311b}$$

$$\delta X_{[1]}{}^{a[3],b[2]} = \mathrm{d}\beta^{a[3],b[2]}. \tag{311c}$$

The primary zero-form $C_{3,2,2}$ is the first in the tower

$$\mathcal{T}(h^{(1)}_{2,1,1}) = \left\{ C^{(n)}_{3,2,2,1^n} \,|\, n \in \mathbb{N} \right\} = \left\{ C^{(0)}_{3,2,2}, C^{(1)}_{3,2,2,1}, C^{(2)}_{3,2,2,1,1}, \dots \right\}. \tag{312}$$

Decomposing the fields and parameters into irreducible components, we obtain

$$e_{a_1 a_2|b,c} = h^{(1)}_{a_1 a_2,b,c} + \widehat{A}_{a_1 a_2(b,c)}, \qquad \lambda_{a|b,c} = \lambda^{(1)}{}_{a,b,c} + \lambda^{(2)}{}_{a(b,c)}. \tag{313}$$

$$\square \otimes \square\square = \square\square\square \oplus \begin{smallmatrix}\square\square\\\square\end{smallmatrix}, \qquad \square \otimes \square\square = \square\square\square \oplus \begin{smallmatrix}\square\square\\\square\end{smallmatrix}. \tag{314}$$

and the irreducible fields transform as

$$\delta h^{(1)}_{a_1 a_2,b,c} = \partial_{[a_1}\lambda^{(1)}{}_{a_2],b,c} + \frac{1}{4}\partial_{[a_1}\lambda^{(2)}{}_{a_2](b,c)} - \frac{3}{8}\partial_{(b|}\lambda^{(2)}{}_{a_1 a_2,|c)}, \tag{315a}$$

$$\delta \widehat{A}_{a_1 a_2 a_3,b} = \frac{9}{8}\partial_{[a_1}\lambda^{(2)}{}_{a_2 a_3],b} - \alpha_{a_1 a_2 a_3,b}. \tag{315b}$$

The purpose of the $\alpha_{3,1}$ parameter is to shift away the extra field, but this is very different from the off-shell picture [74] where $\widehat{A}_{3,1}$ cannot be eliminated from the higher dual action, and where both $\delta h^{(1)}_{2,1,1}$ and $\delta\widehat{A}_{3,1}$ also include strange terms containing the vector gauge parameter of the dual graviton at level one. The gauge transformations found above match the those of the $A_1^{+++}$ non-linear realisation up to certain factors, but neither of these frameworks is able to reproduce the strange intertwined gauge transformations in the higher dual action principle for linearised gravity in four dimensions. There is an extra two-form field at level one and an extra $\widehat{A}_{3,1}$ field at level two, and these fields precisely match the generators of the $\ell_2$ representation of $A_1^{+++}$ at levels zero and one [74].

The first-order duality relation between the dual graviton $h^{(0)}_{1,1}$ at level one and the higher dual graviton $h^{(1)}_{2,1,1}$ at level two was found [74] to take the form

$$\omega^{(1)}_{a|b[3],c} \propto \varepsilon_{b[3]}{}^{p} \omega^{(0)}_{c|ap}, \tag{316}$$

where $\omega^{(1)}{}_{[1]}{}^{3,1}$ is the first-order connection in the $h^{(1)}_{2,1,1}$ unfolded equations (310a) and (310b). Taking derivatives leads to a curvature relation between the primary zero-form $C^{(0)}_{3,2,2}$ of $h^{(1)}_{2,1,1}$ to the zero-form $\widetilde{C}^{(1)}_{2,2,1} \in \mathcal{T}(h^{(0)}_{1,1})$:

$$C^{(0)}{}_{a_1 a_2 a_3, b_1 b_2, c_1 c_2} = \varepsilon_{a_1 a_2 a_3}{}^{d} \widetilde{C}^{(1)}{}_{b_1 b_2, c_1 c_2, d}. \tag{317}$$

This is analogous to (162) in eleven dimensions, and it is the same as equations (2.33) and (4.36) in reference [74] that we worked out, respectively, from the $A_1^{+++}$ non-linear realisation and a higher dual action principle featuring both $h^{(1)}_{2,1,1}$ and $\widehat{A}_{3,1}$. Working on-shell and writing the zero-forms in terms of their respective fields, we find the $h^{(1)}_{2,1,1}$ equations of motion expressed as trace constraints on its curvature:

$$(\mathrm{Tr}_{1,2})^2(C^{(0)}_{3,2,2}) = (\mathrm{Tr}_{1,3})^2(C^{(0)}_{3,2,2}) = 0, \qquad \mathrm{Tr}_{2,3}(C^{(0)}_{3,2,2}) = 0, \tag{318}$$

where $C^{(0)}_{3,2,2}$ is proportional to the curvature tensor $\partial_{[c_1}\partial_{[b_1}\partial_{[a_1}h^{(1)}{}_{a_2 a_3],b_2],c_2]}$. The first of these linearised equations was found in the $A_1^{+++}$ non-linear realisation and both of them have been obtained from a higher dual action – see equations (2.38), (4.37) and (4.38) in [74].

At level three there are two fields in the $A_1^{+++}$ non-linear realisation: the second higher dual graviton $h^{(2)}_{2,2,1,1}$ and a non-dynamical field $B_{3,2,1}$ that has not yet been studied. The first few unfolded equations for the $h^{(2)}_{2,2,1,1}$ field are given by

$$\mathrm{d}e_{[2]}{}^{a[2],b,c} + h_d\,\omega_{[1]}{}^{d\langle a[2],b,c\rangle} = 0, \tag{319a}$$

$$\mathrm{d}\omega_{[2]}{}^{a[3],b,c} + h_{d[2]} X_{[1]}{}^{a[3],d[2](b,c)} = 0, \tag{319b}$$

$$\mathrm{d}X_{[1]}{}^{a[3],b[3],c} + h_d\,Y_{[1]}{}^{a[3],b[3],dc} = 0, \tag{319c}$$

$$\mathrm{d}Y_{[1]}{}^{a[3],b[3],c[2]} + h_{d[2]} C^{a[3],b[3],c[2],d[2]} = 0, \tag{319d}$$

and their gauge transformations are

$$\delta e_{[2]}{}^{a[2],b,c} = \mathrm{d}\lambda_{[1]}{}^{a[2],b,c} + h_d\,\alpha_{[1]}{}^{d\langle a[2],b,c\rangle}, \tag{320a}$$

$$\delta\omega_{[2]}{}^{a[3],b,c} = \mathrm{d}\alpha_{[1]}{}^{a[3],b,c} - h_{d[2]}\beta^{a[3],d[2](b,c)}, \tag{320b}$$

$$\delta X_{[1]}{}^{a[3],b[3],c} = \mathrm{d}\beta^{a[3],b[3],c} + h_d\,\gamma^{a[3],b[3],dc}, \tag{320c}$$

$$\delta Y_{[1]}{}^{a[3],b[3],c[2]} = \mathrm{d}\gamma^{a[3],b[3],c[2]}, \tag{320d}$$

where $\langle\cdots\rangle$ denotes a projection onto the GL(11) irreducible $\mathbb{Y}[2,1,1]$ tableau. Decomposing all the fields into irreducible components, we find

$$e_{[2]}{}^{2,1,1} = h_{2,2,1,1} + \widehat{A}_{3,1,1,1} + \widehat{A}_{3,2,1} + \widehat{A}_{4,1,1}. \tag{321}$$

$$\square \otimes \text{(tableau)} = \text{(tableau)} \oplus \text{(tableau)} \oplus \text{(tableau)} \oplus \text{(tableau)}. \tag{322}$$

Similarly, decomposing the gauge parameters leads to

$$\lambda_{[1]}{}^{2,1,1} = \lambda^{(1)}_{2,1,1,1} + \lambda^{(2)}_{2,2,1} + \lambda^{(3)}_{3,1,1}, \quad \square \otimes \text{(tableau)} = \text{(tableau)} \oplus \text{(tableau)} \oplus \text{(tableau)}, \tag{323}$$

$$\alpha_{[1]}{}^{3,1,1} = \alpha^{(1)}_{3,1,1,1} + \alpha^{(2)}_{3,2,1} + \alpha^{(3)}_{4,1,1}, \quad \square \otimes \text{(tableau)} = \text{(tableau)} \oplus \text{(tableau)} \oplus \text{(tableau)}. \tag{324}$$

The extra fields $\{\widehat{A}_{3,1,1,1}, \widehat{A}_{3,2,1}, \widehat{A}_{4,1,1}\}$ are known to appear inside the action principle for the second higher dual graviton [74] and they are found in the $\ell_2$ representation of $A_1^{+++}$ at level two. Similarly, the gauge parameters are found in the $\ell_1$ representation of $A_1^{+++}$ at level three. The extra fields can all be set to zero using the components of the $\alpha_{[1]}{}^{2,1,1}$ parameter.

Now consider the second field $B_{3,2,1}$ at level three. The first three unfolded equations are

$$\mathrm{d}e_{[3]}{}^{a[2],b} + h_{c[2]}\,\omega_{[2]}{}^{c[2]\langle a[2],b\rangle} = 0, \tag{325a}$$
$$\mathrm{d}\omega_{[2]}{}^{a[4],b} + h_{c[2]}X_{[1]}{}^{a[4],bc[2]} = 0, \tag{325b}$$
$$\mathrm{d}X_{[1]}{}^{a[4],b[3]} + h_{c[2]}C^{a[4],b[3],c[2]} = 0, \tag{325c}$$

and their associated gauge transformations are given by

$$\delta e_{[3]}{}^{a[2],b} = \mathrm{d}\lambda_{[2]}{}^{a[2],b} - h_{c[2]}\,\alpha_{[1]}{}^{c[2]\langle a[2],b\rangle}, \tag{326a}$$
$$\delta\omega_{[2]}{}^{a[4],b} = \mathrm{d}\alpha_{[1]}{}^{a[4],b} - h_{c[2]}\beta^{a[4],bc[2]}, \tag{326b}$$
$$\delta X_{[1]}{}^{a[4],b[3]} = \mathrm{d}\beta^{a[4],b[3]}, \tag{326c}$$

where $\langle\cdots\rangle$ denotes a projection onto the $\mathbb{Y}[2,1]$ tableau. Decomposing the fields and gauge parameters, we find

$$e_{[3]}{}^{2,1} = B_{3,2,1} + \widehat{B}_{4,1,1} + \widehat{B}_{4,2}, \quad \square \otimes \text{(tableau)} = \text{(tableau)} \oplus \text{(tableau)} \oplus \text{(tableau)}, \tag{327}$$

$$\lambda_{[2]}{}^{2,1} = \lambda^{(1)}_{2,2,1} + \lambda^{(2)}_{3,1,1} + \lambda^{(3)}_{3,2}, \quad \square \otimes \text{(tableau)} = \text{(tableau)} \oplus \text{(tableau)} \oplus \text{(tableau)}, \tag{328}$$

$$\alpha_{[1]}{}^{4,1} = \alpha^{(1)}_{4,1,1} + \alpha^{(2)}_{4,2}, \quad \square \otimes \text{(tableau)} = \text{(tableau)} \oplus \text{(tableau)}. \tag{329}$$

As before, the extra fields are found in the $\ell_2$ representation of $A_1^{+++}$ and they can both be set to zero using $\alpha_{[1]}{}^{4,1}$. The components of $\lambda_{[2]}{}^{2,1}$ are found in the $\ell_1$ representation.

In the same way that fields beyond $E_{11}$ and its $\ell_2$ representation may need to be added to the $E_{11}$ non-linear realisation, at level three in $A_1^{+++}$ we notice that there is only one field of symmetry type $\mathbb{Y}[4,1,1]$ in the $\ell_2$ representation of $A_1^{+++}$ while the unfolded spectrum has two of them, one for each of the fields that we unfold at level three. It may be possible that these two extra fields are one and the same, or that one of them must lie beyond $A_1^{+++}$ and its $\ell_2$ representation. In order to be certain we would need to extend the $A_1^{+++}$ non-linear realisation to incorporate dynamical $\ell_2$ fields, but that is beyond the scope of this paper.

**Unfolding at higher levels.** Now we extend our analysis to arbitrarily high levels. In order to unfold the $n^{\text{th}}$ higher dual graviton $h^{(n)}_{2^n,1,1}$ at level $n+1$ in the $A^{+++}_1$ non-linear realisation, we introduce a tower of variables

$$e_{[9]}{}^{9^{n-1},8,1}, \quad \omega_{[9]}{}^{10,9^{n-2},8,1}, \quad X_{[9]}{}^{10^2,9^{n-3},8,1}, \quad X_{[9]}{}^{10^3,9^{n-4},8,1}, \quad \dots \tag{330}$$

Schematically, the first unfolded equation is

$$\mathrm{d}e_{[2]}{}^{2^{n-1},1,1} + h_1\,\omega_{[2]}{}^{3,2^{n-2},1,1} = 0, \tag{331a}$$

and it is invariant under the gauge symmetries

$$\delta e_{[2]}{}^{2^{n-1},1,1} = \mathrm{d}\lambda_{[1]}{}^{2^{n-1},1,1} + h_1\,\alpha_{[1]}{}^{3,2^{n-2},1,1}, \tag{332a}$$

$$\delta \omega_{[2]}{}^{3,2^{n-2},1,1} = \mathrm{d}\alpha_{[1]}{}^{3,2^{n-2},1,1}. \tag{332b}$$

The tower (330) continues as

$$\dots, \quad X_{[2]}{}^{3^{n-2},2,1,1}, \quad X_{[2]}{}^{3^{n-1},1,1}, \quad X_{[1]}{}^{3^n,1}, \quad X_{[1]}{}^{3^n,2}, \quad C^{3^n,2,2}, \quad \dots, \tag{333}$$

where the primary zero-form $C_{3^n,2,2}$ is the first in the tower

$$\mathcal{T}(h^{(n)}_{2^n,1,1}) = \{C^{(n)}_{3^n,2,2,1^n} \mid n \in \mathbb{N}\} = \left\{C^{(0)}_{3^n,2,2}, C^{(1)}_{3^n,2,2,1}, C^{(2)}_{3^n,2,2,1,1}, \dots \right\}. \tag{334}$$

We can use the generalised Poincaré lemma to express $C^{(0)}_{3^n,2,2}$ as the curvature tensor

$$C^{(0)}{}_{a^1[3],\dots,a^n[3],b[2],c[2]} = \partial_{[c_1}\partial_{[b_1}\partial_{[a^n_1|}\dots\partial_{[a^2_1}\partial_{[a^1_1}h^{(n)}{}_{a^1_2 a^1_3],a^2_2 a^2_3],\dots,|a^n_2 a^n_3],b_2],c_2]}. \tag{335}$$

One immediately notices that this curvature is gauge-invariant under

$$\delta h^{(n)}{}_{a^1[2],\dots,a^n[2],b,c} = \left[\partial_{[a^n_1|}\lambda^{(1)}{}_{a^1[2],\dots,a^{n-1}[2],|a^n_2],b,c} + \partial_c\lambda^{(2)}{}_{a^1[2],\dots,a^n[2],b}\right]_{2^n,1,1}. \tag{336}$$

The unfolded formulations of the graviton $h_{1,1}$, dual graviton $h^{(0)}_{1,1}$, and higher dual gravitons $h^{(n)}_{2^n,1,1}$ involve first-order connections

$$\omega_{[1]}{}^2, \quad \omega^{(0)}_{[1]}{}^2, \quad \omega^{(1)}_{[1]}{}^{3,1}, \quad \omega^{(2)}_{[2]}{}^{3,1,1}, \quad \omega^{(3)}_{[2]}{}^{3,2,1,1}, \quad \dots, \quad \omega^{(n)}_{[2]}{}^{3,2^{n-2},1,1}, \quad \dots, \tag{337}$$

and we use all these variables to write our infinite tower of first-order on-shell duality relations, starting with (306) and (316), followed by the duality relation

$$\omega^{(2)}_{a[2]|b[3],c,d} \propto \varepsilon_{b[3]}{}^p\,\omega^{(1)}_{c|pa[2],d} \tag{338}$$

between $h^{(2)}_{2,2,1,1}$ at level three and $h^{(1)}_{2,1,1}$ at level two. Taking derivatives leads to

$$\partial_{[d_1}\partial_{[c_1}\partial_{[b_1}\partial_{[a_1}h^{(2)}{}_{a_2 a_3],b_2 b_3],c_2],d_2]} \propto \varepsilon_{a_1\dots a_3}{}^e\partial_e\partial_{[d_1}\partial_{[c_1}\partial_{[b_1}h^{(1)}{}_{b_2 b_3],c_2],d_2]}, \tag{339}$$

and taking appropriate traces leads to the equations of motion for the $h^{(2)}_{2,2,1,1}$ field in terms of its curvature tensor $C^{(0)}_{3,3,2,2}$:

$$
\begin{aligned}
(\mathrm{Tr}_{1,2})^3(C_{3,3,2,2}) &= 0, & (\mathrm{Tr}_{1,3})^2(C_{3,3,2,2}) &= 0, & (\mathrm{Tr}_{2,3})^2(C_{3,3,2,2}) &= 0, \\
\mathrm{Tr}_{3,4}(C_{3,3,2,2}) &= 0, & (\mathrm{Tr}_{1,4})^2(C_{3,3,2,2}) &= 0, & (\mathrm{Tr}_{2,4})^2(C_{3,3,2,2}) &= 0.
\end{aligned}
\tag{340}
$$

Equation (339) can be written in terms of $C^{(0)}_{3,3,2,2} \in \mathcal{T}(h^{(2)}_{2,2,1,1})$ and $C^{(1)}_{3,2,2,1} \in \mathcal{T}(h^{(1)}_{2,1,1})$:

$$C^{(0)}{}_{a_1 a_2 a_3, b_1 b_2 b_3, c_1 c_2, d_1 d_2} = \varepsilon_{a_1 a_2 a_3}{}^e C^{(1)}{}_{b_1 b_2 b_3, c_1 c_2, d_1 d_2, e} \,. \tag{341}$$

Integrating three times leads to the first-order relation

$$\partial_{[a_1} h^{(2)}{}_{a_2 a_3], b_1 b_2, c, d} + \partial_{[b_1} \Xi_{b_2]|a_1 a_2 a_3, c, d} \propto \varepsilon_{a_1 a_2 a_3}{}^e \left( \partial_{[e} h^{(1)}{}_{b_1 b_2], c, d} + \partial_{(c|} \Theta_{e b_1 b_2, |d)} \right) . \tag{342}$$

The arbitrary tensors $\Xi_{1|3,1,1}$ and $\Theta_{3,1}$ have the same tensor structure as $\alpha_{[1]}{}^{3,1,1}$ in (324) and $\alpha_{3,1}$ in (315b), respectively, so we identify $\Theta_{3,1}$ with the extra field $\widehat{A}_{3,1}$ at level two that is shifted away by $\alpha_{3,1}$, and we identify the components of $\Xi_{1|3,1,1}$ with the extra fields in (321) at level three that are shifted away using the $\alpha_{[1]}{}^{3,1,1}$ parameter.

Now we will propose a chain of first-order duality relations for higher dual gravity fields $h^{(n)}_{2^n,1,1}$ for $n \geq 2$ in terms of their first-order connections in (337).

$$\omega^{(3)}_{a[2]|b[3],c[2],d,e} \propto \varepsilon_{b[3]}{}^p \omega^{(2)}_{a[2]|pc[2],d,e} \,, \tag{343a}$$

$$\omega^{(4)}_{a[2]|b[3],c[2],d[2],e,f} \propto \varepsilon_{b[3]}{}^p \omega^{(3)}_{a[2]|pc[2],d[2],e,f} \,, \tag{343b}$$

$$\vdots$$

$$\omega^{(n)}_{a[2]|b[3],c[2],d^1[2],\dots,d^{n-3}[2],e,f} \propto \varepsilon_{b[3]}{}^p \omega^{(n-1)}_{a[2]|pc[2],d^1[2],\dots,d^{n-3}[2],e,f} \,. \tag{343c}$$

Taking derivatives leads naturally to relations between zero-forms $C^{(0)}_{3^n,2,2} \in \mathcal{T}(h^{(n)}_{2^n,1,1})$ and $\widetilde{C}^{(n)}_{2,2,1^n} \in \mathcal{T}(h^{(0)}_{1,1})$ that are analogous to (288) in eleven dimensions:

$$C^{(0)}{}_{a^1[3],\dots,a^n[3],b[2],c[2]} = \varepsilon_{a^1[3]}{}^{d_1} \cdots \varepsilon_{a^n[3]}{}^{d_n} \widetilde{C}^{(n)}{}_{b[2],c[2],d_1,\dots,d_n} \,. \tag{344}$$

Working on-shell, the Lorentz irreducibility properties of $\widetilde{C}^{(n)}_{2,2,1^n}$ are exchanged under (344) with constraints on $C^{(0)}_{3^n,2,2}$, including higher trace constraints that we interpret as the linearised equations of motion for the $h^{(n)}_{2^n,1,1}$ field:

$$\begin{aligned} (\mathrm{Tr}_{i,j})^3(C_{3^n,2,2}) &= 0 \,, & (\mathrm{Tr}_{i,n+1})^2(C_{3^n,2,2}) &= 0 \,, \\ \mathrm{Tr}_{n+1,n+2}(C_{3^n,2,2}) &= 0 \,, & (\mathrm{Tr}_{i,n+2})^2(C_{3^n,2,2}) &= 0 \,, \end{aligned} \qquad 1 \leq i < j \leq n \,. \tag{345}$$

The zero-form relations (344) for adjacent higher duals $h^{(n)}_{2^n,1,1}$ and $h^{(n-1)}_{2^{n-1},1,1}$ together imply a new relation between $C^{(0)}_{3^n,2,2} \in \mathcal{T}(h^{(n)}_{2^n,1,1})$ and $C^{(1)}_{3^{n-1},2,2,1} \in \mathcal{T}(h^{(n-1)}_{2^{n-1},1,1})$:

$$C^{(0)}{}_{a^1[3],a^2[3],\dots,a^n[3],b[2],c[2]} = \varepsilon_{a^1[3]}{}^d C^{(1)}{}_{a^2[3],\dots,a^n[3],b[2],c[2],d} \,. \tag{346}$$

Using (335) to express these zero-forms in terms of their respective dual fields, (346) becomes the on-shell duality relation

$$\partial_{[c_1} \partial_{[b_1} \partial_{[a_1^n|} \cdots \partial_{[a_1^2} \partial_{[a_1^1} h^{(n)}{}_{a_2^1 a_3^1], a_2^2 a_3^2], \dots, |a_2^n a_3^n], b_2], c_2]}$$
$$\propto \varepsilon_{a^1[3]}{}^d \partial_d \partial_{[c_1} \partial_{[b_1} \partial_{[a_1^n|} \cdots \partial_{[a_1^2} h^{(n-1)}{}_{a_2^2 a_3^2], \dots, |a_2^n a_3^n], b_2], c_2]} \,. \tag{347}$$

Repeated integration of (347) for $n > 2$ leads to

$$\partial_{[a_1^1} h^{(n)}{}_{a_2^1 a_3^1], a^2[2], \dots, a^n[2], b, c} + \partial_{[a_1^2} \Xi_{a_2^2]|a^1[3], a^3[2], \dots, a^n[2], b, c} + \partial_{(b|} \Upsilon_{a^1[3], a^2[2], \dots, a^n[2], |c)}$$
$$\propto \varepsilon_{a^1[3]}{}^d \left( \partial_{[d} h^{(n-1)}{}_{a_1^2 a_2^2], a^3[2], \dots, a^n[2], b, c} + \partial_{[a_1^3} \Theta_{a_2^3]|d a^2[2], a^4[2], \dots, a^n[2], b, c} \right) . \tag{348}$$

Table 5: Unfolding different sets of $A_1^{+++}$ fields from levels one to four.

| fields | $A_1^{+++}$ | $\mathcal{U}_{(2)}$ | $\mathcal{U}_{(3)}$ | $\mathcal{U}_{(4)}$ | $\mathcal{U}_{(\alpha^2=2)}$ | $\ell_2$ | $\alpha^2$ |
|---|---|---|---|---|---|---|---|
| $h_{1,1}^{(0)}$ | 1 | 1 | 1 | 1 | 1 | 0 | 2 |
| $B_2$ | 0 | 1 | 1 | 1 | 1 | 1 | 0 |
| $h_{2,1,1}^{(1)}$ | 1 | 1 | 1 | 1 | 1 | 0 | 2 |
| $B_{3,1}$ | 0 | 1 | 1 | 1 | 1 | 1 | $-2$ |
| $h_{2,2,1,1}^{(2)}$ | 1 | 1 | 1 | 1 | 1 | 0 | 2 |
| $B_{3,1,1,1}$ | 0 | 1 | 1 | 1 | 1 | 1 | 0 |
| $B_{3,2,1}$ | 1 | 1 | 2 | 2 | 1 | 1 | $-4$ |
| $C_{4,1,1}$ | 0 | 1 | 2 | 2 | 1 | 1 | $-6$ |
| $C_{4,2}$ | 0 | 0 | 1 | 1 | 0 | 1 | $-8$ |
| $h_{2,2,2,1,1}^{(3)}$ | 1 | 1 | 1 | 1 | 1 | 0 | 2 |
| $B_{3,2,1,1,1}$ | 1 | 1 | 2 | 2 | 1 | 1 | $-2$ |
| $B_{3,2,2,1}$ | 2 | 1 | 3 | 3 | 1 | 2 | $-6$ |
| $B_{3,3,1,1}$ | 1 | 0 | 1 | 1 | 0 | 1 | $-8$ |
| $B_{3,3,2}$ | 1 | 0 | 1 | 1 | 0 | 1 | $-10$ |
| $C_{4,1,1,1,1}$ | 0 | 0 | 1 | 1 | 0 | 1 | $-4$ |
| $C_{4,2,1,1}$ | 1 | 1 | 5 | 6 | 1 | 4 | $-10$ |
| $C_{4,2,2}$ | 0 | 0 | 3 | 3 | 0 | 2 | $-12$ |
| $C_{4,3,1}$ | 1 | 0 | 2 | 3 | 0 | 2 | $-14$ |

It is clear that $\Xi_{1|3,2^{n-2},1,1}$ and $\Theta_{1|3,2^{n-3},1,1}$ have the same structure as the $\alpha$ gauge parameters in (332a), so their components are identified with the extra fields in $e_{[2]}{}^{2^{n-1},1,1}$ and $e_{[2]}{}^{2^{n-2},1,1}$, respectively, or with the components of the gauge parameters that can be shifted away using gauge-for-gauge transformations.

The first relation (306), similar to (46) in eleven dimensions, was found in the non-linear realisation [19] and it can also be worked out by integrating (307) in the unfolded formulation of gravity and dual gravity. The two-form gauge parameters need to be related by a Hodge duality analogous to (47). At the next level, (316) is the duality relation between the dual graviton $h_{1,1}^{(0)}$ and the first higher dual graviton $h_{2,1,1}^{(1)}$, and it can be obtained by integrating the curvature relation (317). Up to pure gauge terms which are absorbed here by the introduction of extra fields, (316) matches the duality relation in the non-linear realisation – see equation (2.31) of [74]. Then we have equation (338) which is the duality relation between the first and second higher dual gravitons. Lastly we have an infinite family of duality relations (343a), (343b), and (343c) relating adjacent pairs of higher dual gravitons at arbitrarily high levels. All the duality relations hold exactly and not as equivalence relations up to pure gauge terms, and the parameters must obey constraints that relate them like the constraints in Section 5. Similar to our duality relations in eleven dimensions, there is no clear origin of $\Upsilon_{3,2^{n-1},1}$ in the unfolded equations and gauge symmetries, so they are not included in duality relations (343a), (343b), or (343c). We summarise this infinite tower of duality relations with a diagram:

$$\omega_{[1]}{}^2 \quad \longleftrightarrow \quad \omega^{(0)}{}_{[1]}{}^2 \quad \longleftrightarrow \quad \omega^{(1)}{}_{[1]}{}^{3,1} \quad \longleftrightarrow \quad \omega^{(2)}{}_{[2]}{}^{3,1,1} \quad \longleftrightarrow \quad \omega^{(3)}{}_{[2]}{}^{3,2,1,1} \quad \longleftrightarrow \quad \cdots \tag{349}$$

Let us conclude this section with a counting, similar to that of Section 5.5, of the extra fields that appear when unfolding $A_1^{+++}$ at low levels. In Table 5, the columns labelled $\mathcal{U}_{(2)}$, $\mathcal{U}_{(3)}$, $\mathcal{U}_{(4)}$, and $\mathcal{U}_{(\alpha^2=2)}$ count the unfolded spectra when we unfold: (1) fields with blocks of at most two antisymmetric indices (i.e. $h_{2,\ldots,2,1,1}$), (2) fields with blocks of at most three indices, (3) fields with blocks of at most four indices (i.e. all $A_1^{+++}$ fields), and (4) the fields corresponding

to real $A_1^{+++}$ roots. The spectra $\mathcal{U}_{(2)}$ and $\mathcal{U}_{(\alpha^2=2)}$ are the same here (since the first $\alpha^2 = 2$ field with a block of three indices is $B_{3,2,2,1,1,1,1,1}$ at level six) and up to level four they are both more than accounted for by $A_1^{+++}$ and its $\ell_2$ representation. The other two spectra $\mathcal{U}_{(3)}$ and $\mathcal{U}_{(4)}$ already surpass $A_1^{+++}$ and $\ell_2$ at level three: there is only one $C_{4,1,1}$ in $\ell_2$ but we need two. At the next level we find three fields in $\ell_2$ that are unused in $\mathcal{U}_{(4)}$ : $B_{3,2,2,1}$ , $B_{3,3,1,1}$, , and $B_{3,3,2}$ . We also find that more fields need to be added, namely one $C_{4,2,1,1}$ and one $C_{4,2,2}$ .

# 7 Frame-like actions for higher dual fields

## 7.1 Higher dual three-form in eleven dimensions

Another motivation for the introduction of higher connections in the unfolded formulation of various dynamical systems is that they are needed off-shell – at the level of actions. If it is true that, on-shell, these connections are either pure gauge or expressed as successive derivatives of the metric-like potential, then off-shell they are independent fields and instrumental in the construction of an action principle from which the dynamics follows. Until now, we have worked entirely at the level of unfolded equations of motion, but here we extend our analysis off-shell by completing the construction of an action principle for the $A_{9,3}$ field that was initiated in [23] along the lines of [80]. A parent action for $A_{9,3}$ was presented in [23] in terms of the (frame-like) variables of the unfolded formalism, and here we obtain a simple and transparent form of the higher dual action. This provides a direct link between the unfolded formulation of $A_{9,3}$ and the action presented here. We will use these techniques again in Section 7.2 to work out an analogous frame-like action for higher dual gravity in four dimensions.

Our starting point is the Maxwell three-form action in the Palatini formulation. There are two independent fields: a scalar-valued three-form field $A_{[3]}$ and a zero-form $F^{a[4]}$ valued in the rank-four antisymmetric Lorentz representation. The action is given by

$$S[A,F] = \int_{\mathcal{M}_{11}} \left( \mathrm{d}A_{[3]} + \frac{1}{2} h_{a[4]} F^{a[4]} \right) F^{b[4]} H_{b[4]} , \tag{350}$$

where $H_{b[4]}$ is the seven-form $\frac{1}{7!} \varepsilon_{b[4]c[7]} h^{c[7]}$ and $\mathcal{M}_{11}$ denotes our eleven-dimensional spacetime. The three-form action (350) is invariant under the usual gauge transformation

$$\delta A_{[3]} = \mathrm{d}\lambda_{[2]} , \tag{351}$$

and its equations of motion are given by

$$\mathrm{d}A_{[3]} + h_{a[4]} F^{a[4]} = 0 , \tag{352a}$$

$$\mathrm{d}F^{a[4]} H_{a[4]} = 0 . \tag{352b}$$

The second equation is equivalent to the on-shell relation

$$\mathrm{d}F^{a[4]} + h_b F^{a[4],b} = 0 , \tag{353}$$

where the zero-form $F^{a[4],b}$ transforms in the irreducible Lorentz representation $\mathbb{Y}[4,1]$ . To be precise, the equation of motion for $F^{a[4]}$ is (352a) and it implies $\partial_{[a} F_{bcde]} = 0$ , while the equation of motion for $A_{[3]}$ is the Maxwell equation $\partial^a F_{abcd} = 0$ . These two constraints are equivalent to the Lorentz irreducibility properties of $F^{a[4],b}$ . It is important that (352a) and (353) reproduce the first two unfolded equations for the three-form (56) and (59).

From the action (350), we construct the parent action

$$S[A, F, e, t] = \int_{\mathcal{M}_{11}} \left[ \left( dA_{[3]} + \frac{1}{2} h_{a[4]} F^{a[4]} + h_{a[3]} t_{[1]}{}^{a[3]} \right) F^{b[4]} H_{b[4]} + t_{[1]a[3]} de_{[9]}{}^{a[3]} \right], \quad (354)$$

featuring the one-form $t_{[1]}{}^{a[3]}$ and nine-form $e_{[9]}{}^{a[3]}$ along with the original fields $A_{[3]}$ and $F^{a[4]}$. This parent action is invariant under the following gauge transformations:

$$\delta A_{[3]} = d\lambda_{[2]} + h_{a[3]} \psi^{a[3]}, \quad (355a)$$

$$\delta F^{a[4]} = 0, \quad (355b)$$

$$\delta t_{[1]}{}^{a[3]} = d\psi^{a[3]}, \quad (355c)$$

$$\delta e_{[9]}{}^{a[3]} = d\tilde{\lambda}_{[8]}{}^{a[3]}. \quad (355d)$$

As for any $p$-form gauge theory, there are gauge-for-gauge (reducibility) transformations for $\lambda_{[2]}$ and $\tilde{\lambda}_{[8]}{}^{a[3]}$. Note that we do not identify the independent fields in the parent action (354) with the analogous objects in Section 4.1 since they do not transform in the same way. For example, none of the irreducible components of $e_{[9]}{}^{a[3]}$ can be shifted away from the parent action since it only transforms with the differential gauge parameter $\tilde{\lambda}_{[8]}{}^{a[3]}$.

The equations of motion of (354) lead to the on-shell relations

$$dA_{[3]} + h_{a[4]} F^{a[4]} + h_{a[3]} t_{[1]}{}^{a[3]} = 0, \quad (356a)$$

$$dF^{a[4]} + h_b F^{a[4],b} = 0, \quad (356b)$$

$$dt_{[1]}{}^{a[3]} = 0, \quad (356c)$$

$$de_{[9]}{}^{a[3]} - h^{a[3]b[7]} (*F)_{b[7]} = 0, \quad (356d)$$

where $(*F)_{b[7]} = \frac{1}{7!} \varepsilon_{b[7]c[4]} F^{c[4]}$. The field $e_{[9]}{}^{a[3]}$ effectively acts as a Lagrange multiplier for the constraint (356c) that is solved identically (albeit locally) by

$$t_{[1]}{}^{a[3]} = d\zeta^{a[3]}, \quad (357)$$

for some zero-form $\zeta^{a[3]}$. When the above expression for $t_{[1]}{}^{a[3]}$ is substituted inside the parent action, $t_{[1]}{}^{a[3]}$ effectively drops out from the action upon absorbing the three-form $h_{a[3]} \zeta^{a[3]}$ in a redefinition of $A_{[3]}$. The parent action (354) therefore reduces to the usual frame-like action for the three-form field (350). As a general rule, the actions obtained from the parent action upon the elimination of (generalised) auxiliary fields only propagate the degrees of freedom of the original field that is being dualised.

Now we will work out a frame-like action for the higher dual field $A_{9,3}$ that will propagate the degrees of freedom of the three-form by construction. The first observation is that $F_{a[4]}$ is auxiliary, so it can be expressed algebraically in terms of other fields through its equations of motion. The second observation is that the original three-form $A_{[3]}$ can be completely gauged away using the $\psi^{a[3]}$ parameter, leaving a residual gauge symmetry whereby the residual gauge parameters $\lambda_{[2]}$ and $\psi^{a[3]}$ are related as

$$\psi_{a[3]} = -\partial_{[a_1} \lambda_{a_2 a_3]}. \quad (358)$$

In this gauge, one eliminates the auxiliary field $F^{a[4]}$ through equation (356a), yielding

$$h_{a[4]} F^{a[4]} = -h_{a[3]} t_{[1]}{}^{a[3]}. \quad (359)$$

This means that $F_{a_1 a_2 a_3 a_4}$ is set equal to the antisymmetric component $t_{[a_1|a_2 a_3 a_4]}$ in the gauge where $A_{[3]}$ is zero. The parent action (354) now reduces to the dual action

$$S[e,t] = \int \left[ -\frac{12}{11!} \varepsilon_{m[11]} h^{m[11]} t^{[a_1|a_2 a_3 a_4]} t_{[a_1|a_2 a_3 a_4]} + t_{[1]a[3]} \mathrm{d} e_{[9]}{}^{a[3]} \right]. \tag{360}$$

Even now having eliminated the $A_{[3]}$ field, the nine-form $e_{[9]}{}^{a[3]}$ is again a Lagrange multiplier for the constraint $\mathrm{d} t_{[1]}{}^{a[3]} = 0$ that is solved by $t_{[1]}{}^{a[3]} = \mathrm{d} A^{a[3]}$, thereby resurrecting the original Maxwell three-form and its second-order action.

Our dual action can be written in the form

$$S[e,t] = \int \mathrm{d}^{11} x \left[ t_{a|b_1 b_2 b_3} \left( \frac{4!}{2} t^{[a|b_1 b_2 b_3]} + \frac{1}{9!} \varepsilon^{c_1 \cdots c_{10} a} \partial_{c_1} e_{c_2 \cdots c_{10}|}{}^{b_1 b_2 b_3} \right) \right]. \tag{361}$$

This is a first-order action principle for $e_{[9]}{}^{a[3]}$ which contains the higher dual field $A_{9,3}$ as one of its irreducible components. As explained in [23], the independent field $t_{a|b[3]}$ plays the role of the spin connection. Our dual action now takes the form $\langle \mathrm{d} e + \frac{1}{2} \sigma_- \omega | \omega \rangle$ of a generic frame-like action for mixed-symmetry fields [78]. This is more obvious if we define

$$\widetilde{\omega}_{a_1 a_2 a_3 | b} := t_{b | a_1 a_2 a_3}, \tag{362}$$

so that the roles of the form and frame indices are exchanged. Dualising the vector index as in (84) leads to the connection $\omega_{[3]}{}^{a[10]}$. Eliminating $t_{[1]}{}^{a[3]}$ produces a second-order action for $e_{[9]}{}^{a[3]}$ featuring all its irreducible components: the higher dual field $A_{9,3}$ and two extra fields $\widehat{A}_{10,2}$ and $\widehat{A}_{11,1}$. It was possible in Section 4.1 to gauge away these extra fields using algebraic shift symmetries. However, in [74] we observed that it is not possible to eliminate extra fields from higher dual action principles, so they must remain here too.

Looking at our parent action (354), notice that its last equation of motion (356d) can be expressed as the first unfolded equation (72a) for the $A_{9,3}$ field if we define

$$\omega_{[3]}{}^{a[3]b[7]} := \frac{1}{7!} h^{a[3]} \varepsilon^{b[7]c[4]} F_{c[4]}. \tag{363}$$

In components this is equivalent to

$$\omega_{a_1 a_2 a_3 | b_1 \cdots b_{10}} = -\frac{4!}{10!} \varepsilon_{b_1 \cdots b_{10}}{}^{c} F_{c a_1 a_2 a_3}, \tag{364}$$

as given in [23]. Thus the unfolded formulation of the higher dual field $A_{9,3}$ in Section 4.1 is compatible with our dual action principle (361) provided we interpret (364) as a first-order duality relation between the three-form $A_3$ and the $A_{9,3}$ field. The difference between (364) and (114) is that equation (364) is automatically gauge-invariant, while (114) was only gauge-invariant when $\alpha_{[2]}{}^{10}$ is constrained to be pure gauge-for-gauge.

Lastly, we will decompose $t_{[1]}{}^{a[3]}$ into irreducible components:

$$t_{a|b[3]} = F_{ab[3]} + Y_{b[3],a}, \qquad \square \otimes \begin{array}{c}\square\\\square\\\square\end{array} = \begin{array}{c}\square\\\square\\\square\\\square\end{array} \oplus \begin{array}{cc}\square&\square\\\square\\\square\end{array}. \tag{365}$$

As mentioned previously, in the gauge where we set the three-form $A_{[3]}$ to zero, there is still some residual gauge symmetry enjoyed by $t_{a|b[3]}$ such that it transforms as

$$\delta t_{a|b[3]} = -\partial_a \partial_{[b_1} \lambda_{b_2 b_3]}. \tag{366}$$

Consequently, the mixed-symmetry component $Y_{3,1}$ inherits all the gauge symmetry:

$$\delta F_{a[4]} = 0, \qquad \delta Y_{a[3],b} = -\partial_b \partial_{[a_1} \lambda_{a_2 a_3]}. \tag{367}$$

Our dual action can now be written as

$$S[e, F, Y] = \int \left[ -\frac{12}{11!} h^{b[11]} \varepsilon_{b[11]} F^{a[4]} F_{a[4]} + \left( F_{ab[3]} + Y_{b[3],a} \right) h^a \, \mathrm{d} e_{[9]}{}^{b[3]} \right]. \tag{368}$$

The equations of motion for this action are

$$\mathcal{E}_{[2]}{}^{a_1 a_2 a_3} := h_b \left( \mathrm{d} Y^{a_1 a_2 a_3, b} - \mathrm{d} F^{a_1 a_2 a_3 b} \right) = 0, \tag{369a}$$

$$\mathcal{E}^{a_1 a_2 a_3, b} := h^b \mathrm{d} e_{[9]}{}^{a_1 a_2 a_3} - h^{[b} \mathrm{d} e_{[9]}{}^{a_1 a_2 a_3]} = 0, \tag{369b}$$

$$\mathcal{E}^{a_1 a_2 a_3 a_4} := h^{[a_1} \mathrm{d} e_{[9]}{}^{a_2 a_3 a_4]} - \frac{4!}{11!} h^{b[11]} \varepsilon_{b[11]} F^{a_1 a_2 a_3 a_4} = 0. \tag{369c}$$

Solving the first equation of motion (369a) once again revives the Maxwell three-form and its second-order action principle. The second and third equations (369b) and (369c) are two orthogonal projections of the equation of motion for $t_{[1]}{}^{a[3]}$ in the dual action (361). The third equation on its own is just a projection of the first unfolded equation (72a) where we impose the duality relation (364). Separately, the second equation is an unusual differential equation for the $e_{[9]}{}^{a[3]}$ field. It is only by taking (369b) and (369c) together that we obtain the first unfolded equation (72a).

## 7.2 Higher dual gravity in four dimensions

We conclude by working out an action principle for the first higher dual graviton $h_{2,1,1}$ along the lines of Section 7.1. This will shed light on the gauge symmetries that we found using the metric-like formulation of higher dual gravity [74]. Our starting point is the frame-like action for dual gravity

$$S[e, \omega] = \int_{\mathcal{M}_4} \left( \mathrm{d} e_{[1]}{}^a + \frac{1}{2} h_b \, \omega_{[1]}{}^{ab} \right) \omega_{[1]}{}^{cd} H_{acd}, \tag{370}$$

where $\mathcal{M}_4$ is our four-dimensional space-time. The equations of motion of (370) are equivalent to the on-shell relations

$$\mathrm{d} e_{[1]}{}^a + h_b \, \omega_{[1]}{}^{ab} = 0, \tag{371a}$$

$$\mathrm{d} \omega_{[1]}{}^{ab} + h_{cd} \, C^{ab, cd} = 0. \tag{371b}$$

Moreover, (370) is invariant under the gauge symmetries

$$\delta e_{[1]}{}^a = \mathrm{d} \epsilon^a + h_b \, \alpha^{ab}, \tag{372a}$$

$$\delta \omega_{[1]}{}^{a[2]} = \mathrm{d} \alpha^{a[2]}. \tag{372b}$$

Importantly, these match the relations and gauge symmetries of the unfolded formulation of gravity in Section 3.2, and they are equivalent to those of dual gravity in four dimensions.

From the action principle (370), we construct the parent action

$$S[e, \omega, t, \tilde{e}] = \int_{\mathcal{M}_4} \left[ \left( \mathrm{d} e_{[1]}{}^a + \frac{1}{2} h_b \, \omega_{[1]}{}^{ab} + h_b \, t_{[1]}{}^{a,b} \right) \omega_{[1]}{}^{cd} H_{acd} + t_{[1]a,b} \, \mathrm{d} \tilde{e}_{[2]}{}^{a,b} \right], \tag{373}$$

for which the equations of motion are equivalent to the on-shell relations

$$\mathrm{d}e_{[1]}{}^{a} + h_{b}\,\omega_{[1]}{}^{ab} + h_{b}\,t_{[1]}{}^{a,b} = 0\,, \tag{374a}$$

$$\mathrm{d}\omega_{[1]}{}^{a[2]} + h_{b[2]}\,C^{a[2],b[2]} = 0\,, \tag{374b}$$

$$\mathrm{d}t_{[1]}{}^{a,b} = 0\,, \tag{374c}$$

$$\mathrm{d}\tilde{e}_{[2]}{}^{a,b} - 2\,h^{c(a}(*\omega)_{[1]}{}^{b)}{}_{c} = 0\,, \tag{374d}$$

where $(*\omega)_{[1]}{}^{a[2]} = \frac{1}{2}\varepsilon^{a[2]}{}_{b[2]}\,\omega_{[1]}{}^{b[2]}$. The parent action is invariant under the gauge symmetries

$$\delta e_{[1]}{}^{a} = \mathrm{d}\epsilon^{a} + h_{b}\,\alpha^{ab} + h_{b}\,\psi^{a,b}\,, \tag{375a}$$

$$\delta\omega_{[1]}{}^{a[2]} = \mathrm{d}\alpha^{a[2]}\,, \tag{375b}$$

$$\delta t_{[1]}{}^{a,b} = \mathrm{d}\psi^{a,b}\,, \tag{375c}$$

$$\delta\tilde{e}_{[2]}{}^{a,b} = \mathrm{d}\tilde{\epsilon}_{[1]}{}^{a,b} + 2\,h^{c(a}(*\alpha)^{b)}{}_{c}\,, \qquad \delta\tilde{\epsilon}_{[1]}{}^{a,b} = \mathrm{d}\tilde{\epsilon}^{a,b}\,, \tag{375d}$$

where $(*\alpha)^{a[2]} = \frac{1}{2}\varepsilon^{a[2]}{}_{b[2]}\,\alpha^{b[2]}$. The equation of motion (374c) implies that $t_{[1]}{}^{a,b} = \mathrm{d}\beta^{a,b}$ for some symmetric tensor $\beta^{a,b}$ and so $t_{[1]}{}^{a,b}$ can be gauged away using $\psi^{a,b}$ in (375c). As a result, the parent action (373) reduces to the usual frame-like Fierz-Pauli action (370).

The equations and gauge symmetries found here are very different to those of [77, 78] where Labastida fields with generic Young tableaux were unfolded. The Labastida field with tableau $\mathbb{Y}[2,1,1]$ in four dimensions has the same symmetry type as our higher dual graviton, but the towers of $p$-forms and their gauge symmetries in their unfolded formulations are not the same, and accordingly the propagating physical degrees of freedom are different.

Now we will derive from our parent action a frame-like action for the higher dual graviton. The symmetric and antisymmetric components of $e_{[1]}{}^{a}$ can be gauged away using $\alpha^{ab}$ and $\psi^{a,b}$, so the whole field $e_{[1]}{}^{a}$ can be shifted away leaving residual symmetry to be discussed below. In this gauge, imposing the equation of motion (374a) allows us to write

$$h_{b}\,t_{[1]}{}^{a,b} = -h_{b}\,\omega_{[1]}{}^{ab}\,, \tag{376}$$

and the parent action then reduces to

$$S = \int_{\mathcal{M}_{4}} \left( -\frac{1}{2}h_{b}\,\omega_{[1]}{}^{ab}\omega_{[1]}{}^{cd}H_{acd} + t_{[1]a,b}\,\mathrm{d}\tilde{e}_{[2]}{}^{a,b} \right)\,. \tag{377}$$

In the last term it should be understood that some components of $t$ are to be expressed in terms of $\omega$ according to (376). In other words, $t$ is not completely independent of $\omega$ in (377). The independent fields are $\tilde{e}$, the totally symmetric part of $t$, the totally antisymmetric part of $\omega$, and the mixed-symmetry parts of $t$ and $\omega$ which are the same due to (376).

As mentioned above, $\omega_{[1]}{}^{a[2]}$ and $t_{[1]}{}^{a,b}$ still enjoy some residual gauge symmetry that leaves the gauge $e_{[1]}{}^{a} = 0$ unchanged. The residual gauge parameters are related by

$$\alpha^{ab} = \partial^{[a}\epsilon^{b]}\,, \qquad \psi^{a,b} = -\partial^{(a}\epsilon^{b)}\,. \tag{378}$$

The components of these one-forms

$$\omega_{[1]}{}^{ab} = h_{c}\,\omega^{c|ab}\,, \qquad t_{[1]}{}^{a,b} = h_{c}\,t^{c|a,b}\,, \tag{379}$$

transform under this residual symmetry as

$$\delta\omega^{a|bc} = \partial^{a}\partial^{[b}\epsilon^{c]}\,, \qquad \delta t^{a|b,c} = -\partial^{a}\partial^{(b}\epsilon^{c)}\,. \tag{380}$$

Equation (378) implies that the gauge transformation (375d) becomes

$$\delta \tilde{e}_{[2]}{}^{a,b} = \mathrm{d}\tilde{\epsilon}_{[1]}{}^{a,b} - h^{c(a}\varepsilon^{b)}{}_{cij}\partial^i \epsilon^j \,, \tag{381}$$

or in components,

$$\delta \tilde{e}_{ab|c,d} = \partial_{[a}\epsilon_{b]|c,d} + \eta_{[a(c}\varepsilon_{d)b]ij}\partial^i \epsilon^j \,. \tag{382}$$

Now decompose $\tilde{e}_{[2]}{}^{a,b}$ and $\tilde{\epsilon}_{[1]}{}^{a,b}$ into irreducible fields and parameters:

$$\tilde{e}_{ab|c,d} = A_{ab,c,d} + \widehat{A}_{ab(c,d)} \,, \qquad\qquad \tilde{\epsilon}_{a|b,c} = \lambda_{a,b,c} + \mu_{a(b,c)} \,. \tag{383}$$

$$\square \otimes \square\square = \begin{array}{c}\square\square\square\\\square\end{array} \oplus \begin{array}{c}\square\square\\\square\end{array} \,, \qquad\qquad \square \otimes \square\square = \square\square\square \oplus \begin{array}{c}\square\square\\\square\end{array} \,. \tag{384}$$

These fields can be expressed in terms of $\widetilde{e}_{[2]}{}^{a,b}$ as

$$\widehat{A}_{abc,d} = \frac{3}{2}\tilde{e}_{[ab|c],d} \,, \qquad A_{ab,c,d} = \frac{1}{2}\tilde{e}_{ab|c,d} + \tilde{e}_{[a(c|d),b]} \,, \tag{385}$$

and we find that the gauge transformations of the irreducible fields are

$$\delta A_{ab,c,d} = \partial_{[a}\lambda_{b],c,d} + \frac{1}{4}\partial_{[a}\mu_{b](c,d)} - \frac{3}{8}\partial_{(c|}\mu_{ab,|d)} + \frac{1}{2}\eta_{[a(c}\varepsilon_{d)b]ij}\partial^i\epsilon^j + \frac{1}{4}\eta_{cd}\varepsilon_{abij}\partial^i\epsilon^j \,, \tag{386a}$$

$$\delta \widehat{A}_{abc,d} = \frac{9}{8}\partial_{[a}\mu_{bc],d} - \frac{3}{4}\eta_{d[a}\varepsilon_{bc]ij}\partial^i\epsilon^j \,. \tag{386b}$$

Up to factors, these are precisely the gauge transformations of the higher dual graviton $A_{2,1,1}$ and the extra field $\widehat{A}_{3,1}$ in the metric-like action for higher dual gravity [74]. The extra field is once again crucial to the propagation of the correct degrees of freedom. The dual graviton transforms with a vector gauge parameter $\epsilon^a$ and it was unexpected that the higher dual graviton would also transform with it. However, in this section we have found that $\epsilon^a$ arises naturally as a consequence of residual gauge symmetry.

# 8  Conclusion

In this paper we applied the unfolded formalism to the fields in $E_{11}$. We proposed first-order, gauge-invariant, on-shell duality relations for the infinite set of higher dual fields in the $E_{11}$ non-linear realisation which contain the dynamical degrees of freedom. These relations are all expressed in terms of the first-order connections that are used in the unfolded formulation of each higher dual field in the gravity, three-form, and six-form sectors of the theory. Although one can formulate the duality relations as equivalence relations, it is interesting to formulate them as conventional equations that are gauge-invariant. The unfolded formalism introduces extra fields into the duality relations which ensures that they are gauge-covariant provided that we impose an infinite tower of gauge parameter constraints that were obtained in this paper.

Taking derivatives of these first-order duality relations led to an infinite number of duality relations between the curvatures associated with higher dual fields in $E_{11}$. Working on-shell, we found that the constraints on these curvatures are exchanged, corresponding to the usual exchange between the equations of motion and Bianchi identities between dual fields. Taking traces of these curvature relations led to the linearised equations of motion for all higher dual fields in the $E_{11}$ theory. For dual fields at higher levels, there are more and more independent equations of motion (i.e. higher trace constraints on the curvature) for one and the same field. However, there is only one relation, an algebraic redefinition in fact, between the curvatures

of any pair of dual fields. We have shown how to integrate these curvature relations to find first-order duality relations, where the precise meaning of the extra fields becomes apparent.

There are two sources of ambiguity when applying the unfolded formalism to the non-linear realisation of $E_{11}$ . The only degrees of freedom in the theory are those of the graviton and the three-form, and these are related to an infinite number of dual fields by first-order duality relations. It is in this way that the infinite number of duality symmetries in $E_{11}$ is realised. While one must unfold all these fields associated with the dynamical degrees of freedom, it is not so clear which other fields in $E_{11}$ need to be unfolded. Should one, for example, unfold the fields with one block of ten antisymmetric indices which lead to the gauged supergravities? The prototypical example is the $B_{10,1,1}$ field at level four which leads to Romans theory. The other ambiguity stems from the fact that one could use fields in $E_{11}$ with blocks of ten or eleven indices in the unfolding process rather than introducing extra fields.

The origin of the extra fields in the theory was discussed in Section 5.5. An extension of the non-linear realisation featuring these extra fields found in the unfolded formalism needs to be compatible with $E_{11}$ symmetry, so it made sense to search inside highest weight representations of $E_{11}$ . If only the fields associated with the dynamical degrees of freedom are unfolded, then the $\ell_2$ representation by itself is able to provide all the extra fields. The lowest level field that it contains is a nine-form, precisely the field that needs to accompany the dual graviton in its duality relations.

In Section 6, the non-linear realisation of $A_1^{+++}$ was analysed in the same way. We unfolded the fields up to level three, all the higher dual fields $h_{2,...,2,1,1}$ at arbitrarily high levels, and we wrote down first-order duality relations between them leading to linearised equations of motion for all the higher dual fields. We then integrated all these equations to find the most general first-order duality relations between the fields. Then we discussed the origin of the extra fields and, similar to the $E_{11}$ case, we observed that the $\ell_2$ representation of $A_1^{+++}$ is the natural candidate for a source of extra fields. Duality relations for the recently constructed non-linear realisation of $K_{27} = D_{24}^{+++}$ [81] were quickly proposed in Appendix B. A consistent extension of the non-linear realisations of $E_{11}$, $A_1^{+++}$ , and $K_{27}$ , featuring the $\ell_2$ representations of each algebra, should contain the duality relations that we gave in this paper.

First-order actions have been worked out in Section 7 for the higher dual three-form $A_{9,3}$ in $E_{11}$ and the higher dual graviton $h_{2,1,1}$ in $A_1^{+++}$. A second-order 'metric-like' higher dual action for the latter was previously given in [74], where we obtained intertwined gauge transformations between the higher dual graviton and the extra field that came with it. In the present paper we have shown that these intertwined gauge transformations emerge in a very elementary way due to residual gauge symmetry.

In this paper we have not used $E_{11}$ symmetry to formulate the dynamics. Rather, we have taken the $E_{11}$ fields and worked out their unfolded formulations. It would be interesting to have an unfolded formalism with $E_{11}$ symmetries built into it so that the resulting equations and gauge transformations would automatically respect $E_{11}$ symmetry. This would necessarily involve extending space-time to the generalised $E_{11}$ space-time [9] rather than the usual eleven dimensions that we have considered here.

First-order duality relations for $E_{11}$ fields were also proposed in [18, 48, 57] and one may ask whether or not there is a link between those and the relations proposed in the present paper. Moreover, fully non-linear equations of motion and duality relations were neither considered nor constructed here. The non-linear dual graviton equation of motion was obtained in terms of the components of the $E_{11}$ Maurer-Cartan form in [14]. It would be interesting to extend the infinite set of the duality relations proposed here to the non-linear level. One way to do this would be to use an $E_{11}$-invariant unfolding formalism since they would automatically include all the extra fields. In the full non-linear theory, a non-linear extension of the first-order connections should feature in the duality relations. For example, the field

strength $F_7$ of the six-form field $A_6$ would be replaced by $G_7 := F_7 - \frac{1}{2}A_3 F_4$ (with seven indices antisymmetrised), and this non-linearity is built into the $E_{11}$ non-linear realisation from the start. For example, the component of the $E_{11}$ Maurer-Cartan form at level two is $G_7$. A possible non-linear completion of our linearised analysis should incorporate all $E_{11}$ Maurer-Cartan form components and all the necessary extra fields into our unfolded equations.

## Acknowledgments

N.B. and J.A.O. wish to thank Martin Cederwall, Jakob Palmkvist, Zhenya Skvortsov, and Per Sundell for useful discussions. J.A.O. would like to thank P.W. and P.P.C. for hospitality at King's College London across this project.

**Funding information** The work of J.A.O. was supported by the Fonds National de la Recherche Scientifique (FNRS), grant number FC 43791. P.W. would like to thank the STFC, grant numbers ST/P000258/1 and ST/T000759/1, for support. The work of N.B. was partially supported by the F.R.S.-FNRS PDR Grant No. T.0022.19 "Fundamental issues in extended gravity", Belgium.

## A   Representations of $E_{11}$

For the convenience of the reader, here we give tables of generators of $E_{11}$ and some of its most important representations, computed using SimpLie [93]. Generators of the $i^{\text{th}}$ fundamental representation $\ell_i$ are obtained as follows. First we extend $E_{11}$ to the algebra $E_{11}^{(i)}$ by attaching a new vertex denoted $*$ to the $i^{\text{th}}$ vertex of the $E_{11}$ Dynkin diagram using a single edge. Then we must restrict the Kac label of the new vertex to be equal to one. In other words, a generic generator in $E_{11}^{(i)}$ is associated with a root $\alpha = \sum_{i=1}^{11} k_i \alpha_i + k_* \alpha_*$ and then the integer coefficient $k_*$ must be fixed equal to one, so that in the decomposition $E_{11}^{(i)} \to E_{11}$ we consider level one. Taking the usual decomposition $E_{11} \to \text{GL}(11)$ leads to generators at each level written as $A_{10}$ tensors. This procedure can be used to work out more general highest weight representations by adding more vertices and restricting the simple root coefficients to the Dynkin labels of the representation being considered. The $\mu$ column gives the multiplicity of each generator.

Table 6: The adjoint representation of $E_{11}$ from level zero to level six.

| $l$ | $A_{10}$ weight | $E_{11}$ root $\alpha$ | $\alpha^2$ | $\mu$ | field |
|---|---|---|---|---|---|
| 0 | [1, 0, 0, 0, 0, 0, 0, 0, 0, 1] | (1, 1, 1, 1, 1, 1, 1, 1, 1, 1, 0) | 2 | 1 | $h_a{}^b$ |
| 0 | [0, 0, 0, 0, 0, 0, 0, 0, 0, 0] | (0, 0, 0, 0, 0, 0, 0, 0, 0, 0, 0) | 0 | 1 | |
| 1 | [0, 0, 0, 0, 0, 0, 0, 1, 0, 0] | (0, 0, 0, 0, 0, 0, 0, 0, 0, 1) | 2 | 1 | $A_3$ |
| 2 | [0, 0, 0, 0, 1, 0, 0, 0, 0, 0] | (0, 0, 0, 0, 0, 1, 2, 3, 2, 1, 2) | 2 | 1 | $A_6$ |
| 3 | [0, 0, 1, 0, 0, 0, 0, 0, 0, 1] | (0, 0, 0, 1, 2, 3, 4, 5, 3, 1, 3) | 2 | 1 | $h_{8,1}$ |
| 4 | [0, 1, 0, 0, 0, 0, 0, 1, 0, 0] | (0, 0, 1, 2, 3, 4, 5, 6, 4, 2, 4) | 2 | 1 | $A_{9,3}$ |
| 4 | [1, 0, 0, 0, 0, 0, 0, 0, 0, 2] | (0, 1, 2, 3, 4, 5, 6, 7, 4, 1, 4) | 2 | 1 | $B_{10,1,1}$ |
| 4 | [0, 0, 0, 0, 0, 0, 0, 0, 0, 1] | (1, 2, 3, 4, 5, 6, 7, 8, 5, 2, 4) | −2 | 1 | $C_{11,1}$ |
| 5 | [0, 1, 0, 0, 1, 0, 0, 0, 0, 0] | (0, 0, 1, 2, 3, 5, 7, 9, 6, 3, 5) | 2 | 1 | $A_{9,6}$ |
| 5 | [1, 0, 0, 0, 0, 0, 1, 0, 0, 1] | (0, 1, 2, 3, 4, 5, 6, 8, 5, 2, 5) | 2 | 1 | $B_{10,4,1}$ |
| 5 | [0, 0, 0, 0, 0, 0, 0, 1, 0, 1] | (1, 2, 3, 4, 5, 6, 7, 8, 5, 2, 5) | 0 | 1 | $C_{11,3,1}$ |
| 5 | [0, 0, 0, 0, 0, 0, 1, 0, 0, 0] | (1, 2, 3, 4, 5, 6, 7, 9, 6, 3, 5) | −2 | 1 | $C_{11,4}$ |
| 6 | [0, 1, 1, 0, 0, 0, 0, 0, 0, 1] | (0, 0, 1, 3, 5, 7, 9, 11, 7, 3, 6) | 2 | 1 | $h_{9,8,1}$ |

| 6 | [1, 0, 0, 0, 1, 0, 0, 0, 1, 0] | (0, 1, 2, 3, 4, 6, 8, 10, 6, 3, 6) | 2 | 1 | $B_{10,6,2}$ |
| 6 | [1, 0, 0, 1, 0, 0, 0, 0, 0, 1] | (0, 1, 2, 3, 5, 7, 9, 11, 7, 3, 6) | 0 | 1 | $B_{10,7,1}$ |
| 6 | [1, 0, 1, 0, 0, 0, 0, 0, 0, 0] | (0, 1, 2, 4, 6, 8, 10, 12, 8, 4, 6) | −2 | 1 | $B_{10,8}$ |
| 6 | [0, 0, 0, 0, 0, 0, 1, 1, 0, 0] | (1, 2, 3, 4, 5, 6, 7, 9, 6, 3, 6) | 2 | 1 | $C_{11,4,3}$ |
| 6 | [0, 0, 0, 0, 0, 1, 0, 0, 0, 2] | (1, 2, 3, 4, 5, 6, 8, 10, 6, 2, 6) | 2 | 1 | $C_{11,5,1,1}$ |
| 6 | [0, 0, 0, 0, 1, 0, 0, 0, 0, 1] | (1, 2, 3, 4, 5, 7, 9, 11, 7, 3, 6) | −2 | 2 | $C_{11,6,1}$ |
| 6 | [0, 0, 0, 1, 0, 0, 0, 0, 0, 0] | (1, 2, 3, 4, 6, 8, 10, 12, 8, 4, 6) | −4 | 1 | $C_{11,7}$ |

Table 7: The $\ell_1$ representation of $E_{11}$ from level zero to level four.

| $l$ | $A_{10}$ weight | $E_{11}^{(1)}$ root $\alpha$ | $\alpha^2$ | $\mu$ | coordinate |
|---|---|---|---|---|---|
| 0 | [1, 0, 0, 0, 0, 0, 0, 0, 0, 0] | (0, 0, 0, 0, 0, 0, 0, 0, 0, 0, 1) | 2 | 1 | $x^a$ |
| 1 | [0, 0, 0, 0, 0, 0, 0, 0, 1, 0] | (1, 1, 1, 1, 1, 1, 1, 1, 0, 0, 1, 1) | 2 | 1 | $z_2$ |
| 2 | [0, 0, 0, 0, 0, 1, 0, 0, 0, 0] | (1, 1, 1, 1, 1, 1, 2, 3, 2, 1, 2, 1) | 2 | 1 | $z_5$ |
| 3 | [0, 0, 0, 1, 0, 0, 0, 0, 0, 1] | (1, 1, 1, 1, 2, 3, 4, 5, 3, 1, 3, 1) | 2 | 1 | $z_{7,1}$ |
| 3 | [0, 0, 1, 0, 0, 0, 0, 0, 0, 0] | (1, 1, 1, 2, 3, 4, 5, 6, 4, 2, 3, 1) | 0 | 1 | $z_8$ |
| 4 | [0, 0, 1, 0, 0, 0, 0, 1, 0, 0] | (1, 1, 1, 2, 3, 4, 5, 6, 4, 2, 4, 1) | 2 | 1 | $z_{8,3}$ |
| 4 | [0, 1, 0, 0, 0, 0, 0, 0, 0, 2] | (1, 1, 2, 3, 4, 5, 6, 7, 4, 1, 4, 1) | 2 | 1 | $z_{9,1,1}$ |
| 4 | [0, 1, 0, 0, 0, 0, 0, 0, 1, 0] | (1, 1, 2, 3, 4, 5, 6, 7, 4, 2, 4, 1) | 0 | 1 | $z_{9,2}$ |
| 4 | [1, 0, 0, 0, 0, 0, 0, 0, 0, 1] | (1, 2, 3, 4, 5, 6, 7, 8, 5, 2, 4, 1) | −2 | 2 | $z_{10,1}$ |
| 4 | [0, 0, 0, 0, 0, 0, 0, 0, 0, 0] | (2, 3, 4, 5, 6, 7, 8, 9, 6, 3, 4, 1) | −4 | 1 | $z_{11}$ |

Table 8: The $\ell_2$ representation of $E_{11}$ from level zero to level three.

| $l$ | $A_{10}$ weight | $E_{11}^{(2)}$ root $\alpha$ | $\alpha^2$ | $\mu$ | field |
|---|---|---|---|---|---|
| 0 | [0, 1, 0, 0, 0, 0, 0, 0, 0, 0] | (0, 0, 0, 0, 0, 0, 0, 0, 0, 0, 1) | 2 | 1 | $\phi_9$ |
| 1 | [1, 0, 0, 0, 0, 0, 0, 0, 1, 0] | (0, 1, 1, 1, 1, 1, 1, 1, 0, 0, 1, 1) | 2 | 1 | $\phi_{10,2}$ |
| 1 | [0, 0, 0, 0, 0, 0, 0, 0, 0, 1] | (1, 2, 2, 2, 2, 2, 2, 1, 0, 1, 1) | 0 | 1 | $\phi_{11,1}$ |
| 2 | [1, 0, 0, 0, 0, 1, 0, 0, 0, 0] | (0, 1, 1, 1, 1, 1, 2, 3, 2, 1, 2, 1) | 2 | 1 | $\phi_{10,5}$ |
| 2 | [0, 0, 0, 0, 0, 0, 0, 1, 0, 1] | (1, 2, 2, 2, 2, 2, 2, 1, 0, 2, 1) | 2 | 1 | $\phi_{11,3,1}$ |
| 2 | [0, 0, 0, 0, 0, 0, 1, 0, 0, 0] | (1, 2, 2, 2, 2, 2, 3, 2, 1, 2, 1) | 0 | 1 | $\phi_{11,4}$ |
| 3 | [1, 0, 0, 1, 0, 0, 0, 0, 0, 1] | (0, 1, 1, 1, 2, 3, 4, 5, 3, 1, 3, 1) | 2 | 1 | $\phi_{10,7,1}$ |
| 3 | [1, 0, 1, 0, 0, 0, 0, 0, 0, 0] | (0, 1, 1, 2, 3, 4, 5, 6, 4, 2, 3, 1) | 0 | 1 | $\phi_{10,8}$ |
| 3 | [0, 0, 0, 0, 0, 1, 0, 0, 1, 0] | (1, 2, 2, 2, 2, 3, 4, 2, 1, 3, 1) | 2 | 1 | $\phi_{11,5,2}$ |
| 3 | [0, 0, 0, 0, 1, 0, 0, 0, 0, 1] | (1, 2, 2, 2, 3, 4, 5, 3, 1, 3, 1) | 0 | 2 | $\phi_{11,6,1}$ |
| 3 | [0, 0, 0, 1, 0, 0, 0, 0, 0, 0] | (1, 2, 2, 2, 3, 4, 5, 6, 4, 2, 3, 1) | −2 | 2 | $\phi_{11,7}$ |

Table 9: The $\ell_{10}$ representation of $E_{11}$ from level zero to level two.

| $l$ | $A_{10}$ weight | $E_{11}^{(10)}$ root $\alpha$ | $\alpha^2$ | $\mu$ | field |
|---|---|---|---|---|---|
| 0 | [0, 0, 0, 0, 0, 0, 0, 0, 0, 1] | (0, 0, 0, 0, 0, 0, 0, 0, 0, 0, 1) | 2 | 1 | $\phi_{11,1}$ |
| 1 | [0, 0, 0, 0, 0, 1, 0, 0, 0, 0] | (0, 0, 0, 0, 0, 0, 1, 1, 1, 1, 1) | 2 | 1 | $\phi_{11,4}$ |
| 2 | [0, 0, 0, 0, 1, 0, 0, 0, 0, 1] | (0, 0, 0, 0, 1, 2, 3, 2, 1, 2, 1) | 2 | 1 | $\phi_{11,6,1}$ |
| 2 | [0, 0, 0, 1, 0, 0, 0, 0, 0, 0] | (0, 0, 0, 0, 1, 2, 3, 4, 3, 2, 2, 1) | 0 | 1 | $\phi_{11,7}$ |

# B  Unfolding $K_{27}$ at low levels

In this appendix we will briefly sketch the unfolding of dual fields in the non-linear realisation of $K_{27} = D_{24}^{+++}$ which was recently constructed in [81]. This algebra was conjectured a long time ago to be the symmetry of the closed bosonic string [1]. The non-linear realisation features the graviton $h_{1,1}$ and the dilaton $\phi$ at level $(0,0)$, the Kalb-Ramond two-form $A_2$ at level $(0,1)$, and its electromagnetic dual $A_{22}$ at level $(1,0)$. The dual graviton $h_{23,1}$ and dual dilaton $\phi_{24}$ are found at level $(1,1)$, and among the infinite number of fields at higher levels the theory contains an all the higher dual fields $h_{24,\dots,24,23,1}$, $\phi_{24,\dots,24}$, $A_{24,\dots,24,2}$, and $A_{24,\dots,24,22}$. The level is a pair of integers since $K_{27}$ is decomposed with respect to its $A_{25}$ subalgebra, and the pair of Kac labels associated with the remaining two vertices in the Dynkin diagram become the level. The non-linear realisation contains duality relations between the graviton, dilaton, two-form, and their electromagnetic duals. Equations of motion for these three fields were computed by taking derivatives of the duality relations, and they were separately derived from $K_{27}$ symmetry.

The unfolded formulation of the dual graviton is essentially the same as that of Section 3.3. The first two unfolded equations of the dilaton and the Kalb-Ramond field are

$$\mathrm{d}\phi + h_a F^a = 0\,, \qquad\qquad \mathrm{d}A_{[2]} + h_{a[3]} F^{a[3]} = 0\,, \tag{B.1}$$

$$\mathrm{d}F^a + h_b F^{a,b} = 0\,, \qquad\qquad \mathrm{d}F^{a[3],b} + h_b F^{a[3]} = 0\,, \tag{B.2}$$

and they take the same form as the unfolded equations (56) and (59) for the three-form in eleven dimensions. The zero-forms $F_a$ and $F_{a[3]}$ are the field strengths of the dilaton $\phi$ and the two-form $A_{a_1 a_2}$, and they are the first of two infinite towers of zero-forms that one needs in order to write down all the unfolded equations:

$$\mathcal{T}(\phi) = \{F_{1^{n+1}}^{(n)} \mid n \in \mathbb{N}\} = \{F_1, F_{1,1}, F_{1,1,1}, \dots\}\,, \tag{B.3}$$

$$\mathcal{T}(A_2) = \{F_{3,1^n}^{(n)} \mid n \in \mathbb{N}\} = \{F_3, F_{3,1}, F_{3,1,1}, \dots\}\,. \tag{B.4}$$

Solving the higher unfolded equations, one finds that these zero-forms can be expressed as

$$F_{a_1,\dots,a_n}^{(n)} \propto \partial_{a_1} \cdots \partial_{a_n} \phi\,, \qquad F_{a_1 a_2 a_3, b_1,\dots,b_n}^{(n)} \propto \partial_{b_1} \cdots \partial_{b_n} \partial_{[a_1} A_{a_2 a_3]}\,. \tag{B.5}$$

The first unfolded equations for the dual dilaton $\phi_{24}$ and dual Kalb-Ramond field $A_{22}$ are

$$\mathrm{d}\phi_{[24]} + h_{a[25]} F^{a[25]} = 0\,, \qquad \mathrm{d}A_{[22]} + h_{a[23]} F^{a[23]} = 0\,, \tag{B.6}$$

where $F_{a[25]}$ and $F_{a[23]}$ are the first zero-forms in the zero-form towers

$$\mathcal{T}(\phi_{24}) = \{F_{25,1^n}^{(n)} \mid n \in \mathbb{N}\}\,, \qquad \mathcal{T}(A_{22}) = \{F_{23,1^n}^{(n)} \mid n \in \mathbb{N}\}\,. \tag{B.7}$$

So far, we have found first-order variables are $F_a$ and $F_{a[25]}$ in the dilaton sector, and $F_{a[3]}$ and $F_{a[23]}$ in the two-form sector. The obvious duality relations that we can write down are

$$F_a \propto \varepsilon_a{}^{b_1 \cdots b_{25}} F_{b_1 \cdots b_{25}}\,, \qquad F_{a_1 a_2 a_3} \propto \varepsilon_{a_1 a_2 a_3}{}^{b_1 \cdots b_{23}} F_{b_1 \cdots b_{23}}\,. \tag{B.8}$$

Taking derivatives leads to the linearised equations of motion

$$\partial^a F_a = 0\,, \qquad \partial^a F_{a b_1 \cdots b_{24}} = 0\,, \qquad \partial^a F_{a b_1 b_2 b_3} = 0\,, \qquad \partial^a F_{a b_1 \cdots b_{22}} = 0\,. \tag{B.9}$$

At the linearised level, these duality relations and equations of motion match those of the $K_{27}$ non-linear realisation [81].

**Higher dual dilatons.** At higher levels one might want to unfold the $n^{\text{th}}$ higher dual dilaton $\phi_{24^n}^{(n)} = \phi_{24,\dots,24}^{(n)}$, and for this purpose we introduce a tower of objects

$$\left\{ e_{[24]}^{24^{n-1}}, \quad \omega_{[24]}^{25,24^{n-2}}, \quad X_{[24]}^{25^2,24^{n-3}}, \quad \dots, \quad X_{[24]}^{25^{n-1}}, \quad C^{25^n}, \quad \dots \right\}. \tag{B.10}$$

We propose first-order duality relations for the higher dual dilaton fields:

$$\omega_{a[24]|b[25]}^{(1)} \propto \varepsilon_{b[25]}{}^p F_{pa[24]}, \tag{B.11a}$$

$$\omega_{a[24]|b[25],c[24]}^{(2)} \propto \varepsilon_{b[25]}{}^p \omega_{a[24]|pc[24]}^{(1)}, \tag{B.11b}$$

$$\omega_{a[24]|b[25],c[24],d[24]}^{(3)} \propto \varepsilon_{b[25]}{}^p \omega_{a[24]|pc[24],d[24]}^{(2)}, \tag{B.11c}$$

$$\vdots$$

$$\omega_{a[24]|b[25],c[24],d^1[24],\dots,d^{n-2}[24]}^{(n)} \propto \varepsilon_{b[25]}{}^p \omega_{a[24]|pc[24],d^1[24],\dots,d^{n-2}[24]}^{(n-1)}. \tag{B.11d}$$

Taking derivatives leads to the expected gauge-invariant on-shell curvature relations between zero-forms $F_{25^{n+1}}^{(0)} \in \mathcal{T}(\phi_{24^n}^{(n)})$ and $F_{25,1^n}^{(n)} \in \mathcal{T}(\phi_{24})$ of the form

$$F_{a^1[25],\dots,a^n[25],b[25]}^{(0)} \propto \varepsilon_{a^1[25]}{}^{p_1} \cdots \varepsilon_{a^n[25]}{}^{p_n} F_{b[25],p_1,\dots,p_n}^{(n)}. \tag{B.12}$$

The trace and over-antisymmetrisation constraints on $F_{25,1^n}^{(n)}$ lead to the linearised equations of motion for the higher dual fields, expressed as trace constraints on the primary zero-form:

$$(\text{Tr}_{i,j})^{25}(F_{25^n}) = 0, \qquad 1 \le i < j \le n. \tag{B.13}$$

**Higher dual Kalb-Ramond fields.** For the first higher dual fields $A_{24,2}^{(1)}$ and $A_{24,22}^{(1)}$ in the two-form sector, we introduce their corresponding towers of unfolded variables

$$\left\{ e_{[24]}^2, \quad \omega_{[2]}^{25}, \quad C^{25,2}, \quad \dots \right\}, \tag{B.14}$$

$$\left\{ e_{[24]}^{22}, \quad \omega_{[22]}^{25}, \quad C^{25,22}, \quad \dots \right\}. \tag{B.15}$$

Similarly, for the higher dual fields $A_{24,\dots,24,2}^{(n)}$ and $A_{24,\dots,24,22}^{(n)}$ at higher levels, we introduce

$$\left\{ e_{[24]}^{24^{n-1},2}, \quad \omega_{[24]}^{25,24^{n-2},2}, \quad \dots \right\}, \tag{B.16}$$

$$\left\{ e_{[24]}^{24^{n-1},22}, \quad \omega_{[24]}^{25,24^{n-2},22}, \quad \dots \right\}. \tag{B.17}$$

We propose duality relations for the higher dual Kalb-Ramond fields $A_{24^n,2}^{(n)}$ of the form

$$\omega_{a[2]|b[25]}^{(1)} \propto \varepsilon_{b[25]}{}^p F_{pa[2]}, \tag{B.18a}$$

$$\omega_{a[24]|b[25],c[2]}^{(2)} \propto \varepsilon_{b[25]}{}^p \omega_{c[2]|pa[24]}^{(1)}, \tag{B.18b}$$

$$\omega_{a[24]|b[25],c[24],d[2]}^{(3)} \propto \varepsilon_{b[25]}{}^p \omega_{a[24]|pc[24],d[2]}^{(2)}, \tag{B.18c}$$

$$\vdots$$

$$\omega_{a[24]|b[25],c[24],d^1[24],\dots,d^{n-3}[24],e[2]}^{(n)} \propto \varepsilon_{b[25]}{}^p \omega_{a[24]|pc[24],d^1[24],\dots,d^{n-3}[24],e[2]}^{(n-1)}. \tag{B.18d}$$

Similarly, our relations for the 'magnetic' higher dual Kalb-Ramond fields $A^{(n)}_{24^n,22}$ are

$$\omega^{(1)}_{a[22]|b[25]} \propto \varepsilon_{b[25]}{}^p F_{pa[22]}, \tag{B.19a}$$

$$\omega^{(2)}_{a[24]|b[25],c[22]} \propto \varepsilon_{b[25]}{}^p \omega^{(1)}_{c[22]|pa[24]}, \tag{B.19b}$$

$$\omega^{(3)}_{a[24]|b[25],c[24],d[22]} \propto \varepsilon_{b[25]}{}^p \omega^{(2)}_{a[24]|pc[24],d[22]}, \tag{B.19c}$$

$$\vdots$$

$$\omega^{(n)}_{a[24]|b[25],c[24],d^1[24],\ldots,d^{n-3}[24],e[22]} \propto \varepsilon_{b[25]}{}^p \omega^{(n-1)}_{a[24]|pc[24],d^1[24],\ldots,d^{n-3}[24],e[22]}. \tag{B.19d}$$

As before, taking derivatives leads to relations between $F^{(0)}_{25^n,3} \in \mathcal{T}(A^{(n)}_{24^n,2})$ and $F^{(n)}_{3,1^n} \in \mathcal{T}(A_2)$:

$$F^{(0)}_{a^1[25],\ldots,a^n[25],b[25]} \propto \varepsilon_{a^1[25]}{}^{p_1} \cdots \varepsilon_{a^n[25]}{}^{p_n} F^{(n)}_{b[25],p_1,\ldots,p_n}, \tag{B.20}$$

and also between $F^{(0)}_{25^n,23} \in \mathcal{T}(A^{(n)}_{24^n,22})$ and $F^{(n)}_{23,1^n} \in \mathcal{T}(A_{22})$:

$$F^{(0)}_{a^1[25],\ldots,a^n[25],b[25]} \propto \varepsilon_{a^1[25]}{}^{p_1} \cdots \varepsilon_{a^n[25]}{}^{p_n} F^{(n)}_{b[25],p_1,\ldots,p_n}. \tag{B.21}$$

The irreducibility properties of the zero-forms in $\mathcal{T}(A_2)$ and $\mathcal{T}(A_{22})$ lead to the linearised equations of motion for the all higher dual fields $A^{(n)}_{24^n,2}$ in the $K_{27}$ non-linear realisation:

$$(\text{Tr}_{i,j})^{25}(F_{25^n,3}) = 0, \qquad (\text{Tr}_{i,n+1})^3(F_{25^n,3}) = 0, \qquad 1 \leq i < j \leq n. \tag{B.22}$$

Similarly, the higher dual fields $A^{(n)}_{24^n,22}$ obey the linearised equations

$$(\text{Tr}_{i,j})^{25}(F_{25^n,23}) = 0, \qquad (\text{Tr}_{i,n+1})^3(F_{25^n,23}) = 0, \qquad 1 \leq i < j \leq n. \tag{B.23}$$

As in Section 5, integrating up these equations of motion would lead to the most general first-order on-shell duality relations which coincide with those that we presented in this appendix. Given the relevance of $K_{27}$ symmetry to effective theories of closed strings, one might like to investigate the role of these higher duality symmetries in the full theory.

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
