# Peer review of "Unfolding $E_{11}$"

_SciPost Physics, doi:SciPost Phys. 18, 149 (2025)_

## Round 1 · Referee Report · Anonymous (Referee 1) · 2025-1-31

Strengths

  1. fascinating subject matter generally
  2. interesting open question addressed

Weaknesses

  1. introduction too roundabout, not to the point
  2. presentation of results tends to be somewhat imprecise

Report

The aim of the article, first alluded to on its p 6, is to clarify aspects of the (Hodge-)duality structure of fields in and beyond the E11-formulation of 11D supergravity by "proposing" suitable duality relations in an "unfolded" formulation of the (expected) linearized equations of  motions for these fields.

The bulk of the article goes iteratively through low-level examples of the field components, each time (1.) "introducing a set of variables", then (2.) stating "unfolded" equations of motion for them  and (3.) "proposing" duality relations. My understanding is that thisn each case the implied second-order EoMs are found to match expectations obtained from elsewhere, which thus justifies, a posteriori, the Ansatz of "variables" and the proposed duality relations. on them.

But the precise rules of the game which the authors play and the conclusions which they draw could be stated more explicitly and more up-front: Sections 1 & 2 are mainly a broad review of the E11 program in general, of the kind one might write for a grant proposal, rather than a concrete motivation of and introduction to the bulk of the article.  Much work is done in the text by pointing the reader instead to reference [17].

While therefore I cannot quite commit to judging the success that the authors have with their program (though I have no reason to doubt it), I can attest that the broader program of "unfolded EoMs subject to duality relations" is indeed most central for 11d SuGra due to the following result (not mentioned in the manuscript):

With the full field content of 11D sugra considered (hence crucially including also the gravitino, which the present authors seem to disregard) and thus working on super-spacetime, it turns out that the full (not just linearized) unfolded EoMs of the superized 3-form field (in other parts of the literature  known as the "duality-symmetric" formulation of the C-field Bianchi identity) already implies *all* of the SuGra equations of motion, to all order and including the duality between the 3-form and the 6-form. [Thm . 3.1 of doi:10.1007/JHEP07(2024)082]

For this reason I am sure that the manuscript under consideration is onto something important and eventually worth publishing. I would ask the authors to add a clearer description of their accomplishment: The text should make it clearer: What exactly is assumed, what exactly does it mean to "propose" in this context and how exactly is these proposals are being vindicated.

Requested changes

Besides the points raised in the report, here is a list of minor comments, going linearly through the text:

p. 5:  grammar: "on empty column"

p. 6: "In this paper..." This might better be said much earlier

p. 6: "we apply the unfolded formalism [55,56]": 
not sure that the terminology "unfolded" already appears in [55,56].  Better to also point to [63,64] or similar, for clarity.

p. 7: grammar: "the E11"

p. 7:  "we write \Psi_{4|3,2,2,1}" the last ":
"1" seems to be a typo and should be omitted

pp. 8, 9: the statements about the fields (such "is a higher dual", or "plays a role in gauge supergravity") should be referenced

p. 9: Sentences like 
  "The full non-linear equations for the fields follow uniquely from the non-linear realisation."
do not quite parse. An equation can follow from another equation or generally from another logical statement, but not from a mere thing (like a coset space aka "non-linear realization").

p. 13 "although it is very well known":
still, give a reference

p. 14: "The first two unfolded equations are...":
It may be worth saying *why* it is the case that this "are the first two equations". I gather it is an Ansatz that is justified by its implications. Generally, the rules of the game of "unfolding", as used here, would be worth stating more explicitly.

p. 18 equation (3.40):
here it may be worth re-amplifying that on the right the equation for F7 is only shown to linear order: To higher order it is not F7 itself but  only the non-linear combination F7 - 1/2 A3 F4  that has a 6-form potential.

p. 18 equation (3.24):
here and from looking at reference [17], it seems that the objects F^(n) must be understood as Taylor coefficients around a chosen spacetime point.(?)
Which would mean that the claims about reproducing the (linearized) equations of motion of SuGra pertain only to a formal neighbourhood of any one point.
This needs clarification.

p. 20 "its descendants do not need to be completely traceless on-shel" 
Best to add the argument or reference for how this claim comes about.

Recommendation

Ask for major revision

---

## Round 1 · Referee Report · Anonymous (Referee 2) · 2025-2-6

Strengths

  1. The paper contains new results
  2. The unfloded formulation is clearly and nicely explained
  3. The paper is well-written and the results are clearly explained

Weaknesses

  1. A few statements are conjectural
  2. In some cases the conjectural nature of the claims in not clearly stated

Report

The paper deals with the unfolded formulation of the equations for the fields that occur in the E11 non-linear realisation. This gives rise to an infinite number of fields that do not belong to E11, and the authors conjecture a possible origin of such fields as representations of E11. Using the unfolded formulation, one constructs gauge invariant quantities which the authors use to propose at linearised level an infinite set of duality relations between the fields in E11. I am concerned about how the duality relations arise in E11 already at the level of the 3-form and 6-form. These relations are consistent with the E11 symmetry, but as far as I understand are not really derived from the dynamics, but rather imposed. I would like the authors to make more clear comments on this issue, which I think is very important and would be beneficial for the paper.

Requested changes

  1. On the third paragraph of the introduction, I would like the authors to explain in more detail how the duality relations can be read or derived in the non-linear realisation.
  2. I would like the authors to motivate more the statement below eqs. (4.45), (5.10) and (5.29) that the proportionality coefficients in the duality relations are fixed by E11 symmetry.

Recommendation

Ask for minor revision

---

## Round 2 · Author Response

[RESPONSE TO REFEREE 1]
We thank Referee 1 for their careful reading of the manuscript and suggestions for the second version of our article. Here we summarise the changes that we have implemented.
First of all, following the suggestions of the referee, we have clarified precisely what we have achieved in this paper. We have rewritten various parts of the draft in order to state more clearly our main results and how they were obtained. Referee 1 suggests that Sections 1 and 2 amount to "a broad review of the E11 program in general, of the kind one might write for a grant proposal, rather than a concrete motivation of and introduction to the bulk of the article." We feel, after implementing many changes, that our introduction (Section 1) now properly motivates and introduces the article. In particular, we have concretely stated our objectives several times and we have given a detailed outline of the paper. On the other hand, Section 2 remains largely unchanged since our aim is not to review the E11 program extensively (as has already been done in arXiv:1609.06863 long ago) but rather to propose an extension of E11 theory featuring gauge-covariant duality relations. We have not worked out precisely how to introduce these extra fields in a way that respects E11 symmetry, and this is left to future work, but we now explain in our article that we should look for extra fields inside representations of E11 due to the natural action of E11 on such representations.
We would also like to defend our frequent citation of arXiv:1502.07909 (reference 17 in the first version, 23 in the second version). Our article should really be considered to be a sequel to that paper, and we have taken a lot of its ideas much further and extended its analysis to the infinite set of dual fields in the E11 theory. We concede that we should have been more transparent with what we are doing, and we believe that our modifications have achieved this.
In our new introduction, we have cited other articles on duality-symmetric formulations of supergravity, including the paper that was mentioned in the report. We have explained that the fermionic degrees of freedom are not included in E11 theory, but they are included in the paper that is mentioned in the report, and we cite a number of key papers that have tried to incorporate fermions into the E10 and E11 programs.
[RESPONSE TO REFEREE 2]
This report has a slightly different focus to the first, but it does share the criticism that we have not been clear about the conjectural nature of some of our statements. We thank Referee 2 for also bringing this to our attention. We explain in the new version of our paper that E11 symmetry determines the "form" (i.e. the tensor structure and precise combination of terms) of the equations of motion and duality relations, including the duality relation between the three-form and six-form fields that was specifically pointed out.
As requested, we have made comments throughout the new version of this paper to explain how precisely the equations of the E11 theory are obtained, and we frequently refer the reader to the original E11 papers where this was done long ago.
We thank Referee 1 for their careful reading of the manuscript and suggestions for the second version of our article. Here we summarise the changes that we have implemented.
First of all, following the suggestions of the referee, we have clarified precisely what we have achieved in this paper. We have rewritten various parts of the draft in order to state more clearly our main results and how they were obtained. Referee 1 suggests that Sections 1 and 2 amount to "a broad review of the E11 program in general, of the kind one might write for a grant proposal, rather than a concrete motivation of and introduction to the bulk of the article." We feel, after implementing many changes, that our introduction (Section 1) now properly motivates and introduces the article. In particular, we have concretely stated our objectives several times and we have given a detailed outline of the paper. On the other hand, Section 2 remains largely unchanged since our aim is not to review the E11 program extensively (as has already been done in arXiv:1609.06863 long ago) but rather to propose an extension of E11 theory featuring gauge-covariant duality relations. We have not worked out precisely how to introduce these extra fields in a way that respects E11 symmetry, and this is left to future work, but we now explain in our article that we should look for extra fields inside representations of E11 due to the natural action of E11 on such representations.
We would also like to defend our frequent citation of arXiv:1502.07909 (reference 17 in the first version, 23 in the second version). Our article should really be considered to be a sequel to that paper, and we have taken a lot of its ideas much further and extended its analysis to the infinite set of dual fields in the E11 theory. We concede that we should have been more transparent with what we are doing, and we believe that our modifications have achieved this.
In our new introduction, we have cited other articles on duality-symmetric formulations of supergravity, including the paper that was mentioned in the report. We have explained that the fermionic degrees of freedom are not included in E11 theory, but they are included in the paper that is mentioned in the report, and we cite a number of key papers that have tried to incorporate fermions into the E10 and E11 programs.
[RESPONSE TO REFEREE 2]
This report has a slightly different focus to the first, but it does share the criticism that we have not been clear about the conjectural nature of some of our statements. We thank Referee 2 for also bringing this to our attention. We explain in the new version of our paper that E11 symmetry determines the "form" (i.e. the tensor structure and precise combination of terms) of the equations of motion and duality relations, including the duality relation between the three-form and six-form fields that was specifically pointed out.
As requested, we have made comments throughout the new version of this paper to explain how precisely the equations of the E11 theory are obtained, and we frequently refer the reader to the original E11 papers where this was done long ago.

---

## Round 2 · List of Changes

[SPECIFIC CHANGES FOR REFEREE 1]
(PAGE 3) We have provided a clarified introduction (Section 1) where we have stated what we achieved, how we did it, and why it was relevant for us to do so. The first two paragraphs are completely new. Aspects of unfolding were explained in the text and in footnote 1, and we explained the origin of exterior differential systems in footnote 2. This concerns some comments made in the main report, and also the requested change [p.6: ``In this paper..." This might better be said much earlier].
(PAGE 4) It is explained precisely how to construct the equations of motion and duality relations in the E11 theory. This is explained in the main text (especially the second paragraph that begins with ``The large symmetries...'') and also in footnote 3. However, the point of our paper is not to rederive these equations but rather to propose extensions of them, so we did not compute them from scratch. This concerns some comments made in the main report, and also the requested change [p.9: Sentences like ``The full non-linear equations for the fields follow uniquely from the non-linear realisation.'' do not quite parse. An equation can follow from another equation or generally from another logical statement, but not from a mere thing (like a coset space aka ``non-linear realization'').].
(PAGES 5 & 6) In the paragraph beginning "There are various...", we have added a number of references on the different interesting formulations of the equations of motion for supergravity, including the duality-symmetric superspace formulation of which we were informed by Referee1. Not only this, but we highlighted the fact that E theory at present only contains bosonic degrees of freedom. This paragraph continues onto page 6, and we cited some progress that has been made to incorporate the fermionic degrees of supergravity into the E10 and E11 programs. This concerns some comments made in the main report.
(PAGE 6) We corrected a typo in the paragraph beginning "Parent actions...". This concerns the requested change [p.5: grammar: "on empty column"].
(PAGE 7) A rewritten outline of the paper explains precisely what our procedure will be when we unfold each field. Towards the end of the paragraph that begins "In Section 5, we...", we followed the suggestion of the referee by explaining that our proposed duality relations must (and do) "match" those that have already been found in the E11 theory. This concerns some comments made in the main report.
(PAGE 7) We have followed the recommendation of Referee 1 concerning certain references. This concerns the requested change [p.6: "we apply the unfolded formalism [55,56]": not sure that the terminology "unfolded" already appears in [55,56]. Better to also point to [63,64] or similar, for clarity.].
(PAGE 8) We corrected a grammatical error and a typo noticed by Referee 1. This concerns the requested changes [p.7: grammar: "the E11"] and [p.7: "we write $\Psi_{4|3,2,2,1}$" the last "1" seems to be a typo and should be omitted].
(PAGE 10) We cited the appropriate literature arXiv:0705.0752 (reference 27 in the new version). This concerns the requested change [pp.8,9: the statements about the fields (such "is a higher dual", or "plays a role in gauge supergravity") should be referenced].
(PAGE 11) As we explained in detail in our new introduction and at various points in the main body, in the paragraph starting "The form of...", we highlighted the idea that E11 symmetry determines the form (i.e. the tensor structure and the precise combination of terms) in the duality relations and equations of motion of the E11 theory. This concerns some comments made in the main report.
(PAGE 14) At the beginning of Section 3.2, we gave two references (75 and 76 in the new version) for the unfolding of linearised gravity, something that we described as "well-known". We also motivated the introduction of unfolded variables by referring back to our earlier discussion of the unfolded formalism in equation (3.5). This concerns the requested change [\textit{p.13 "although it is very well known": still, give a reference}].
(PAGE 15) In the first line of the paragraph that begins "Writing the schematic...", we explained that our unfolded equations are now being written completely. In footnote 6, we explained precisely why the unfolded equations are called "first", "second", and so on. In a nutshell, they are related to Cartan's first and second structure equations. This concerns the requested change [p.14: "The first two unfolded equations are...": It may be worth saying *why* it is the case that this "are the first two equations". I gather it is an Ansatz that is justified by its implications. Generally, the rules of the game of "unfolding", as used here, would be worth stating more explicitly].
(PAGES 20 & 21) Footnote 8 explains that we are truly working at the linearised level and that the non-linear completion of the field strength of the six-form potential would feature a necessary non-linear term. This is also explained in detail at the end of our conclusions in Section 8 on page 80, beginning from the sentence which starts "In the full non-linear theory..." Footnote 9 explains in more detail the idea that the higher zero-forms are interpreted as Taylor coefficients for fields expanded in some neighbourhood around a point in space-time. We would also like to point out to the referee equation (5.82) on page 43 and the discussion around it, where we explained how this is relevant in the context of dual fields -- see arXiv:1502.07909. (This again justifies our frequent citation of that paper.) In footnote 10 on page 21, we clarified once again that we are considering linearised duality relations, with some additional comments about generalised coordinates. This concerns some comments in the main report and also the requested changes [p.18 equation (3.40): here it may be worth re-amplifying that on the right the equation for F7 is only shown to linear order: To higher order it is not F7 itself but only the non-linear combination F7 - 1/2 A3 F4 that has a 6-form potential.] and [p.18 equation (3.24): here and from looking at reference [17], it seems that the objects $F^{(n)}$ must be understood as Taylor coefficients around a chosen spacetime point.(?) Which would mean that the claims about reproducing the (linearized) equations of motion of SuGra pertain only to a formal neighbourhood of any one point. This needs clarification.].
(PAGE 22) As recommended, just under equation (4.2), we gave a reference for the claim that higher zero-forms do not need to be completely traceless for a higher dual field. This concerns the requested change [p.20 "its descendants do not need to be completely traceless on-shell" Best to add the argument or reference for how this claim comes about.].
(PAGE 28) In the first four lines of page 28, we explained, as already explained in detail in the new introduction, that E11 symmetry determines the tensor structure and the precise combination of terms in the equations of the theory. At the linearised level, the proportionality coefficients in the duality relations are irrelevant as they can always be changes by a rescaling of the fields. They will only be fixed in the non-linear equations. This concerns the requested change [p.9: Sentences like "The full non-linear equations for the fields follow uniquely from the non-linear realisation." do not quite parse. An equation can follow from another equation or generally from another logical statement, but not from a mere thing (like a coset space aka "non-linear realization").].
(PAGE 30) In Section 5, just before Section 5.1 begins, we explained (as in the outline of the paper in the introduction) the procedure that we followed in order to unfold each of the higher dual fields in E11. This concerns some comments made in the main report.
[SPECIFIC CHANGES FOR REFEREE 2]
(PAGE 4) It is explained precisely how to construct the equations of motion and duality relations in the E11 theory. This is explained in the main text (especially the second paragraph which begins "The large symmetries...") and also in footnote 3. However, the point of our paper is not to rederive these equations but rather to propose extensions of them, so we did not compute them from scratch. This concerns the requested change [1. On the third paragraph of the introduction, I would like the authors to explain in more detail how the duality relations can be read or derived in the non-linear realisation.].
(PAGE 11) As we explained in detail in our new introduction and at various points in the main body, in the paragraph starting "The form of...", we highlighted the idea that E11 symmetry determines the form (i.e. the tensor structure and the precise combination of terms) in the duality relations and equations of motion of the E11 theory. This concerns the requested change [1. On the third paragraph of the introduction, I would like the authors to explain in more detail how the duality relations can be read or derived in the non-linear realisation.].
(PAGE 28) In the first four lines of page 28, we explained that E11 symmetry determines the structure and the combination of terms in the equations of the theory. At the linearised level, the proportionality coefficients in the duality relations are irrelevant as they can always be changes by a rescaling of the fields. They will only be fixed in the non-linear equations. This concerns the requested change [2. I would like the authors to motivate more the statement below eqs. (4.45), (5.10) and (5.29) that the proportionality coefficients in the duality relations are fixed by E11 symmetry.].
(PAGE 31) We explained how E11 symmetry determines the structure of the equations of the theory, but since we are working at the linearised level, any coefficient can be absorbed by a redefinition of the fields. This concerns the requested change [2. I would like the authors to motivate more the statement below eqs. (4.45), (5.10) and (5.29) that the proportionality coefficients in the duality relations are fixed by E11 symmetry.].
(PAGE 34) Under equation (5.29), we gave another explanation along the lines of the ones on pages 28 and 31. This concerns the requested change [2. I would like the authors to motivate more the statement below eqs. (4.45), (5.10) and (5.29) that the proportionality coefficients in the duality relations are fixed by E11 symmetry.].
[ADDITIONAL CHANGES]
(PAGE 29) A new paragraph starts with "To be precise...". Here we explained how, after imposing gauge-invariance leading to constraints on the gauge parameters, we are still able to eliminate some, but not all, extra fields. This led to the duality relation that was obtained in the previous work using a different method -- see arXiv:1502.07909.
(PAGE 30) Immediately before Section 5.1, we have clarified the interpretation of curvature constraints as equations of motion, and we explained that the algebraic relations between curvatures really do give us dual equations of motion.
(PAGE 34) At the top of the page we elaborated on residual gauge symmetry.
(PAGE 3) We have provided a clarified introduction (Section 1) where we have stated what we achieved, how we did it, and why it was relevant for us to do so. The first two paragraphs are completely new. Aspects of unfolding were explained in the text and in footnote 1, and we explained the origin of exterior differential systems in footnote 2. This concerns some comments made in the main report, and also the requested change [p.6: ``In this paper..." This might better be said much earlier].
(PAGE 4) It is explained precisely how to construct the equations of motion and duality relations in the E11 theory. This is explained in the main text (especially the second paragraph that begins with ``The large symmetries...'') and also in footnote 3. However, the point of our paper is not to rederive these equations but rather to propose extensions of them, so we did not compute them from scratch. This concerns some comments made in the main report, and also the requested change [p.9: Sentences like ``The full non-linear equations for the fields follow uniquely from the non-linear realisation.'' do not quite parse. An equation can follow from another equation or generally from another logical statement, but not from a mere thing (like a coset space aka ``non-linear realization'').].
(PAGES 5 & 6) In the paragraph beginning "There are various...", we have added a number of references on the different interesting formulations of the equations of motion for supergravity, including the duality-symmetric superspace formulation of which we were informed by Referee1. Not only this, but we highlighted the fact that E theory at present only contains bosonic degrees of freedom. This paragraph continues onto page 6, and we cited some progress that has been made to incorporate the fermionic degrees of supergravity into the E10 and E11 programs. This concerns some comments made in the main report.
(PAGE 6) We corrected a typo in the paragraph beginning "Parent actions...". This concerns the requested change [p.5: grammar: "on empty column"].
(PAGE 7) A rewritten outline of the paper explains precisely what our procedure will be when we unfold each field. Towards the end of the paragraph that begins "In Section 5, we...", we followed the suggestion of the referee by explaining that our proposed duality relations must (and do) "match" those that have already been found in the E11 theory. This concerns some comments made in the main report.
(PAGE 7) We have followed the recommendation of Referee 1 concerning certain references. This concerns the requested change [p.6: "we apply the unfolded formalism [55,56]": not sure that the terminology "unfolded" already appears in [55,56]. Better to also point to [63,64] or similar, for clarity.].
(PAGE 8) We corrected a grammatical error and a typo noticed by Referee 1. This concerns the requested changes [p.7: grammar: "the E11"] and [p.7: "we write $\Psi_{4|3,2,2,1}$" the last "1" seems to be a typo and should be omitted].
(PAGE 10) We cited the appropriate literature arXiv:0705.0752 (reference 27 in the new version). This concerns the requested change [pp.8,9: the statements about the fields (such "is a higher dual", or "plays a role in gauge supergravity") should be referenced].
(PAGE 11) As we explained in detail in our new introduction and at various points in the main body, in the paragraph starting "The form of...", we highlighted the idea that E11 symmetry determines the form (i.e. the tensor structure and the precise combination of terms) in the duality relations and equations of motion of the E11 theory. This concerns some comments made in the main report.
(PAGE 14) At the beginning of Section 3.2, we gave two references (75 and 76 in the new version) for the unfolding of linearised gravity, something that we described as "well-known". We also motivated the introduction of unfolded variables by referring back to our earlier discussion of the unfolded formalism in equation (3.5). This concerns the requested change [\textit{p.13 "although it is very well known": still, give a reference}].
(PAGE 15) In the first line of the paragraph that begins "Writing the schematic...", we explained that our unfolded equations are now being written completely. In footnote 6, we explained precisely why the unfolded equations are called "first", "second", and so on. In a nutshell, they are related to Cartan's first and second structure equations. This concerns the requested change [p.14: "The first two unfolded equations are...": It may be worth saying *why* it is the case that this "are the first two equations". I gather it is an Ansatz that is justified by its implications. Generally, the rules of the game of "unfolding", as used here, would be worth stating more explicitly].
(PAGES 20 & 21) Footnote 8 explains that we are truly working at the linearised level and that the non-linear completion of the field strength of the six-form potential would feature a necessary non-linear term. This is also explained in detail at the end of our conclusions in Section 8 on page 80, beginning from the sentence which starts "In the full non-linear theory..." Footnote 9 explains in more detail the idea that the higher zero-forms are interpreted as Taylor coefficients for fields expanded in some neighbourhood around a point in space-time. We would also like to point out to the referee equation (5.82) on page 43 and the discussion around it, where we explained how this is relevant in the context of dual fields -- see arXiv:1502.07909. (This again justifies our frequent citation of that paper.) In footnote 10 on page 21, we clarified once again that we are considering linearised duality relations, with some additional comments about generalised coordinates. This concerns some comments in the main report and also the requested changes [p.18 equation (3.40): here it may be worth re-amplifying that on the right the equation for F7 is only shown to linear order: To higher order it is not F7 itself but only the non-linear combination F7 - 1/2 A3 F4 that has a 6-form potential.] and [p.18 equation (3.24): here and from looking at reference [17], it seems that the objects $F^{(n)}$ must be understood as Taylor coefficients around a chosen spacetime point.(?) Which would mean that the claims about reproducing the (linearized) equations of motion of SuGra pertain only to a formal neighbourhood of any one point. This needs clarification.].
(PAGE 22) As recommended, just under equation (4.2), we gave a reference for the claim that higher zero-forms do not need to be completely traceless for a higher dual field. This concerns the requested change [p.20 "its descendants do not need to be completely traceless on-shell" Best to add the argument or reference for how this claim comes about.].
(PAGE 28) In the first four lines of page 28, we explained, as already explained in detail in the new introduction, that E11 symmetry determines the tensor structure and the precise combination of terms in the equations of the theory. At the linearised level, the proportionality coefficients in the duality relations are irrelevant as they can always be changes by a rescaling of the fields. They will only be fixed in the non-linear equations. This concerns the requested change [p.9: Sentences like "The full non-linear equations for the fields follow uniquely from the non-linear realisation." do not quite parse. An equation can follow from another equation or generally from another logical statement, but not from a mere thing (like a coset space aka "non-linear realization").].
(PAGE 30) In Section 5, just before Section 5.1 begins, we explained (as in the outline of the paper in the introduction) the procedure that we followed in order to unfold each of the higher dual fields in E11. This concerns some comments made in the main report.
[SPECIFIC CHANGES FOR REFEREE 2]
(PAGE 4) It is explained precisely how to construct the equations of motion and duality relations in the E11 theory. This is explained in the main text (especially the second paragraph which begins "The large symmetries...") and also in footnote 3. However, the point of our paper is not to rederive these equations but rather to propose extensions of them, so we did not compute them from scratch. This concerns the requested change [1. On the third paragraph of the introduction, I would like the authors to explain in more detail how the duality relations can be read or derived in the non-linear realisation.].
(PAGE 11) As we explained in detail in our new introduction and at various points in the main body, in the paragraph starting "The form of...", we highlighted the idea that E11 symmetry determines the form (i.e. the tensor structure and the precise combination of terms) in the duality relations and equations of motion of the E11 theory. This concerns the requested change [1. On the third paragraph of the introduction, I would like the authors to explain in more detail how the duality relations can be read or derived in the non-linear realisation.].
(PAGE 28) In the first four lines of page 28, we explained that E11 symmetry determines the structure and the combination of terms in the equations of the theory. At the linearised level, the proportionality coefficients in the duality relations are irrelevant as they can always be changes by a rescaling of the fields. They will only be fixed in the non-linear equations. This concerns the requested change [2. I would like the authors to motivate more the statement below eqs. (4.45), (5.10) and (5.29) that the proportionality coefficients in the duality relations are fixed by E11 symmetry.].
(PAGE 31) We explained how E11 symmetry determines the structure of the equations of the theory, but since we are working at the linearised level, any coefficient can be absorbed by a redefinition of the fields. This concerns the requested change [2. I would like the authors to motivate more the statement below eqs. (4.45), (5.10) and (5.29) that the proportionality coefficients in the duality relations are fixed by E11 symmetry.].
(PAGE 34) Under equation (5.29), we gave another explanation along the lines of the ones on pages 28 and 31. This concerns the requested change [2. I would like the authors to motivate more the statement below eqs. (4.45), (5.10) and (5.29) that the proportionality coefficients in the duality relations are fixed by E11 symmetry.].
[ADDITIONAL CHANGES]
(PAGE 29) A new paragraph starts with "To be precise...". Here we explained how, after imposing gauge-invariance leading to constraints on the gauge parameters, we are still able to eliminate some, but not all, extra fields. This led to the duality relation that was obtained in the previous work using a different method -- see arXiv:1502.07909.
(PAGE 30) Immediately before Section 5.1, we have clarified the interpretation of curvature constraints as equations of motion, and we explained that the algebraic relations between curvatures really do give us dual equations of motion.
(PAGE 34) At the top of the page we elaborated on residual gauge symmetry.

---

## Editorial Decision

published